# Investigating molecular crowding within nuclear pores using polarization-PALM

Guo Fu[1], Li-Chun Tu[1†], Anton Zilman[2,3], Siegfried M Musser[1]*

[1]Department of Molecular and Cellular Medicine, College of Medicine, The Texas A&M University Health Science Center, College Station, United States; [2]Department of Physics, University of Toronto, Toronto, Canada; [3]Institute for Biomaterials and Biomedical Engineering, University of Toronto, Toronto, Canada

**Abstract** The key component of the nuclear pore complex (NPC) controlling permeability, selectivity, and the speed of nucleocytoplasmic transport is an assembly of natively unfolded polypeptides, which contain phenylalanine-glycine (FG) binding sites for nuclear transport receptors. The architecture and dynamics of the FG-network have been refractory to characterization due to the paucity of experimental methods able to probe the mobility and density of the FG-polypeptides and embedded macromolecules within intact NPCs. Combining fluorescence polarization, super-resolution microscopy, and mathematical analyses, we examined the rotational mobility of fluorescent probes at various locations within the FG-network under different conditions. We demonstrate that polarization PALM (p-PALM) provides a rich source of information about low rotational mobilities that are inaccessible with bulk fluorescence anisotropy approaches, and anticipate that p-PALM is well-suited to explore numerous crowded cellular environments. In total, our findings indicate that the NPC's internal organization consists of multiple dynamic environments with different local properties.

DOI: https://doi.org/10.7554/eLife.28716.001

*For correspondence:
smusser@tamhsc.edu

Present address: [†]RNA Therapeutics Institute, University of Massachusetts Medical School, Worcester, United States

Competing interests: The authors declare that no competing interests exist.

## Introduction

Intracellular environments are highly crowded, with typical local macromolecular concentrations of ~80–400 mg/mL, and some cellular environments contain only ~50% water (*Kuznetsova et al., 2014*). Under crowded conditions, excluded volume effects and local interactions can change protein activities by over an order of magnitude compared with the 'dilute' solutions typically used for most in vitro studies (*Aumiller et al., 2014*). Crowded conditions can affect protein folding, structure, shape, conformational stability and dynamics, binding interactions, and enzymatic activity (*Kuznetsova et al., 2014*; *Zhou et al., 2008*). Biological polymers play central roles in generating a variety of crowded environments. For example, the polymers in mucus, the extracellular matrix, the cytoskeleton, the vitreous humor of the eye, and the Nuclear Pore Complex (NPC) produce complex environments that restrict diffusion and trap molecules (*Leterrier, 2001*; *Lieleg and Ribbeck, 2011*). In addition, the numerous distinct bodies/granules within the nucleus and the cytoplasm have been interpreted to form via a phase separation-like mechanism due to high local concentrations of self-cohesive nucleic acid and/or intrinsically disordered protein polymers (*Aumiller et al., 2014*; *Toretsky and Wright, 2014*). Characterization of the physical, structural, dynamical, and functional properties of these crowded environments remains challenging due to the dearth of appropriate tools that are needed to investigate the complexity and heterogeneity of these environments on the nanoscale.

One example of a crowded environment is the pore of the NPC, which mediates bidirectional traffic between the cytoplasm and the nucleoplasm of eukaryotic cells. The translocation passageway of the NPC is occupied by hundreds of intrinsically disordered polypeptides (*Lim et al., 2008*;

**eLife digest** Most of the genetic material inside an animal cell is enclosed within a compartment called the nucleus. This compartment is separated from the rest of the cell by the nuclear envelope, a double-membrane structure containing thousands of pores that selectively allow certain molecules (collectively referred to as cargo) to enter and exit the nucleus.

The movement of cargo through the pores is controlled by large groups of proteins called nuclear pore complexes. The pore is at the center of the complex and is filled by a selective barrier made of an extensive network of flexible proteins known as the FG-network. Other proteins known as nuclear transport receptors bind to the proteins in the FG-network and carry cargos through the barrier.

The properties of the nuclear pore barrier and how it rapidly selects the right cargos have been difficult to study, in part, because the barrier network is constantly changing and is crowded with hundreds of transport receptors. New techniques are needed to investigate such highly crowded environments inside cells. Now, Fu et al. use a technique called polarization photoactivated localization microscopy (p-PALM) to explore the molecular crowding within the nuclear pore barrier in human cells. This technique measures the freedom with which a single molecule embedded in the network can rotate, providing information about the local environment. In a crowded environment, it is harder for the probe molecule to rotate as it is more likely to bump into other molecules.

Fu et al. found that there are different levels of crowding within the barrier. This is consistent with previous ideas of how the pore barrier could work, which propose that the nuclear transport receptors are less tightly packed in the center of the FG-network. This enables transport receptor and cargo complexes to move more rapidly through the center of the pore. The molecular crowding in the barrier of nuclear pores parallels that observed in other cellular compartments that also rely on assemblies of proteins with flexible structures. Thus, future work using p-PALM is expected to reveal more details about the biophysical properties of nuclear pores as well as those of other structures inside cells.

DOI: https://doi.org/10.7554/eLife.28716.002

*Peleg and Lim, 2010*; *Suntharalingam and Wente, 2003*), 50–100 nuclear transport receptors (NTRs) (*Lowe et al., 2015*; *Tokunaga et al., 2008*), and protein and nucleic acid cargo complexes moving in opposite directions. NTRs are classified as importins or exportins, reflecting their ability to carry cargos into or out of the nucleus, respectively (for reviews, see [*Chook and Süel, 2011*; *Güttler and Görlich, 2011*; *Jamali et al., 2011*; *Stewart, 2007*; *Wente and Rout, 2010*]). On the nuclear side, RanGTP promotes disassembly of NTR/cargo import complexes, freeing the cargo and allowing NTRs to diffuse back to the cytoplasm (*Chook and Blobel, 2001*; *Izaurralde et al., 1997*; *Rexach and Blobel, 1995*; *Siomi et al., 1997*). NTR/cargo/RanGTP export complexes are disassembled on the cytoplasmic side after GTP hydrolysis, which results from interactions with RanGAP and a Ran-binding protein (RanBP) (*Bischoff and Görlich, 1997*; *Bischoff et al., 1994*; *Güttler and Görlich, 2011*; *Kutay et al., 1997a*; *Okamura et al., 2015*). Many of these assembly and disassembly reactions are coordinated to occur at the cytoplasmic and nucleoplasmic exits of the NPC's central pore (*Sun et al., 2013*; *Sun et al., 2008*). Exactly how cargo complexes are specifically recognized and yet rapidly migrate in milliseconds (*Dange et al., 2008*; *Grünwald and Singer, 2010*; *Kubitscheck et al., 2005*; *Tu et al., 2013*; *Yang et al., 2004*; *Yang and Musser, 2006a*) through the NPC's crowded environment remains enigmatic.

NPCs are large (~60–120 MDa) structures with octagonal rotational symmetry. They are comprised of ~30 different nuclear pore proteins (nucleoporins, or Nups), each of which are thought to be present in an integer multiple of eight copies (*Cronshaw et al., 2002*; *Fahrenkrog and Aebi, 2003*; *Mi et al., 2015*; *Ori et al., 2013*; *Rout and Aitchison, 2001*). The vertebrate NPC has an outer diameter of ~120 nm, and extends ~200 nm along the transport axis (*Fahrenkrog and Aebi, 2003*; *Stoffler et al., 1999*). Eight flexible filaments extend ~50 nm into the cytoplasm, and an additional eight filaments extend ~75 nm into the nucleoplasm and terminate in a ring to form the nuclear basket (*Fahrenkrog and Aebi, 2003*; *Stoffler et al., 2003*). In humans, the hourglass-shaped central pore has a minimum diameter of ~50 nm and a length of ~85 nm (*Maimon et al., 2012*).

Within this large pore and decorating its openings is a network of ~200–250 intrinsically disordered polypeptides, which generates a permeability barrier impeding macromolecular transport (*Lim et al., 2008*; *Ori et al., 2013*; *Peleg and Lim, 2010*; *Suntharalingam and Wente, 2003*) and which is particularly selective against larger cargos (*Mohr et al., 2009*; *Popken et al., 2015*; *Ribbeck and Görlich, 2001*; *Timney et al., 2016*). These disordered polypeptides contain, in total, 3000–4000 phenylalanine-glycine (FG) repeats to which NTRs transiently bind as they carry cargos through NPCs (*Cronshaw et al., 2002*; *Denning et al., 2003*; *Rout et al., 2000*; *Strawn et al., 2004*; *Tran and Wente, 2006*). We term this assembly of intrinsically disordered FG-containing polypeptides the FG-network.

Each FG-containing nucleoporin (FG-Nup) has a globular anchor domain that is embedded in or attached to the NPC scaffold, and thus, it acts as an anchor point for the flexible and mobile FG-domain. The FG-repeat motifs are separated by short (~10–20 amino acid residues), largely hydrophilic segments (*Denning and Rexach, 2007*; *Yamada et al., 2010*). The FG-domains do not form readily recognizable secondary structures, but rather are more appropriately described as flexible polymers with alternating hydrophobic and hydrophilic domains (*Lim et al., 2006*; *Yamada et al., 2010*). The FG-network is sufficiently fluid and mobile that it is rapidly displaced by transporting cargos, which can be up to ~40 nm in diameter (*Frey and Görlich, 2009*; *Hough et al., 2015*; *Lim et al., 2007*; *Milles et al., 2015*; *Panté and Kann, 2002*).

The 'polymer brush' (*Lim et al., 2006*; *Peleg and Lim, 2010*) and 'hydrogel' (*Frey and Görlich, 2007*; *Frey et al., 2006*) models are the most widely cited descriptions of the biophysical nature of FG-polypeptide assemblies. These models are two extremes in the model space describing the potential morphologies and properties of the FG-Nup assemblies within the NPC (*Eisele et al., 2013*; *Vovk et al., 2016*). The polymer brush model postulates that the FG-polypeptides are largely non-interacting (beyond steric repulsion), relatively extended and minimally entangled (*Lim et al., 2006*; *Peleg and Lim, 2010*), and their spatial assemblies are stabilized mostly by entropic forces (*Vovk et al., 2016*). The hydrogel model posits that the FG-polypeptides exhibit significant inter- and intra-strand cohesiveness via FG-FG interactions, which results in a connected dense network (*Frey and Görlich, 2007*; *Frey and Görlich, 2009*; *Frey et al., 2006*; *Hülsmann et al., 2012*). A hybrid, two-gate model postulates brush-like structures on both cytoplasmic and nuclear sides of the NPC, suitable for binding and (dis)assembly reactions, and a central cohesive structure in the center of the pore that provides the permeability barrier (*Patel et al., 2007*). The spatial distribution of functional activities in this two-gate model is supported by single molecule transport results (*Sun et al., 2013*; *Tu et al., 2013*). Quantitative modeling of FG-polypeptide behavior predicts a smooth transition between brush-like and gel-like behaviors in response to relatively small changes in physical properties and favors a picture intermediate between a brush and a gel (*Vovk et al., 2016*). The magnitude of the inter- and intra-chain cohesiveness that differentiates these two descriptions could be different for different FG-polypeptides, or different segments of the same FG-polypeptides, in distinct spatial locations within the NPC (*Vovk et al., 2016*). Avidity calculations indicate that the multivalent affinities of NTRs depend critically upon the local free FG-repeat concentration (*Tu et al., 2013*). In agreement with these predictions, experimental results indicate that some sub-populations of NTRs have very long dissociation times, and therefore, they potentially can form an integral part of the permeability barrier (*Kapinos et al., 2014*; *Lowe et al., 2015*; *Schleicher et al., 2014*). Taken together, these findings suggest that the FG-polypeptides and NTRs act together to form different local environments with different properties within the NPC (*Coalson et al., 2015*; *Ghavami et al., 2014*; *Lowe et al., 2015*; *Eskandari Nasrabad et al., 2016*; *Osmanović et al., 2013*; *Tagliazucchi et al., 2013*; *Yamada et al., 2010*).

Considering the uncertainty in the structural arrangement of and interactions between FG-polypeptides, and knowing that many tens to over a hundred macromolecules (including NTRs, Ran, and cargos) interact with the FG-network during steady-state transport (*Abu-Arish et al., 2009*; *Lowe et al., 2015*; *Tokunaga et al., 2008*), developing a general picture of FG-polypeptide distributions and local crowding conditions, and discerning their functional effects on cargo transport, is a challenging problem, but nevertheless essential for establishing the mechanism of nucleocytoplasmic transport and its implications. Here, we used the super-resolution approach photoactivated localization microscopy (PALM) (*Betzig et al., 2006*) to probe the locations of a number of FG-polypeptides and transport-related proteins within the NPC. Our main focus, however, was on using polarization PALM (p-PALM) (*Gould et al., 2008*) to measure rotational mobility, which is sensitive to local

crowding conditions, and which enables probing of the local properties of crowded macromolecular assemblies that are currently inaccessible by other means. Crucially, we developed a theoretical model that enables detailed analysis of the experimental p-PALM data in terms of rotational diffusion constants. While numerous previous super-resolution approaches on NPCs utilized fixed samples, and most concentrated on scaffold structural questions (*Löschberger et al., 2014*, *2012*; *Lowe et al., 2015*; *Otsuka et al., 2016*; *Pleiner et al., 2016*; *Szymborska et al., 2013*; *Winterflood and Ewers, 2014*), the NPCs in our samples were fully functional since our goal was to probe the properties of the FG-network, which is intrinsically dynamic. The results of our analysis of protein localization and local mobility within the NPC demonstrate that the FG-network is heterogeneous with regard to molecular crowding and that this can be influenced by the presence of embedded proteins, which argues for a remarkable complexity in nucleocytoplasmic trafficking pathways and their regulation.

## Results

### The polarization PALM (p-PALM) method

#### Motivation for the p-PALM method

The mobility of many molecules varies widely and often during their lifetime within cells, dependent on viscosity, crowding and local interactions. Most often measured is translational mobility, and numerous super-resolution light microscopy approaches have been developed over the past decade suitable for this purpose (see [*Huang et al., 2010*; *Huang et al., 2009*] for reviews). Highly localized effects, such as those produced by multiple binding interactions or increases in crowding, often produce small changes in translational mobility (nanometer-scale step sizes in milliseconds) that are very difficult, if not impossible, to detect by these methods. However, these environmental changes can produce significant and detectable changes in *rotational* mobility. We surmised that the rotational mobility of a probe within the FG-network of NPCs would be strongly influenced by the densities of the FG-polypeptides and other macromolecules, such as NTRs, that increase crowding and decrease mobility of the FG-polypeptides, and therefore, we developed a method that could detect differences in rotational mobility.

Our basic approach was to genetically attach a photoactivatable fluorescent protein to different FG-polypeptides, and at different locations within an FG-polypeptide, in order to determine the rotational mobility of this fluorescent probe in different environments within the FG-network. Fluorescence polarization measurements are often used for probing rotational motion and are readily applied at the single molecule level (*Forkey et al., 2000*; *Forkey et al., 2005*; *Ha et al., 1999*; *Harms et al., 1999*; *Lakowicz, 2006*; *Loman et al., 2010*; *Testa et al., 2008*). However, single molecule polarization measurements within the NPC pose a special challenge: since the NPC has eight-fold rotational symmetry, any Nup genetically tagged with a fluorescent protein (or chemically tagged with a dye) will be present in numerous copies, and therefore, due to their proximity, the diffraction-limited emission from individual fluorescent tags will overlap significantly, thereby complicating analysis. Consequently, we combined single molecule polarization measurements with PALM (*Betzig et al., 2006*), in which probe molecules are stochastically and individually photoactivated. In this approach, termed polarization PALM (p-PALM), single fluorescent protein molecules were activated as in PALM, but the emission was split by a polarizer onto separate halves of an EMCCD camera (*Figure 1*), enabling polarization measurements to be made on individual molecules. As we show, there are significant advantages of this single molecule approach over ensemble fluorescence polarization methods.

Similar approaches to the p-PALM method described herein were used previously to detect changes in rotational mobility (*Gould et al., 2008*; *Testa et al., 2008*). However, a quantitative relationship between rotational diffusion constants and p-PALM measurements has not been reported. In our analysis, we explored the different rotational time regimes and now more fully describe the parameter space, which is essential to interpret the results and provides numerous additional insights into the power of the approach. Rotational random walk simulations were used to determine the effect of imaging speed, fluorescence lifetime, anisotropic rotational diffusion, dipole orientation, thresholding, noise level, and numerical aperture over ~10 orders of magnitude of the average rotational diffusion constant, $D_r$. These simulations revealed that p-PALM can detect changes

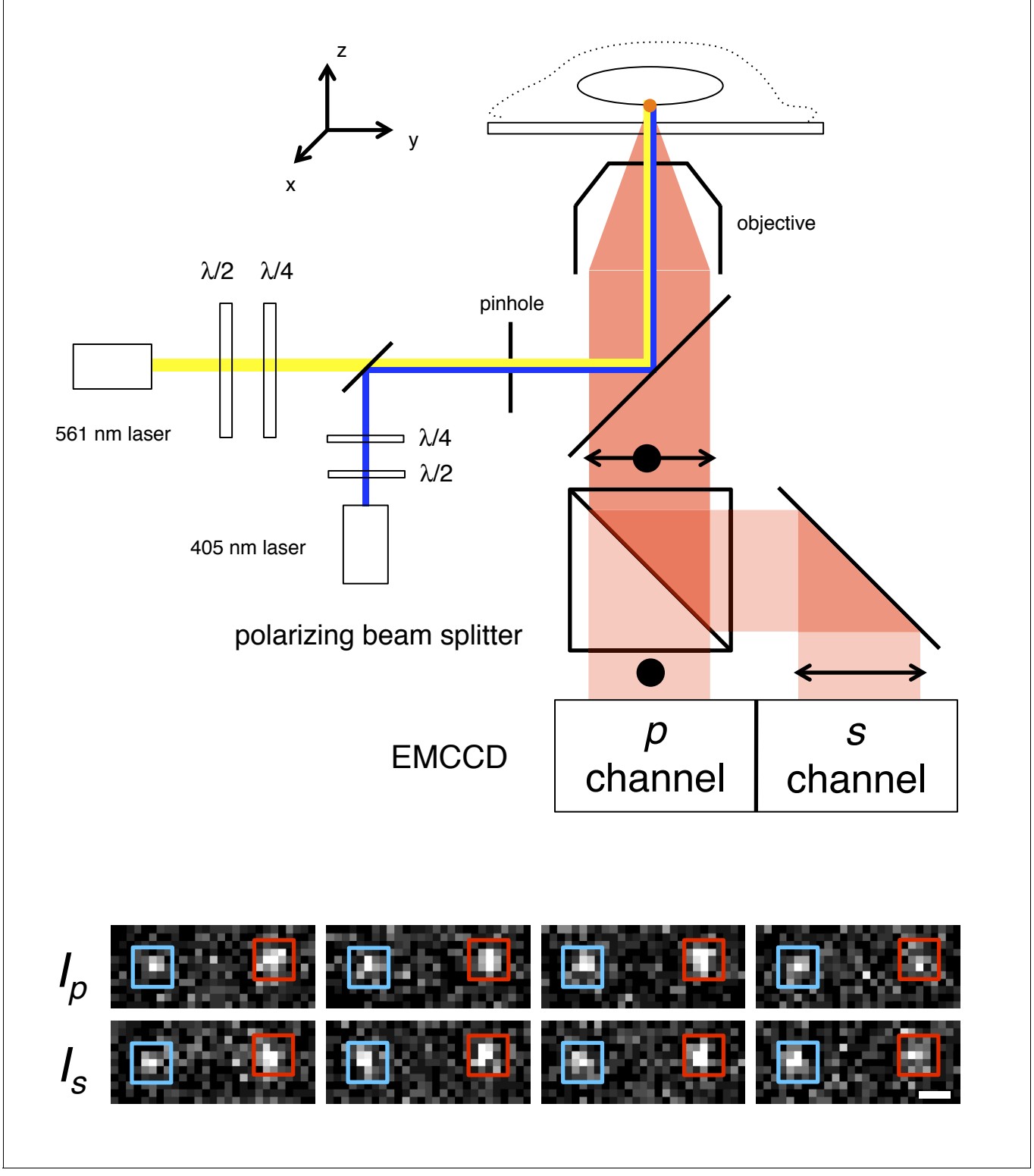

**Figure 1.** p-PALM imaging. The fluorescent protein mEos3 was photoactivated by UV illumination (405 nm), and excited by linearly or circularly polarized 561 nm light. The mEos3 fluorescence emission was separated by a 50:50 polarizing beam splitter and detected on two halves of an EMCCD camera. The images show four successive frame-pairs in which two molecules (*red* and *blue* boxes) of Pom121-mEos3 (see *Figure 1—figure supplements 1,2* for all mEos3 fusion protein constructs used in this work) were detected at the bottom of the nucleus in a permeabilized HeLa cell using circular excitation (see *Video 1*). Fluctuating emission intensities ($I_p$ and $I_s$) result from changes in the molecules' average orientation during the image integration time ($t = 10$ ms). The λ/2 and λ/4 waveplates were used to rotate the angle of linear polarization and to adjust the ellipticity,

*Figure 1 continued on next page*

*Figure 1 continued*

respectively, of the excitation beams. A 300 μm pinhole was used to reduce the illumination area to ~7 μm (narrow-field epifluorescence [*Yang et al., 2004*]). Scale bar: 1 μm.

DOI: https://doi.org/10.7554/eLife.28716.003

The following figure supplements are available for figure 1:

**Figure supplement 1.** mEos3 fusion protein constructs with Pom121, Nup153, and RanGAP.

DOI: https://doi.org/10.7554/eLife.28716.004

**Figure supplement 2.** mEos3 fusion protein constructs with Nup98.

DOI: https://doi.org/10.7554/eLife.28716.005

occurring on timescales that are largely inaccessible by other means. The details of the experimental approach and simulations are described in the **Materials and methods** section. An overview of the approach and the general results of our analysis are summarized in the following sections.

## Outline of the p-PALM method

The main principle of p-PALM experiments is that rotational mobility information is extracted not from the average bulk polarization, but from polarization measurements obtained from thousands of individual molecules that are pooled to generate polarization frequency histograms (*Figure 2*). The primary experimental readouts from these data are the average polarization, $<p>$, and the variance of the polarization distribution, $Var(p)$. In addition, the overall shape of polarization histograms and photon scatterplots can provide additional clues as to the underlying physical constraints on the probe's rotational mobility. Polarization was defined as $p = (gI_p - I_s)/(gI_p + I_s)$, where $I_p$ and $I_s$ are the fluorescence intensities measured for each single molecule spot in the two polarization channels, and $g$ corrects for the different photon collection efficiencies of these channels (*Gould et al., 2008*; *Harms et al., 1999*). We used a measurement timescale (10 ms) comparable to the timescale of protein import by the NPC (*Grünwald et al., 2011*; *Tu and Musser, 2011*). For circularly polarized excitation, the average polarization ($<p>_{cir}$) as well as the peak of the distribution are theoretically always 0 (for uniformly distributed dipole orientations), providing a convenient check on instrument alignment and calibration (see below and **Materials and methods**), but providing no information on rotational mobility. Instead, the histogram width, quantified as $Var(p)$, provides an estimate of the rotational mobility, with increasing width corresponding to decreasing rotational mobility (lower $D_r$ values). Since slowly rotating molecules emit from distinct orientations during the data collection period, a wider range of polarization values are obtained for lower $D_r$ values, whereas rapidly

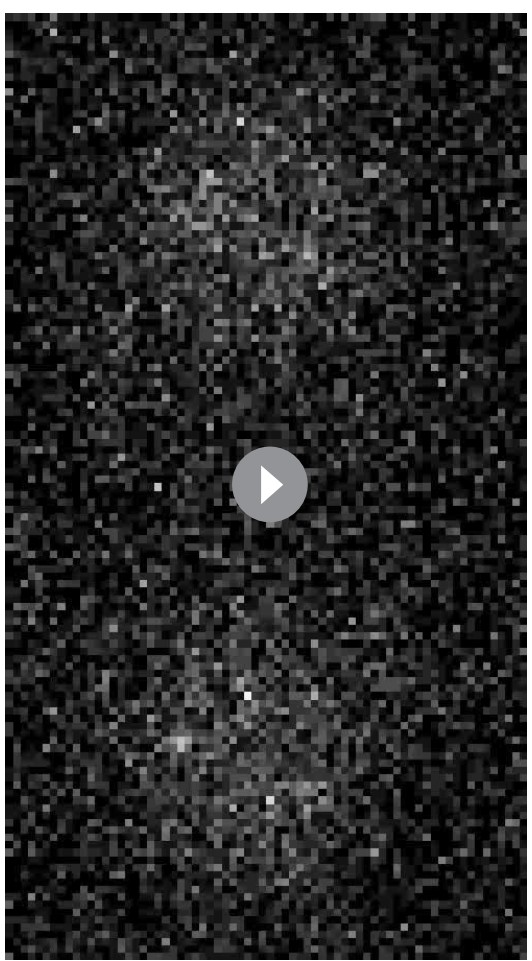

**Video 1.** p-PALM imaging of Pom121-mEos3 at the bottom of the nucleus. The top half is the *p*-channel and the bottom half is the *s*-channel. The round illumination area created by the narrow-field epifluorescence imaging is clearly detectable within the center of the fields. Fluorescent spots that appear and disappear arise from single mEos3 molecules and are clearly correlated between the two channels. $t$ = 10 ms; 240 nm square pixels (see *Figure 1*).

DOI: https://doi.org/10.7554/eLife.28716.006

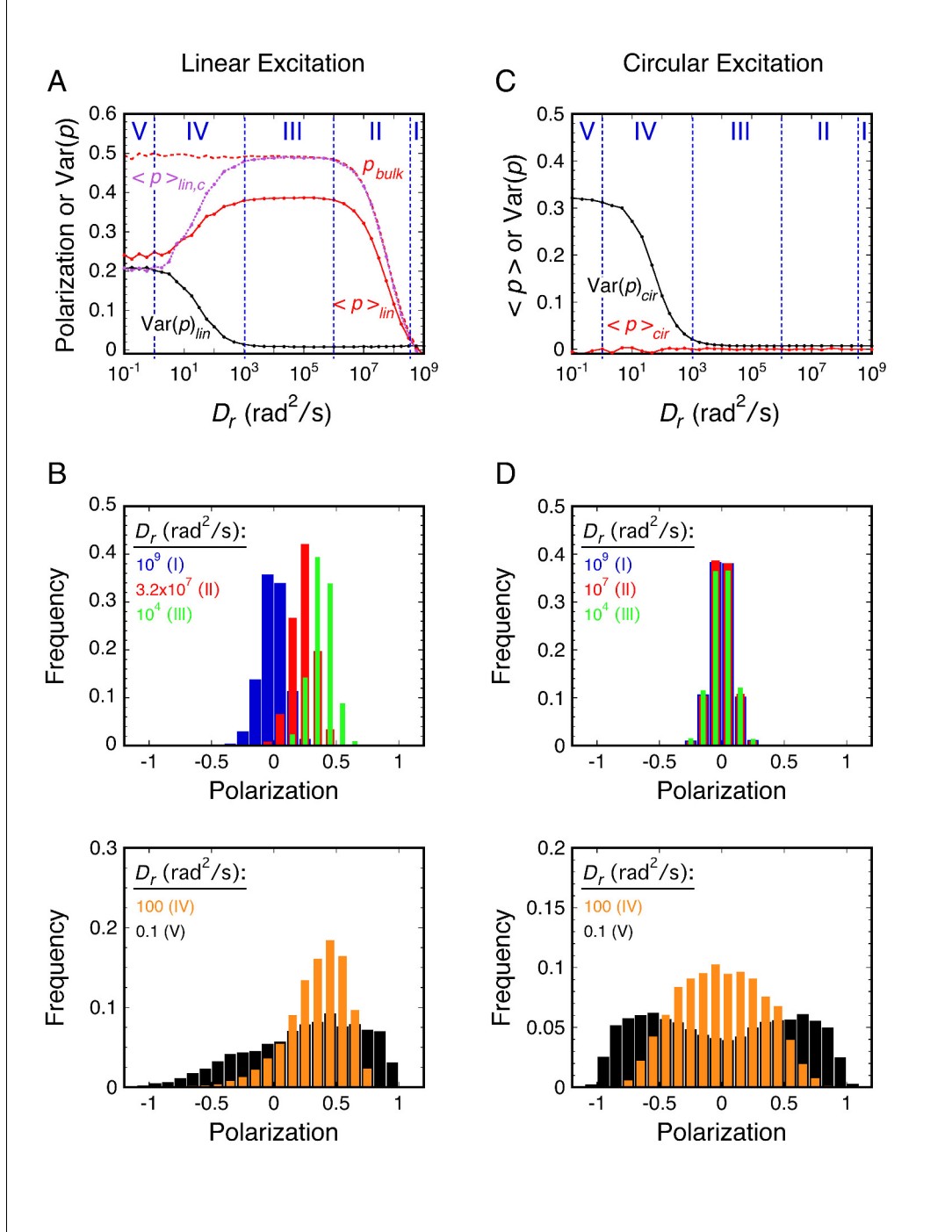

**Figure 2.** Principles of rotational mobility analysis by the p-PALM method. Rotational random walk simulations were used to obtain polarization histograms, and the corresponding mean polarization, <p>, and its variance, Var(p), for different $D_r$ values using linearly (A and B) and circularly (C and D) polarized excitation. The values predicted for single molecule measurements (*solid*) in A and C were calculated from polarization histogram data, such as that shown in B and D. The bulk polarization ($p_{bulk}$) using linear excitation (A) was calculated assuming that the photons from all molecules (N = 10,000 per $D_r$ value) were collected simultaneously in the two polarization channels. <p>$_{lin,c}$ is the mean polarization collected under p-PALM conditions corrected for the polarization mixing by the microscope objective. See text for discussion and **Materials and methods** for simulation details and definitions. In all panels: $D_r = D_x = D_y = D_z = (D_x + D_y + D_z)/3$ (i.e., a spherical particle); the noise (p photons, s photons) was (22, 15) and (15, 15) using linear and circular excitation, respectively; 1400 (circular) or 2800 (linear) rotational random walk steps per simulation, yielding an average of ~350 photons at high $D_r$ values (see **Figure 2—figure supplement 5**); 10,000 initial values per simulation; t = 10 ms; τ (fluorescence lifetime) = 3.5 ns; $θ_{obj}$ = 74.1°. The five rotational diffusion regimes (identified in A and C) are described in **Figure 2—figure supplement 1**. **Figure 2—figure supplements 1–5** illustrate the influence of fluorescence lifetime, integration time, threshold, intensity shape factor, noise, ellipticity, and the number of

*Figure 2 continued on next page*

*Figure 2 continued*

collected photons on $<p>$ and Var($p$). ***Figure 2—figure supplement 6*** illustrates the relationship between the number of measurements and the statistical measurement uncertainty. The anisotropy values, $<r>_{bulk}$ and $<r>_{lin,c}$, are shown in ***Figure 2—figure supplement 7*** and described in **Materials and methods**.

DOI: https://doi.org/10.7554/eLife.28716.007

The following figure supplements are available for figure 2:

**Figure supplement 1.** Principles of rotational mobility analysis by the p-PALM method – no threshold, intensity broadening, or background noise.
DOI: https://doi.org/10.7554/eLife.28716.008
**Figure supplement 2.** Effect of threshold, intensity shape factor, and background noise on $<p>$ and Var($p$).
DOI: https://doi.org/10.7554/eLife.28716.009
**Figure supplement 3.** Effects of image integration time and fluorescence lifetime on $<p>_{lin}$ and Var($p$)$_{cir}$.
DOI: https://doi.org/10.7554/eLife.28716.010
**Figure supplement 4.** Effect of excitation ellipticity on $<p>$ and Var($p$).
DOI: https://doi.org/10.7554/eLife.28716.011
**Figure supplement 5.** Influence of the total photons collected on $<p>$ and Var($p$).
DOI: https://doi.org/10.7554/eLife.28716.012
**Figure supplement 6.** Estimating the error in datasets used to calculate Var($p$)$_{cir}$ – the number of data points required for reasonable estimates.
DOI: https://doi.org/10.7554/eLife.28716.013
**Figure supplement 7.** Comparison of bulk anisotropy with the corrected anisotropy calculated from p-PALM experiments.
DOI: https://doi.org/10.7554/eLife.28716.014

rotating probes yield time-averaged polarizations near zero. For linearly polarized excitation, $<p>_{lin}$ is almost always non-zero and was the parameter used for inferring rotational mobility in this excitation mode. The effects of the $D_r$ on histograms of polarization measurements, $<p>$, and Var($p$) for a spherical particle are shown in ***Figure 2***.

The monomeric fluorescent protein mEos3.1 (hereafter, simply denoted mEos3) (***Zhang et al., 2012***) was used as our probe molecule. Photoactivation of mEos3 by UV (405 nm) light results in conversion (photoactivation) from a 'green' (~500–550 nm emission) to an 'orange' (~570–650 nm emission) fluorescent state, a process that is irreversible due to polypeptide cleavage (***McKinney et al., 2009***; ***Wiedenmann et al., 2004***). The mEos3 proteins were successively and individually photoactivated by continuous low level UV irradiation, and deactivated by photobleaching. PALM images and p-PALM polarization histograms were generated from thousands of position or polarization measurements, respectively, from many tens to hundreds of NPCs. For mEos3 tagged FG-polypeptides, the average number of probes detected per NPC was typically ~3–5. This is consistent with the known photoactivation efficiency of ~50% for photoactivatable proteins (***Durisic et al., 2014***). Narrow-field epifluorescence imaging (***Yang and Musser, 2006b***) was used for single molecule detection. In narrow-field epifluorescence, a small diameter excitation beam is confined to the center of the objective such that only a small area within the sample plane is illuminated (in our case, a 300 μm pinhole yielded an ~7 μm diameter illumination area; see ***Figure 1***). This approach largely eliminated depolarization effects that normally result from focusing an excitation beam toward the *z*-axis (optical axis) by a high numerical aperture (NA = 1.46) objective (***Ha et al., 1999***; ***Olivini et al., 2001***). The p-PALM approach does not require the high spatial precision typically obtained from super-resolution methods, which minimizes the need for high precision image alignment. Rather, polarization measurements are based on *intensities*, and spatial localizations are only necessary for spot correlation between the *p*- and *s*-channels. Thus, relatively low photon counts are acceptable. The excitation intensity was adjusted for all of our experiments so that the average total emission intensity was typically ~300–400 photons ($N_{photons}$). While the number of photons collected from an individual molecule in the two imaging channels depends on the 3D orientation trajectory of the probe dipole during image acquisition, we emphasize that our method does not require knowledge of the individual 3D rotational trajectories of each probe. Instead, the average rotational mobility is inferred from the statistical properties of experimental polarization frequency histograms by comparison with the theoretically predicted values calculated from simulated rotational random walk trajectories (***Figure 2*** and **Materials and methods**).

## Five rotational mobility regimes

Generally, for a given single molecule, the photons collected in a single image correspond to hundreds of excitation/emission cycles, during and between each of which the probe might rotate. Throughout this paper, we have assumed that the excitation and emission dipoles of the fluorescent particle are parallel (hereafter simply referred to as the transition dipole), which is the case for GFP and many fluorophores (*Ha et al., 1999*; *Inoué et al., 2002*). Our theoretical model of rotational diffusion (based on rotational random walk simulations – see **Materials and methods**) revealed five important rotational diffusion regimes in p-PALM experiments (*Figure 2*; more fully described and illustrated in *Figure 2—figure supplement 1*). These rotational diffusion regimes are defined by two time parameters, the fluorescence lifetime of the fluorophore ($\tau_F$) and the image integration time ($t$) (see *Figure 2—figure supplement 2*). The $\tau_F$ determines the time for the molecule to rotate between excitation and fluorescence emission. In combination with the number of photons collected, $t$ determines the average time allotted for the molecule to rotate between potential excitation events ($\tau$). In rotational random walk simulations, $\tau$ was calculated from $t$ and the number of rotational walk steps ($N_S$) as $\tau = t/N_S$, and was typically ~3-7 µs. Photon collection after each rotational walk step was dependent on excitation and emission probabilities of the dipole and the collection efficiency of the microscope channels. $N_S$ was set to approximately yield the experimentally collected number of photons (see **Materials and methods**). In general, the relationship between $D_r$ and $<p>$ or Var($p$) cannot be obtained analytically. However, our simulations of rotational diffusion were verified with an analytical solution for a special case (see **Materials and methods** and **Appendix 2**).

Considering the five rotational diffusion regimes (*Figure 2*), Regimes I and II correspond to the rotational mobility probed in most bulk fluorescence anisotropy experiments, where the increase in anisotropy at lower $D_r$ values is described by the Perrin equation (*Figure 2—figure supplement 3*) (*Lakowicz, 2006*). However, as demonstrated in the following sections, experimental polarization measurements from probes within the crowded FG-network of NPCs correspond to the slower rotational mobilities in Regimes III and IV. Linear excitation is primarily useful for molecules with high rotational mobility (Regimes I and II), where $<p>_{lin}$ varies from 0 to ~0.4 (*Figure 2*). In single molecule experiments at lower $D_r$ values (Regimes IV and V), the fluctuations in $<p>_{lin}$ strongly depend on thresholding and noise levels (*Figure 2—figure supplement 4*). Circular excitation is preferred over linear excitation for Regime IV (*Figure 2*) due to the larger dynamic range in Var($p$)$_{cir}$ and because this parameter is less sensitive to acquisition parameters than $<p>_{lin}$ (*Figure 2—figure supplement 4*). Consequently, we primarily used circular excitation for the results that follow.

## Differences between bulk and single molecule polarization measurements

Importantly, the single molecule average polarization ($<p>_{lin}$) as defined in this paper and the average bulk fluorescence polarization ($p_{bulk}$) obtained using linear excitation differ significantly, as illustrated in *Figure 2A*. There are two reasons for this. First, the intensities collected by a microscope objective are mixtures of parallel and perpendicular intensities. Correcting for these mixed intensities (see **Materials and methods**) yields $<p>_{lin,c}$, which agrees with $p_{bulk}$, except at low $D_r$ values (*Figure 2A*). And second, the remaining difference between $<p>_{lin,c}$ and $p_{bulk}$ that occurs at low rotational mobilities results from different weightings for each molecule's fluorescence emission. That is, $<p>_{lin,c}$ is calculated by weighting each molecule identically (i.e., the polarization of each molecule counts the same no matter how many photons are emitted), whereas $p_{bulk}$ weights the contribution of each molecule to the measured polarization depending on the number of photons emitted. Consequently, since most polarizations are recovered in p-PALM experiments under low rotational mobility conditions, $<p>_{lin}$ (and $<p>_{lin,c}$) tends toward 0 at low $D_r$ values, although this decrease is limited by the sensitivity threshold. In contrast, dipoles oriented at large angles relative to the excitation polarization emit few photons, and therefore provide only a small contribution to $p_{bulk}$. A mathematical description of the differences between p-PALM and bulk polarization measurements is given in **Materials and methods**. Notably, Var($p$) cannot be obtained from bulk measurements because it requires polarization measurements from individual molecules, and hence, the quantification of slow rotational mobility made possible with p-PALM experiments cannot be obtained in a corresponding bulk experiment. Whereas anisotropy is favored as the measurement parameter for bulk fluorescence measurements, which typically have a parallel and two

perpendicular channels (*Lakowicz, 2006*), polarization is a natural parameter for p-PALM measurements since there are only the two detection channels (*p* and *s*), neither of which can be used to directly account for the photons that escape detection.

### Effect of anisotropic rotation of the probe on <*p*> and Var(*p*)

The approach taken in many of the experiments reported herein was to explore various local environments of the FG-network by covalently attaching an mEos3 probe to different FG-polypeptides and measuring its rotational mobility by p-PALM. Under such conditions, the principal rotational diffusion constants of the probe could be differentially affected relative to a freely diffusing spherical particle due to constraints generated by the local environment and/or by the FG-polypeptide to which it was tethered. We therefore used rotational random walk simulations to determine the effect of varying the relationship between the three principal rotational diffusion constants $D_x$, $D_y$, and $D_z$. Somewhat surprisingly, our results and simulations suggested that the probe's rotation mobility behavior was at most only slightly anisotropic or the angle between the dominant rotational axis and the transition dipole was near the magic angle of 54.7° (*Axelrod, 1989*). Consequently, the probe's behavior largely resembled that of an untethered spherical particle. A more detailed discussion of the effects of rotational anisotropy on p-PALM measurements is given in **Appendix 1**.

### Effect of numerical aperture (NA) on <*p*> and Var(*p*)

The NA is an important parameter in p-PALM experiments because it directly influences the experimentally measured values of <*p*> and Var(*p*) (*Equations 7-11*). Knowing the NA, <*p*> can in principle be standardized by converting the experimental value into the corresponding bulk parameter (*Equation 21*). However, there is no corresponding bulk value for Var(*p*). More importantly, the effective NA (NA$_{eff}$) under the acquisition conditions can be significantly different than the nominal NA of the objective. For example, the spherical aberration that results from the refractive index mismatch when using an oil immersion lens for an aqueous sample yields a reduced NA, which is particularly significant when probing the sample far from the coverslip surface. A more detailed discussion of the effect of NA on p-PALM measurements is given in **Appendix 1**. Importantly, our major conclusions are not affected by a moderate uncertainty in the NA$_{eff}$.

## p-PALM measurements of the FG-network of NPCs

### Detecting rotational mobility changes in the FG-network of NPCs via p-PALM

For initial proof-of-concept experiments, mEos3 was fused to the C-terminus of Pom121 (Pom121-mEos3), and a stable HeLa cell line was generated. Pom121 is generally agreed to be a central membrane-integrated Nup (*Antonin et al., 2005*; *Hallberg et al., 1993*; *Söderqvist and Hallberg, 1994*; *Söderqvist et al., 1997*; *Talamas and Hetzer, 2011*; *Yang and Musser, 2006a*) with its N-terminal domain anchored to the NPC scaffold and its C-terminal FG-domain within the FG-network. We used both linearly and circularly polarized excitation beams (ellipticity >100 and <1.1, respectively), and p-PALM data were obtained from permeabilized HeLa cells by focusing on the bottom of cell nuclei. Thus, the sample plane coincided with the plane of the nuclear envelope (NE).

We first examined the as-isolated (wildtype) NPCs in permeabilized cells, and then determined whether addition of the NTR Importin β1 (Imp β1) or the transport inhibitor wheat germ agglutinin (WGA) (*Adam et al., 1990*; *Dabauvalle et al., 1988*; *Finlay and Forbes, 1990*; *Wolff et al., 1988*; *Yoneda et al., 1987*) influenced the rotational mobility of the mEos3 probe (*Figure 3*). Polarization frequency histograms using the collection/integration time *t* = 10 ms (per image) revealed that most measurements for Pom121-mEos3 were close to *p* = 0.4 for linearly polarized excitation and near *p* = 0 for circularly polarized excitation (*Figure 3A and D*). In the presence of Imp β1 (10 μM), the polarization histogram obtained with linear polarization was virtually unchanged and that obtained with circular polarization appeared slightly broader (*Figure 3B and E*). In the presence of WGA (1 mg/mL; ~26 μM), the difference between linear and circular excitation was clearly observed, with circular excitation producing a substantially wider polarization histogram (*Figure 3C and F*). WGA binds to O-GlcNAc-modified Nups, of which there are at least five in humans (*Finlay and Forbes, 1990*; *Hülsmann et al., 2012*). Since the WGA dimer has eight GlcNAc-binding sites (*Schwefel et al., 2010*), WGA molecules can potentially non-covalently 'crosslink' the FG-network,

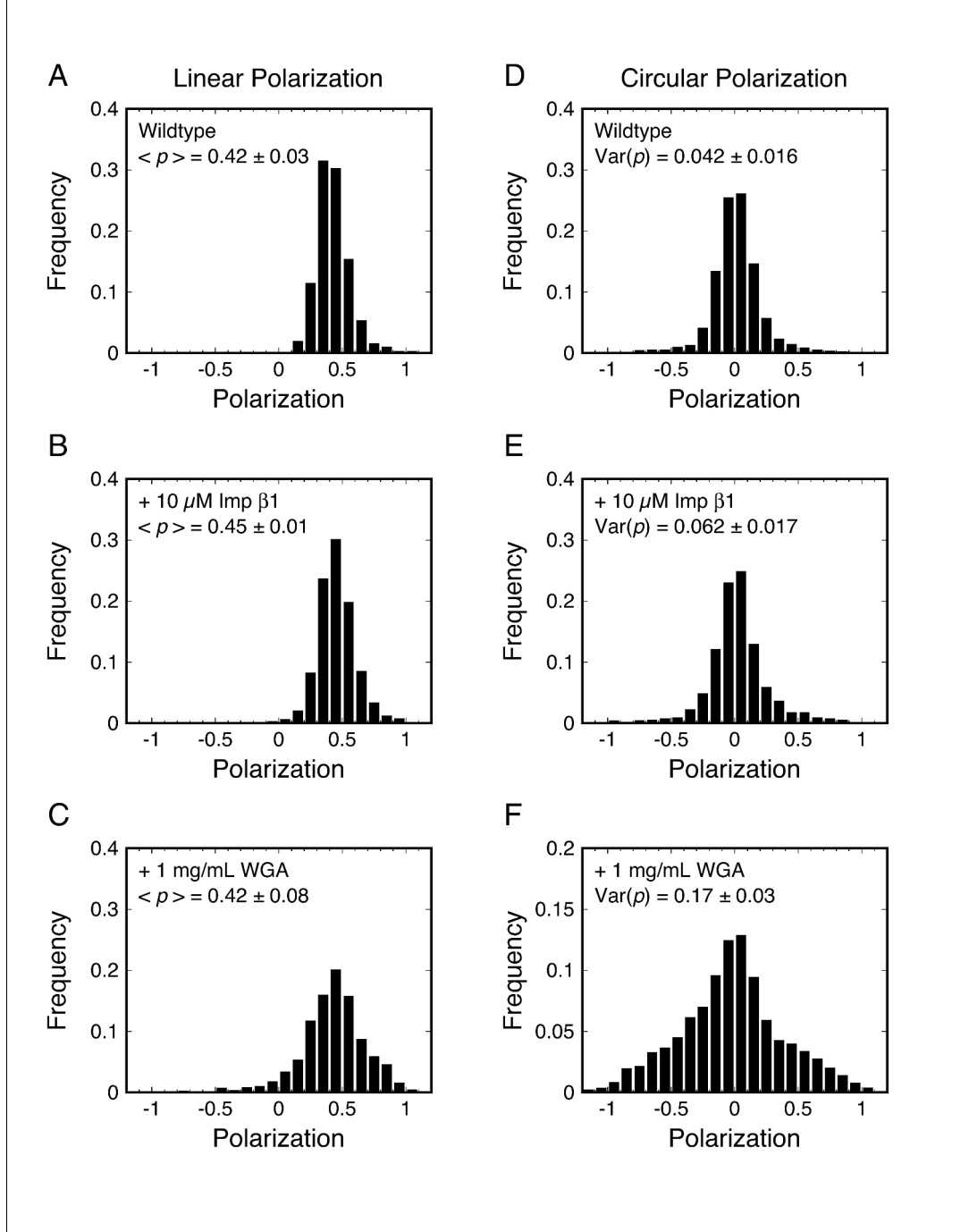

**Figure 3.** p-PALM polarization histograms for Pom121-mEos3. Linearly (**A–C**) and circularly (**D–F**) polarized excitation was used to obtain experimental polarization histograms for mEos3 attached to the C-terminus of Pom121 (Pom121-mEos3) under the indicated conditions. $t$ = 10 ms. See text for discussion.

DOI: https://doi.org/10.7554/eLife.28716.015

The following figure supplement is available for figure 3:

**Figure supplement 1.** p-PALM polarization histograms for mEos3 in 92% glycerol.
DOI: https://doi.org/10.7554/eLife.28716.016

and thus severely inhibit the rotational mobility of proteins embedded within it. The data in *Figure 3* support this hypothesis (compare with *Figure 2*). Overall, these data demonstrate that the p-PALM approach can detect changes in rotational mobility within the FG-network. A reduced rotational mobility likely implies an increased number of contacts with surrounding macromolecules (which could be strongly interconnected), and hence, we interpret a reduced rotational mobility as primarily a result of increased molecular crowding.

## Membrane topology of Pom121

Pom121 is one of three membrane proteins anchoring the NPC to the NE (*Eriksson et al., 2004*; *Stavru et al., 2006*), yet its membrane topology remains unresolved. We considered that the number of transmembrane domains (1 or 2) might be resolved by comparing the effect of WGA on the rotational mobility of the mEos3 probe attached to the N- or C-terminus of Pom121. Most studies describe Pom121 as having a single transmembrane domain (TMD) near its N-terminus, based primarily on a single long sequence of hydrophobic residues (*Antonin et al., 2005*; *Funakoshi et al., 2011*; *Hallberg et al., 1993*; *Söderqvist and Hallberg, 1994*; *Söderqvist et al., 1997*; *Talamas and Hetzer, 2011*). This topology would place the N-terminus of Pom121 in the perinuclear space (between the inner and outer NE membranes), and thus, inaccessible to exogenous reagents such as WGA. However, the hydrophobic transmembrane region of Pom121 is ~44 residues, and thus, it is long enough to span the membrane twice – a possibility that was considered in the initial report identifying this protein (*Hallberg et al., 1993*). In this alternate scenario, the N-terminus of Pom121 would be accessible from (or embedded within) the FG-network (*Figure 4A*), and the rotational mobility of an N-terminally attached probe could potentially be influenced by WGA.

To probe the membrane topology of Pom121, we attached mEos3 to the N-terminus of Pom121 (mEos3-Pom121), and obtained p-PALM measurements under wild-type, +NTR, and +WGA conditions after permeabilization of a stable cell line. WGA had a strong effect on the rotational mobility of the mEos3 probe (*Figure 4B*). These data support the hypothesis that Pom121 has two TMDs, and that the N-terminus of Pom121 is embedded within the FG-network. Alternatively, WGA could have long-range effects that are mediated across the NE membrane, thus reducing rotational mobility of mEos3 within the ER lumen, but we consider this scenario unlikely. These data demonstrate that the p-PALM approach can address structural questions difficult to ascertain with other methods by using the effects of increased crowding to report on the local environment of (or accessibility to) a probe.

## Rotational mobility at different locations within the FG-network

Having established a framework for interpreting p-PALM measurements and demonstrating its utility for addressing structural questions, we then explored the rotational mobility of mEos3 tethered at various locations within the FG-network. We expected that different FG-polypeptide densities and different local concentrations of embedded NTRs and WGA would be reflected in differential effects on rotational mobility, and hence report on different levels of molecular crowding. In addition to the centrally located Pom121, we examined Nup98, a crucial element of the permeability barrier (*Hülsmann et al., 2012*). To probe the nuclear basket, we used Nup153 (*Fahrenkrog et al., 2002*; *Hase and Cordes, 2003*), which plays roles in both protein import and mRNA export and regulates the permeability barrier (*Lowe et al., 2015*; *Makise et al., 2012*; *Shah and Forbes, 1998*; *Ullman et al., 1999*). To probe the cytoplasmic filament region, we used RanGAP, which is essential for promoting GTP hydrolysis resulting in export complex disassembly (*Bischoff and Görlich, 1997*; *Bischoff et al., 1994*; *Güttler and Görlich, 2011*; *Kutay et al., 1997a*; *Okamura et al., 2015*) and which binds in its SUMOylated form to Nup358 (*Reverter and Lima, 2005*), a major component of the cytoplasmic filaments (*Walther et al., 2002*; *Wu et al., 1995*).

p-PALM measurements indicated that WGA had a strong effect on rotational mobility when the mEos3 probe was fused to all four proteins. In contrast, NTRs had a substantially weaker or no effect on probe rotational mobility (*Figure 4B*). Since both NTRs and WGA were added at ~1 mg/mL (~10 μM and 26 μM [dimer], respectively), these data indicate that exogenous NTRs introduce less molecular crowding in the FG-network structure than WGA.

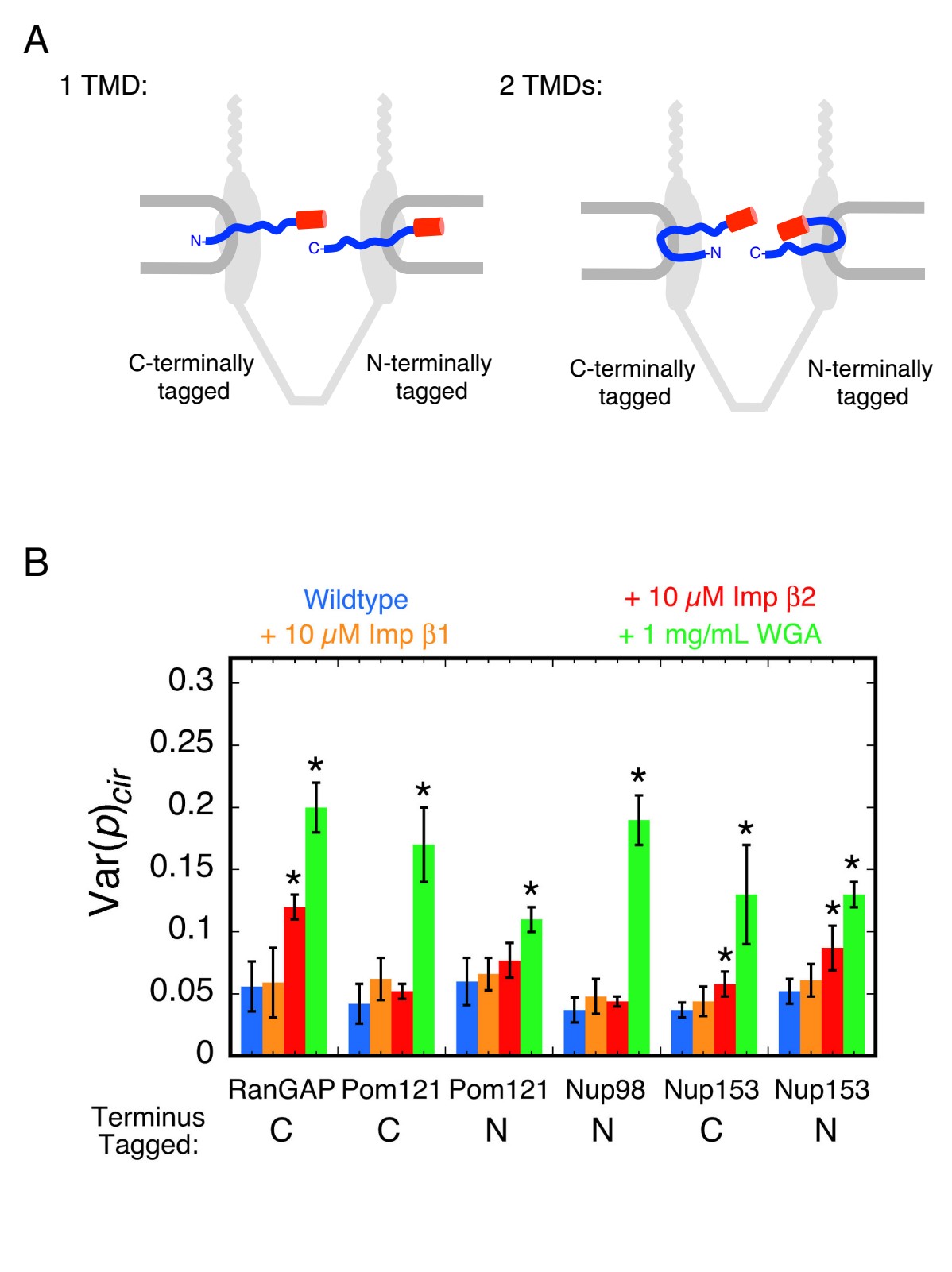

**Figure 4.** Pom121 membrane topology and rotational mobility of the mEos3 probe within the FG-network under various conditions. (**A**) Possible membrane topologies of Pom121. The N-terminus of Pom121 is predicted to be in the lumen of the ER (1 TMD) or in the central pore (2 TMDs). TMD = transmembrane domain. (**B**) $Var(p)_{cir}$ values for various proteins under the indicated conditions. p-PALM data were obtained under the same conditions as in *Figure 3*. The stars (*) indicate significantly different values from the wildtype (*blue*) condition within the same group according to a

*Figure 4 continued*

two-sided Welch's *t*-test (95%). Note that the actual significance level is expected to be higher than indicated since the error bars shown here are wider than a typical standard deviation (see *Figure 2—figure supplement 6*). A significant effect of WGA on the probe attached to the N-terminus of Pom121 suggests that this part of Pom121 is located within the central pore, and not in the ER lumen, and thus, that Pom121 has two TMDs (see (**A**)). 10 μM NTR ≈ 1 mg/mL.

DOI: https://doi.org/10.7554/eLife.28716.017

## Rotational mobility at different positions within the Nup98 disordered domain

Different segments of an FG-polypeptide are expected to sample different regions of the FG-network depending on its anchoring point and the polypeptide length from the anchor point. To probe the molecular crowding in the neighborhood of different segments of an FG-polypeptide, we performed p-PALM measurements on mEos3 probes attached at different positions within the disordered region of Nup98. We chose Nup98 because of its important role in forming the permeability barrier (*Hülsmann et al., 2012*). The domain structure of Nup98 is shown in *Figure 5A*. With the exception of a 56-residue GLEBS domain, the first ~700 residues of this 920 residue protein are largely disordered, and most of this region contains FG repeats. The C-terminal autoproteolytic (APD) domain of Nup98 binds to Nup88 and Nup96, and is therefore thought to anchor Nup98 onto the NPC scaffold (*Griffis et al., 2003*; *Hodel et al., 2002*; *Pleiner et al., 2016*; *Stuwe et al., 2012*).

In the first set of experiments, 109, 399, 499, or 699 residues were deleted from the N-terminus, and the mEos3 probe was attached to the new N-terminus (*Figure 1—figure supplement 1*). These 'tip' labeled, truncated versions of Nup98 were then examined in p-PALM experiments after transient transfections (*Figure 5B*). To our surprise, mEos3-$^{500\text{tip}}$Nup98 and mEos3-$^{700\text{tip}}$Nup98 were not retained at the NE in permeabilized cells (*Figure 5—figure supplement 1*), indicating that the Nup98 C-terminal domain was insufficient to anchor these proteins to the NPC scaffold (it is unclear if any binding to the NE occurs in live cells; see *Figure 5—figure supplement 1*). In separate experiments, we found that residues 1–500 of Nup98 do not bind to the NPC in either permeabilized or live cells (*Figure 5—figure supplement 1*), indicating that both the FG and C-terminal domains of Nup98 appear to be required for localization to the NPC.

In a second set of experiments, we added the deleted portion back to the Nup98 'tip' mutants, thus placing the mEos3 probe in the middle of two sections of Nup98. These 'middle' labeled Nup98 mutants (*Figure 1—figure supplement 1*) were all localized to NPCs. In as-isolated permeabilized cells, rotational mobility was reduced when the mEos3 probe was attached near the Nup98 C-terminal folded domain. These data suggest more crowded conditions near the NPC scaffold than at the tip of Nup98 (which is likely found most of the time toward the center of the pore or on the nucleoplasmic and cytoplasmic sides, i.e., near the 'surface' of the FG-network). Imp β1 did not further affect rotational mobility relative to the as-isolated cells, yet WGA had a significant effect on the probe at all positions (*Figure 5B*). Notably, WGA had the strongest effect on the rotational mobility of the probe on mEos3-$^{400\text{mid}}$Nup98, indicating that larger WGA-induced effects occurred near the middle of the Nup98 disordered region rather than near its FG-polypeptide tip (N-terminus) or the anchor domain (APD domain). These data indicate that different segments of an FG-polypeptide domain experience different environments within the FG-network.

## Mixed populations

In all our p-PALM experiments, the mEos3 probe was attached to a single location on the protein of interest. Nonetheless, different probe molecules could potentially be in different environments of the FG-network due to the length and flexibility of the FG-polypeptide to which they were attached and/or variable environments within the same or different NPCs. The presence of a mixed population is not obviously apparent in values of <*p*> and Var(*p*), since these are obtained by averaging over the population. However, polarization frequency histograms and scatterplots of the photons recovered in the *p*- and *s*-channels (photon scatterplots) both can provide evidence for populations of probes with distinct rotational mobilities. In both cases, the distributions obtained from simulations of molecules with different rotational mobilities can be combined and compared with the experimental data to test/verify the mixed population hypothesis.

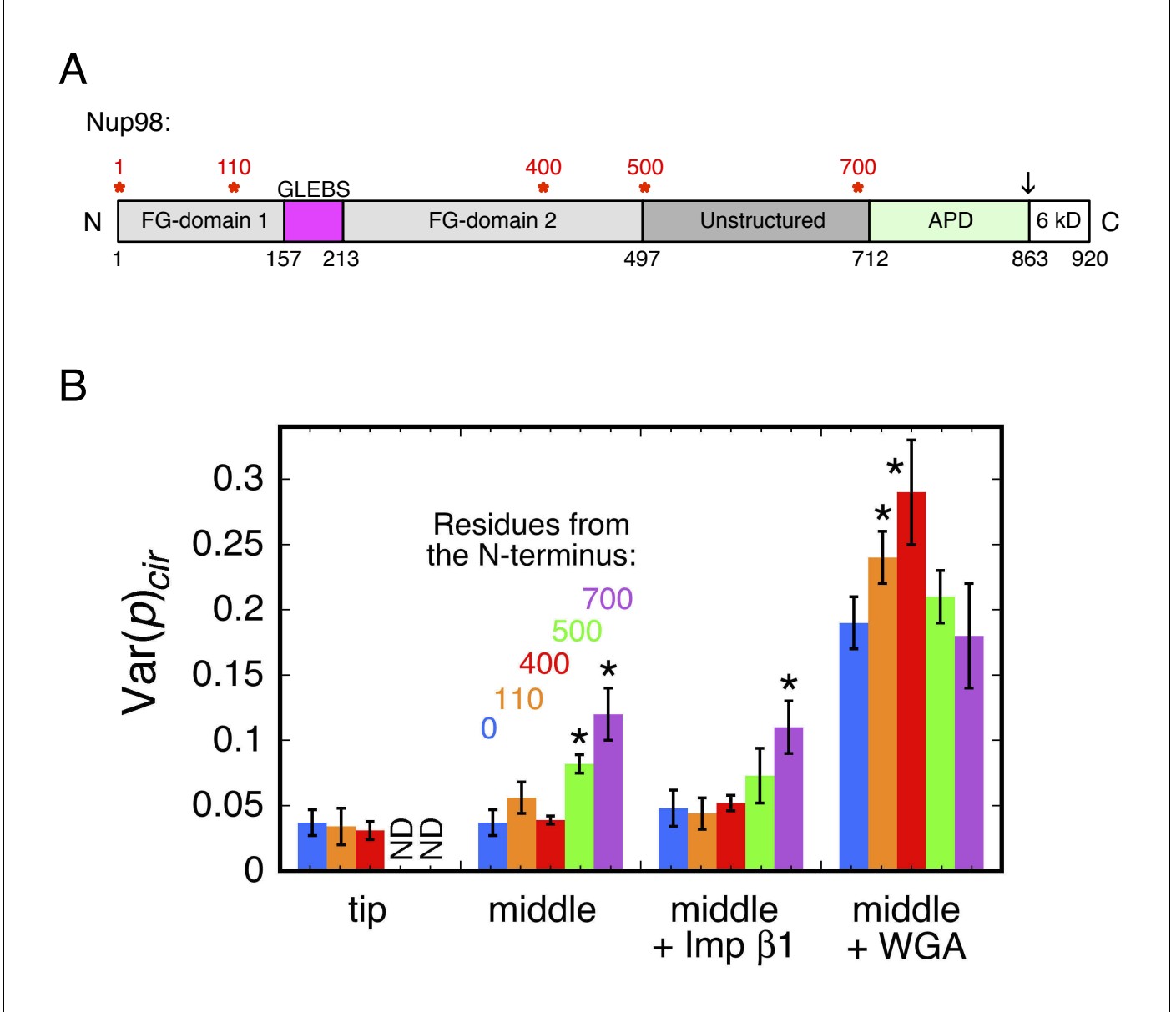

**Figure 5.** Rotational mobility of mEos3 at different positions within the Nup98 disordered domains. (**A**) Human Nup98 domain structure. The GLEBS domain is a binding motif for RNA export factors. The autoproteolytic domain (APD) co-translationally cleaves the C-terminal 6 kDa domain after residue 863 (arrow) (**Rosenblum and Blobel, 1999**). Amino acids numbers are indicated on the bottom and stars (*) indicate the positions at which the mEos3 probe was incorporated (0, 110, 400, 500, 700 residues from the N-terminus). The regions from residues 1–157 and 213–712 are considered disordered. Adapted from (**Ren et al., 2010**). (**B**) Var($p$)$_{cir}$ values obtained under the indicated conditions for the mEos3 probe at different positions within the Nup98 disordered domains (see **A**). p-PALM data were obtained as in **Figure 3**. The stars (*) indicate significantly different values from the wild-type (*blue*) condition within the same group according to a two-sided Welch's *t*-test (95%). Note that the actual significance level is expected to be higher than indicated since the error bars are wider than a typical standard deviation (see **Figure 2—figure supplement 6**).

DOI: https://doi.org/10.7554/eLife.28716.018

The following figure supplement is available for figure 5:

**Figure supplement 1.** NE binding of Nup98 tip mutants.

DOI: https://doi.org/10.7554/eLife.28716.019

There were multiple indications that the mEos3 probes in some of our p-PALM experiments sampled multiple environments distinguishable by different rotational mobilities. One example is shown in *Figure 6*. The broad wings of the polarization histogram in *Figure 3D* were not expected (see *Figure 2D*). Moreover, a photon scatterplot for Pom121-mEos3 reveals a substantially wider distribution of points (*Figure 6A*) than simulated scatterplots that assume a homogeneous population (*Figure 6B and C*). In contrast, a model that assumes a mixture of two sub-populations of molecules having distinct $D_r$ values yields good agreement with the experimental data (*Figure 6D–F*). This example demonstrates that evidence for a mixed population can be obtained from both the polarization histogram and the photon scatterplot of a given dataset. Therefore, for more accurate interpretation, the underlying data should be more carefully examined rather than relying on the summary values $<p>$ and $Var(p)$. **Appendix 1** contains further discussion and additional evidence for mixed populations within our p-PALM datasets.

## PALM of the FG-network of NPCs

### Spatial distribution of mEos3 probes within the FG-network

To assist with the interpretation of the rotational mobility data and verify that the probe-labeled proteins were properly incorporated into NPCs, the locations of the mEos3 probe within the FG-network were examined via PALM. NEs were examined at the nuclear equator, allowing us to obtain localization information vis-à-vis the transport axis (*Figure 7A*). More than 2100 mEos3 localizations were used to generate a 2D density distribution map for each of the constructs used in *Figure 4B* (*Figure 7B–G*). Although previous reports have observed the eight-fold rotational symmetry of NPCs using super-resolution approaches (*Löschberger et al., 2014*; *Löschberger et al., 2012*; *Ori et al., 2013*; *Pleiner et al., 2016*; *Szymborska et al., 2013*), we found it difficult and unreliable to obtain radial distribution maps (when the optical axis was aligned with the transport axis) due to the low number of mEos3 probes detected per NPC (average of ~3–5).

We caution that the lateral dimension was artificially compressed in these PALM density distributions. These maps were generated by determining the position of the NE, and aligning clusters of spots based on their centroids. This procedure positioned the centroids of all clusters on the central axis of the NPC. This approach was necessary since there were at most a few tens of spots per NPC, and we had no independent marker for the NPC scaffold. Thus, while the axial dimension was calibrated based on the NE position, the lateral dimension was artificially squeezed. Consequently, probes on the periphery of the pore may appear to be centrally distributed in these PALM density maps. In particular, while the probes on the N-terminus of Pom121 appear to be in the center of the pore (*Figure 7C*), we consider this to be a consequence of the alignment procedure.

The observed locations vis-à-vis the transport axis for the mEos3 probe attached to the FG-polypeptides generally agreed with previous published results. The mEos3 probe on the N-terminus of Pom121 yielded a distribution pattern along the transport axis that peaked within the central pore (*Figure 7C*), consistent with a short polypeptide segment anchoring it to the NPC scaffold. In contrast, the mEos3 probe on the C-terminus of Pom121 was more widely distributed (*Figure 7D*), consistent with access to a large region of the FG-network due to the length of the FG-polypeptide (*Hallberg et al., 1993*; *Söderqvist and Hallberg, 1994*). The mEos3 probe on the N-terminus of Nup98 was widely distributed (*Figure 7E*), consistent with previous studies that found Nup98 within the central pore and on both the nuclear and cytoplasmic sides of the NPC (*Chatel et al., 2012*; *Frosst et al., 2002*; *Griffis et al., 2003*; *Krull et al., 2004*; *Radu et al., 1995*). The N-terminus of Nup153 was predominantly localized to the nuclear basket, whereas its C-terminal end was predominantly localized closer to the central pore (*Figure 7F and G*). These results are consistent with previous antibody domain mapping studies on Nup153 (*Chatel et al., 2012*; *Fahrenkrog et al., 2002*; *Krull et al., 2004*; *Lim et al., 2007*).

In contrast to the results with the mEos3 probe on FG-polypeptides, which agreed with previous reports, the PALM map for RanGAP was a bit unexpected. RanGAP was predominantly localized in the cytoplasmic filament region, consistent with it being bound to Nup358 in its SUMOylated form (*Hutten et al., 2008*; *Mahajan et al., 1997*; *Matunis et al., 1998*; *Reverter and Lima, 2005*; *Wälde et al., 2012*). However, there were a surprising number of localizations within the central pore and nuclear basket regions (*Figure 7B*), suggesting that the cytoplasmic filaments penetrate into the central pore, and/or that RanGAP can bind to other parts of the FG-network.

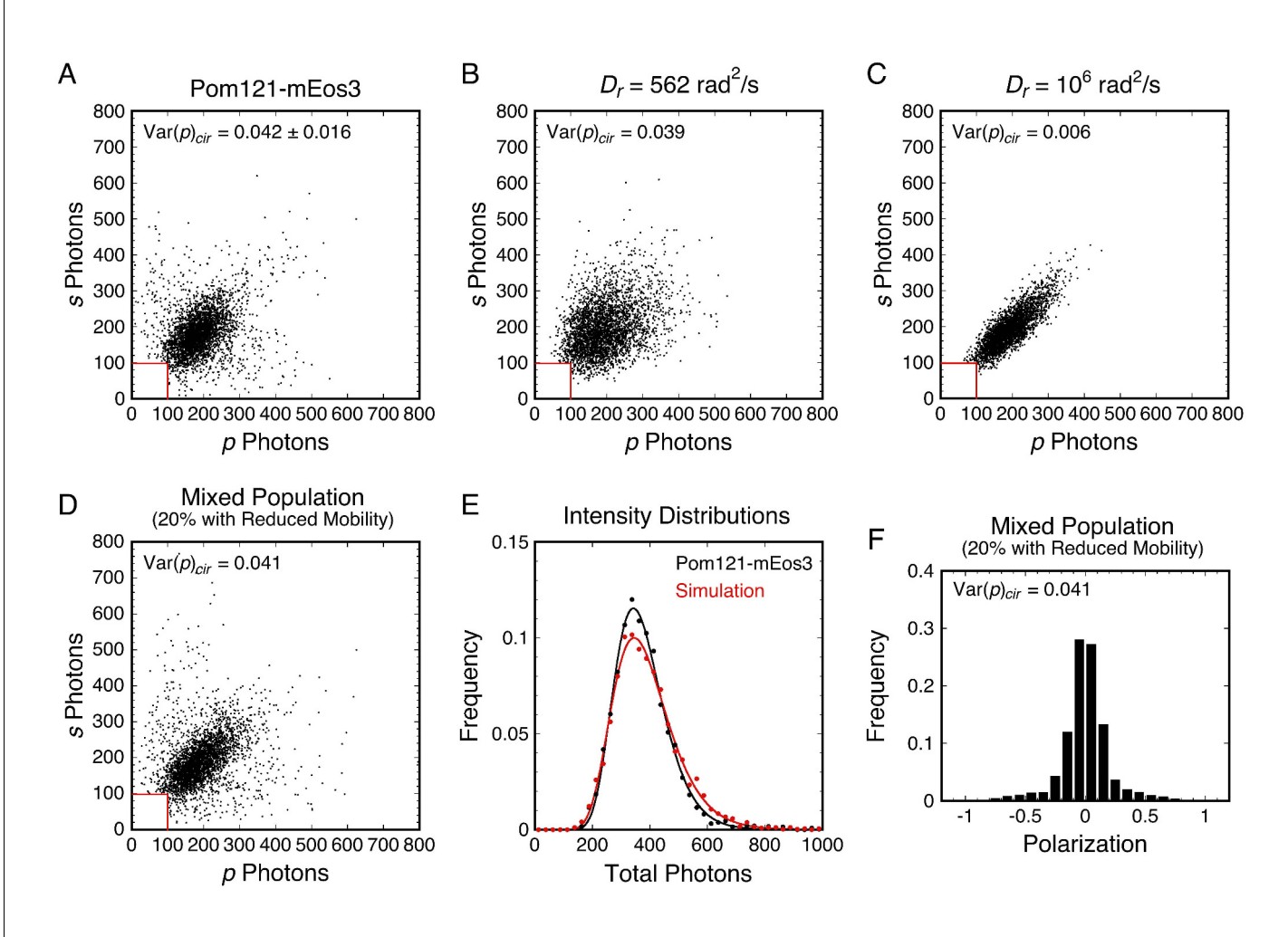

**Figure 6.** Analysis of rotational mobility within mixed populations. (A) Scatterplot of the numbers of photons collected in the *p*- and *s*-polarization channels for Pom121-mEos3 (data from **Figure 3D**; *N* = 3463). Each dot corresponds to one molecule. (B and C) Simulated photon scatterplots under the indicated conditions, assuming the $NA_{eff}$ = 1.02 in water ($\theta_{obj}$ = 50°; see **Appendix 1—figure 4**). The results in (B) were simulated with the value of $D_r$ corresponding approximately to the $Var(p)_{cir}$ in (A), whereas (C) corresponds to a higher rotational mobility at the high end of Regime III. The scatter is significantly wider in (A) than either (B) or (C), suggesting a mixed population. (D) Simulated photon scatterplot for a mixed population consisting of 80% of molecules with $D_{r,1}$ = 3160 rad²/s and 20% with $D_{r,2}$ = 100 rad²/s. Despite a relatively flat $Var(p)_{cir}$ curve in Regime III, the scatterplots become narrower as $D_r$ increases from $10^3$ to $10^6$ rad²/s (**Figure 6—figure supplement 3**). The prevalent rotational mobility in the population was chosen guided by the width of the central scatter in (A) compared with the scatterplots in **Figure 6—figure supplement 3**. (E) Total photon intensity histograms of the experimental results in (A) and the mixed population simulation in (D), fit to a log-normal distribution. (F) Polarization histogram from the results in (D) (compare with **Figure 3D**). These results support the hypothesis that the p-PALM data for Pom121-mEos3 arise from a mixed population. For appropriate visual comparison with the experimental dataset in (A), *N* ≈ 3500 for all simulations. The *red box* near the origin identifies the region eliminated by the 100 photon threshold. **Figure 6—figure supplements 1–8** show additional experimental and simulated photon scatterplots, and the effect of γ under highly anisotropic conditions on polarization histograms.

DOI: https://doi.org/10.7554/eLife.28716.020

The following figure supplements are available for figure 6:

**Figure supplement 1.** Simulated photon scatterplots using circular excitation.

DOI: https://doi.org/10.7554/eLife.28716.021

**Figure supplement 2.** Mixed rotational mobility populations for mEos3-Nup98 and mEos3-$^{700mid}$Nup98.

DOI: https://doi.org/10.7554/eLife.28716.022

**Figure supplement 3.** Mixed rotational mobility populations for mEos3-tagged Pom121.

DOI: https://doi.org/10.7554/eLife.28716.023

**Figure supplement 4.** Polarization histograms for γ far from the magic angle for $D_z/D_{xy}$ = $10^5$.

*Figure 6 continued on next page*

*Figure 6 continued*

DOI: https://doi.org/10.7554/eLife.28716.024

**Figure supplement 5.** Photon scatterplots for data in *Figure 5B* (wt and +Imp β1).

DOI: https://doi.org/10.7554/eLife.28716.025

**Figure supplement 6.** Photon scatterplots for the data in *Figure 3*.

DOI: https://doi.org/10.7554/eLife.28716.026

**Figure supplement 7.** Photon scatterplots and polarization histograms for data in *Figure 5B* (+WGA).

DOI: https://doi.org/10.7554/eLife.28716.027

**Figure supplement 8.** Mixed rotational mobility populations modeling mEos3-Nup98 middle mutants + WGA.

DOI: https://doi.org/10.7554/eLife.28716.028

## Combining PALM and p-PALM to probe for spatially distinct regions of varying rotational mobility

The relatively wide particle distribution maps observed in PALM experiments (*Figure 7*) suggested that regions of different rotational mobilities could potentially be resolved by combining the 2D localization maps generated via PALM with p-PALM rotational mobility information. This combined approach is challenging for the following reasons, which significantly reduced the size of current datasets. First, all p-PALM experiments reported thus far were performed by imaging the bottom of the nucleus, yielding spots from an approximately planar (2D) distribution of NPCs. In contrast, in order to obtain the position of the NE, PALM experiments were performed at the nuclear equator, yielding spots from a pseudo-linear (1D) distribution of NPCs. Thus, in combined PALM/p-PALM experiments, the NE was imaged at the nuclear equator, limiting the number of NPCs that could be simultaneously examined. Second, whereas entire trajectories were used for PALM, a single image per molecule was used for p-PALM to avoid biasing the data (see **Materials and methods**). This p-PALM constraint was retained in combined PALM/p-PALM experiments. And third, while $xy$ spatial information was readily obtained from p-PALM fluorescent spots, localization precision was reduced in combined PALM/p-PALM experiments compared with typical PALM data since the emission intensity was distributed over two images (partially compensated by increasing the excitation intensity). Nonetheless, we demonstrate here that PALM localizations can be combined with p-PALM measurements.

In order to explore the distribution of WGA-binding sites, we focused on the following question: did WGA reduce rotational mobility throughout the FG-network? We examined mEos3-Nup98 and RanGAP-mEos3, both of which yielded probe localizations widely distributed throughout the FG-network (*Figure 7*). The data were divided into those with $|p_{cir}| > 0.3$, which is only expected for molecules with $D_r < \sim 10^3$ rad$^2$/s, and those with $|p_{cir}| < 0.3$, which could be observed for molecules with any $D_r$ value (*Figure 2C and D*). Thus, if WGA reduced rotational mobility in a specific region of the FG-network, the two datasets should yield spatially distinguishable distributions. This was not observed (*Figure 8*). Therefore, the data support the hypothesis that WGA inhibits rotational mobility throughout most, if not all, of the FG-network.

## Discussion

In this study, we have developed a combined experimental and theoretical framework for inferring the local rotational mobility of macromolecules in crowded environments using p-PALM. The p-PALM method was used to examine the macromolecular crowding in the vicinity of mEos3 probes positioned at different locations within the FG-network of NPCs, and PALM was used to determine the spatial distributions of these probes. Our major findings are: (1) different FG-polypeptides and different domains within the same FG-polypeptide experience different environments that are distinguishable by a probe's rotational mobility; (2) in some cases, the binding of NTRs can increase crowding, thus producing significant differences in the properties of the local environments; and (3) WGA strongly influences rotational mobility throughout the FG-network, demonstrating that the local properties of the FG network can be modulated by embedded macromolecules. The implications of these findings provide a substantially improved understanding of the complexities of the FG-network, which we now discuss.

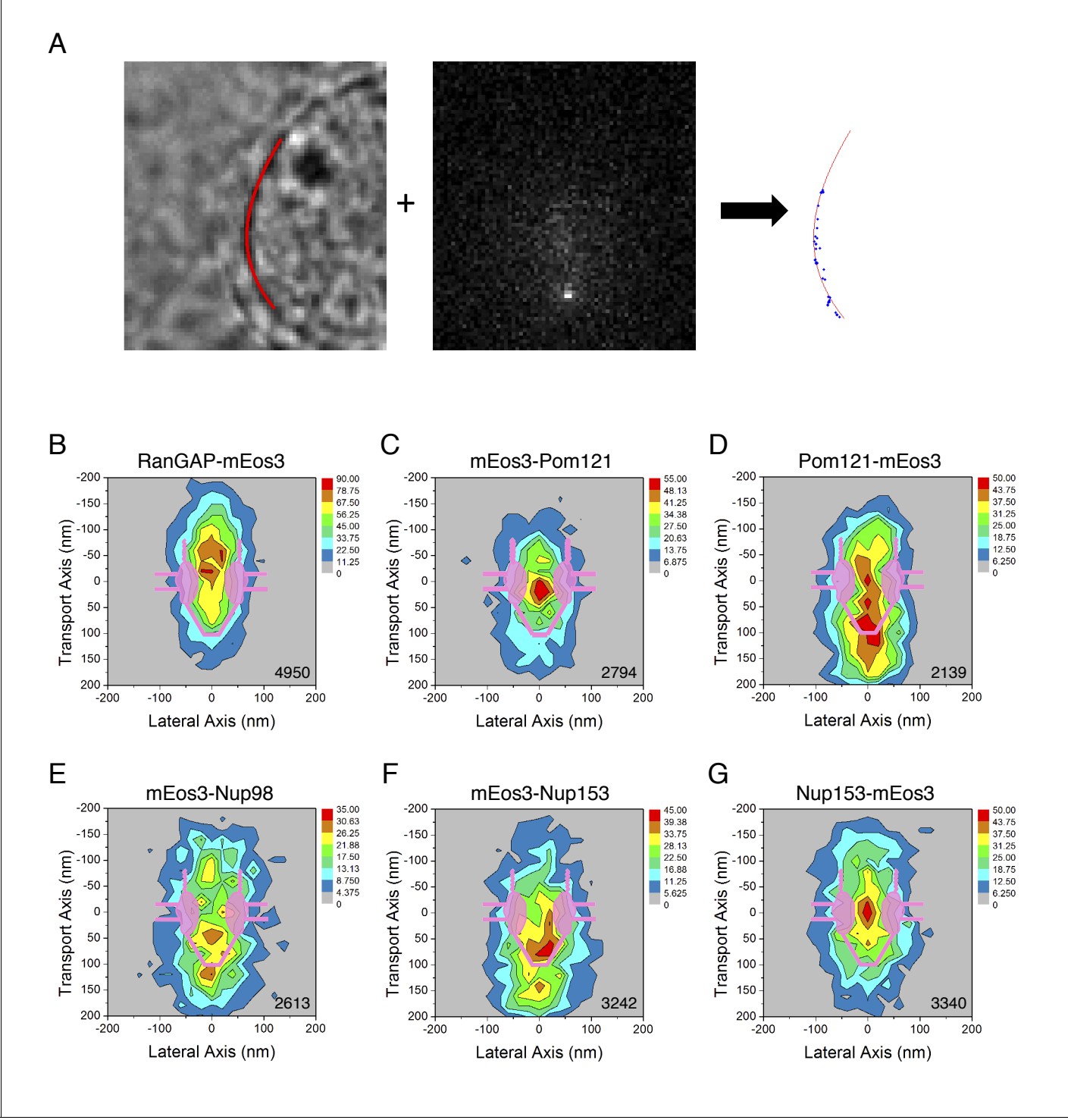

**Figure 7.** PALM 2D distribution maps of the mEos3 probe within the FG-Network attached to various proteins. (**A**) PALM imaging approach. (*left*) Nuclear envelope (NE) position. The position of the NE (*red*) was determined from a bright-field image at the nuclear equator, as described previously (*Yang and Musser, 2006b*). (*middle*) PALM imaging using circular excitation. In this example, the fluorescence from RanGAP-mEos3 was determined as in p-PALM imaging, except that a polarizer was not used and a single image per timepoint was collected (see *Video 2*). (*right*) mEos3 locations vis-à-vis the NE. The position of mEos3 in PALM spots was determined by 2D Gaussian fitting, and mEos3 positions (*blue*) were overlaid onto the NE position (*red*). (**B**)-(**G**) 2D particle distribution maps generated from PALM data. Spots/trajectories from different NPCs in PALM images (**A**) were aligned and overlaid, and two-dimensional (2D) distribution maps were generated by quantifying the number of localizations in a 400 nm x 400 nm area (20 nm

*Figure 7 continued on next page*

*Figure 7 continued*

x 20 nm 'pixels'; see **Materials and methods** for details). The total number of spots/map is indicated in the bottom right corner (from 3 to 6 cells; 6–10 NPCs/cell).

DOI: https://doi.org/10.7554/eLife.28716.029

The following figure supplement is available for figure 7:

**Figure supplement 1.** 2D distribution maps of the mEos3 probe within the FG-Network in the presence of WGA.

DOI: https://doi.org/10.7554/eLife.28716.030

The average positions vis-à-vis the transport axis for the mEos3 probe attached to FG-polypeptides are largely consistent with previous results, as indicted in the **Results** section. Thus, the PALM data support the hypothesis that these mEos3-tagged proteins behave similar to their wild-type counterparts, and therefore, they enable probing of the FG-network. The mEos3 probe on the N-terminus of Nup98 was widely distributed along the transport axis with localizations within the central pore region as well as relatively far from the NPC center on both the nuclear and cytoplasmic sides (*Figure 7E*). This broad distribution pattern is consistent with Nup98 anchoring sites on the inner and outer ring structures of the NPC scaffold (*Kosinski et al., 2016*; *Lin et al., 2016*) and on the cytoplasmic filaments and nuclear basket (*Frosst et al., 2002*; *Stuwe et al., 2012*), which together agree with the high copy number (48) for Nup98 (*Lin et al., 2016*; *Ori et al., 2013*). While the probe on the C-terminus of Pom121 also yielded a broad spatial distribution pattern, consistent with the long C-terminal FG-domain, labeling of the Pom121 N-terminus yielded a narrower distribution, consistent with anchoring of this part of the protein at the NE (*Figure 7C and D*). Two anchoring sites for Pom121 via its N-terminal domain to Nup155 and/or Nup160 on both the inner and outer ring complexes (*Kosinski et al., 2016*; *Lin et al., 2016*) is consistent with a stoichiometry of 16 copies/NPC (*Ori et al., 2013*), although such dual anchoring is unresolvable at our current resolution.

For RanGAP, there were a surprising number of localizations within the central pore and basket regions (*Figure 7B*), seemingly inconsistent with the cytoplasmic distribution expected considering the known binding site for SUMOlyated RanGAP on the cytoplasmic filaments (*Hutten et al., 2008*; *Mahajan et al., 1997*; *Matunis et al., 1998*; *Reverter and Lima, 2005*; *Wälde et al., 2012*). RanGAP has a role in heterochromatin assembly (*Nishijima et al., 2006*), it has both nuclear localization and nuclear export signals (*Feng et al., 1999*), and, although found predominantly at the NE, it is also found in both the cytoplasmic and nucleoplasmic compartments (*Mahajan et al., 1997*; *Matunis et al., 1998*). These data suggest that RanGAP trafficks through the NPC, which could result in trapping within the FG-network during cell permeabilization. Alternatively, RanGAP could have additional roles within the FG-network other than export complex disassembly on the cytoplasmic filaments. Notably, RanGAP is not expected to catalyze disassembly of RanGTP-containing export complexes without a RanBP (*Bischoff and Görlich, 1997*; *Bischoff et al., 1994*; *Güttler and Görlich, 2011*; *Kutay et al., 1997a*; *Okamura et al., 2015*), thus suggesting that only the portion of RanGAP attached to the cytoplasmic filaments (RanBP2) may be active (*Mahajan et al., 1997*; *Matunis et al., 1998*; *Wu et al., 1995*; *Reverter and Lima, 2005* #688; *Yokoyama et al., 1995*).

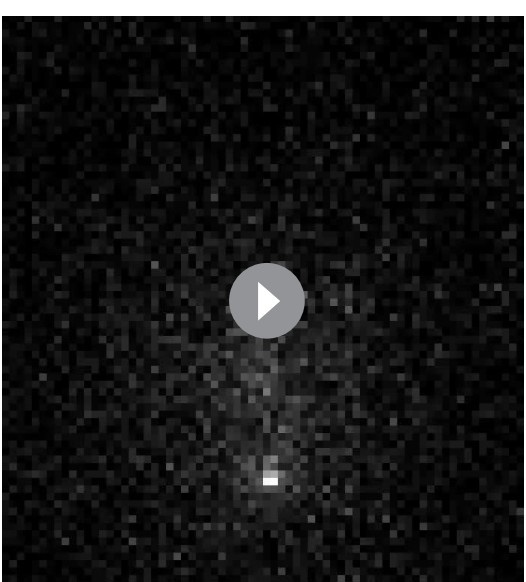

**Video 2.** PALM imaging of RanGAP-mEos3 at the nuclear equator. Imaging conditions were the same as for p-PALM, except that there was no polarizer and only a single image was collected per time interval. Fluorescent spots that appear and disappear arise from single mEos3 molecules. $t$ = 10 ms; 240 nm square pixels (see *Figure 7A*).

DOI: https://doi.org/10.7554/eLife.28716.031

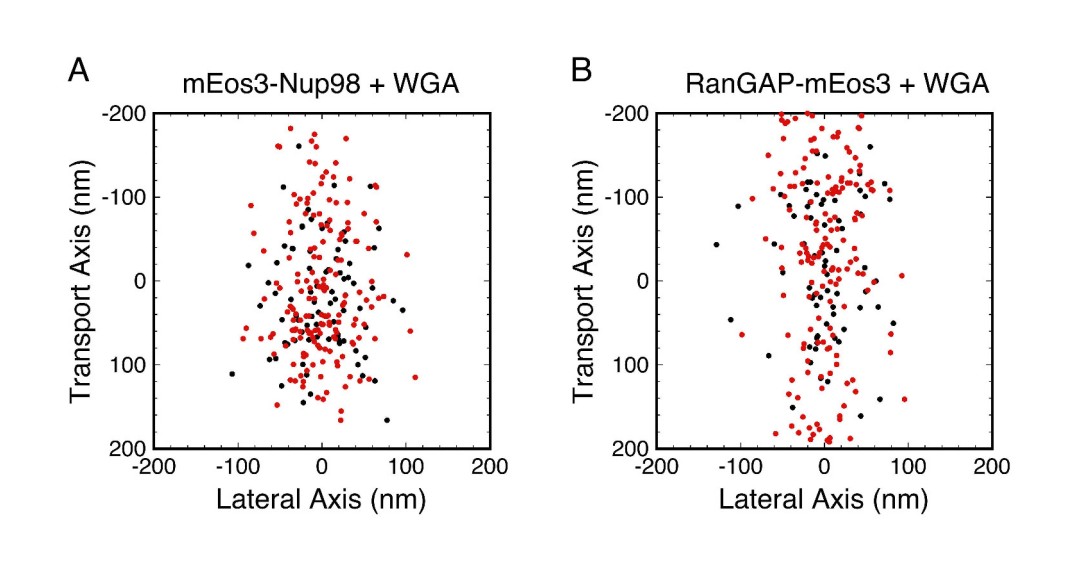

**Figure 8.** Combined PALM and p-PALM measurements of mEos3-Nup98 and RanGAP-mEos3 in the presence of WGA. Data were collected as in *Figure 7*, except that the emission was divided into *p*- and *s*-polarization channels. Each dot corresponds to one image from one probe molecule. (*red*) $1.2 > |p_{cir}| > 0.3$, which is only expected for slowly rotating molecules with $D_r < \sim 10^3$ rad$^2$/s; (*black*) $|p_{cir}| < 0.3$, which could be observed for molecules with any $D_r$ (see *Figure 2C and D*). The two color-coded populations have similar broad distributions in each panel, suggesting that WGA influences rotational mobility throughout most, if not all, of the FG-network. Total number of molecules and distribution widths (mean ± SD along the transport axis): (A) $N = 359$, (*black*) 17 ± 65 nm, (*red*) 5 ± 77 nm; (B) $N = 386$, (*black*) −8 ± 84 nm, (*red*) 21 ± 117 nm.

DOI: https://doi.org/10.7554/eLife.28716.032

The goal of our p-PALM approach was to identify regions of increased crowding within the FG-network and thus map the protein density distribution within the pore. While attaching the mEos3 probe to the end or to the middle of an FG-polypeptide could potentially introduce severe anisotropy in the rotational diffusion constants, the entirety of our results and simulations suggest that the probe's rotation mobility behavior is at most only slightly anisotropic or the angle between the dominant rotational axis and the transition dipole is near the magic angle. In either case, the probe's behavior largely resembled that of an untethered spherical particle. This was a somewhat surprising finding. However, this conclusion greatly simplifies the interpretation of p-PALM data since it substantially limits the parameter space that needs to be considered.

Since the mEos3 probe's rotational mobility behavior resembled that of an isotropic particle, comparison of the experimental $<p>_{lin}$ and Var$(p)_{cir}$ values with the simulation results for a spherical particle (*Figure 2*) enables the rotational mobility of the mEos3 probe under different conditions to be interpreted in terms of the approximate values of the average rotational diffusion coefficient ($D_r$). Assuming an mEos3 fluorescence lifetime > ~3 ns, which is true for most fluorescent proteins (*Bajar et al., 2016*; *Moeyaert et al., 2014*), the $<p>_{lin}$ values for all the conditions tested (*Supplementary file 1*) indicate that $D_r \leq \sim 10^6$ rad$^2$/s (*Figure 2* and Supplements). Since $D_r$ for mEos3 free in solution is ~$10^7$ rad$^2$/s (calculated for a sphere [*Loman et al., 2010*]), the experimental $<p>_{lin}$ values suggest that rotational mobility was reduced by at least an order of magnitude by crowding within the FG-network. Var$(p)_{cir}$ values of ~0.2–0.3 (*Supplementary file 1*) for the mEos3 probe under some conditions, in particular in the presence of WGA, indicate that the $D_r$ was reduced to <~100 rad$^2$/s (*Figure 2* and Supplements), that is, at least a 5 orders of magnitude change in rotational mobility from the free particle (for at least a fraction of the particles in the population, considering that most populations likely consisted of particle distributions with multiple $D_r$ values – see **Appendix 1**). The p-PALM method therefore enables detection of large changes in rotational mobility. Moreover, since the p-PALM technique measures the polarization of individual molecules, which allows calculation of the variance of the polarization, it permits discrimination between rotational diffusion behaviors at much lower $D_r$ values than traditional anisotropy approaches, in which fluorescence depolarization is governed by the fluorescence lifetime (*Lakowicz, 2006*).

Notably, using a bulk fluorescence anisotropy approach on yeast NPCs (*Atkinson et al., 2013*; *Mattheyses et al., 2010*; *Kampmann et al., 2011*), it was reported that some GFP probes were oriented when attached to some FG-polypeptides, particularly when they were near to the NPC scaffold. As the anisotropy signals were weak relative to homogeneous models, it appears that either the percentage of oriented molecules was low, or the orientation bias was weak. For either explanation, the assumption that probes were initially isotropically oriented in our random walk simulations is valid in most cases and leads to only minor errors in other cases. The fact that WGA had substantial effects on rotational mobility, as we observed here, and yet had very little, if any, effect on probe orientation (*Atkinson et al., 2013*) emphasizes the different physical parameters measured by the p-PALM and bulk fluorescence anisotropy approaches.

Under wild-type conditions, $<p>_{lin}$ and Var$(p)_{cir}$ values suggest that the $D_r$ was $10^3$–$10^6$ rad$^2$/s for the large majority of probe molecules, consistent with the high mobility expected for the FG-polypeptides and a dynamically flexible FG-network. Conditions that decreased the rotational mobility of the mEos3 probe have been interpreted to result from an increase in macromolecular crowding. A high density of macromolecules reduces molecular motion (*Dix and Verkman, 2008*; *McGuffee and Elcock, 2010*), presumably through an increased number of contacts with surrounding macromolecules. An mEos3 probe molecule within the FG-network can interact with FG-polypeptides, embedded macromolecules, or both. While the parameter Var$(p)_{cir}$ provided an initial indication of the reduction of rotational mobility due to crowding, it is still a population average, and a more refined picture was obtained by examining the full distribution of polarization values via polarization histograms and photon scatterplots. In multiple instances, two distinct rotational mobilities were necessary to explain the data, indicating heterogeneity in the environment around the different probe molecules. We consider it likely that most, if not all, of the high Var$(p)_{cir}$ values arose from mixed populations (*Figure 6* and discussion in **Appendix 1**), one sub-population of which had a relatively low rotational mobility ($<10^3$ rad$^2$/s). Therefore, despite being fused to a single location in a given protein, mEos3 probes often resided in multiple distinct environments, and variations in Var$(p)_{cir}$ values likely arose from both differences in local protein densities as well as differential partitioning between environments.

In all cases that we examined, WGA had a significant effect on the probe's rotational mobility (*Figures 4B* and *5B*). WGA binds to O-GlcNAc-modified Nups, of which there are at least five in humans (*Finlay and Forbes, 1990*; *Hülsmann et al., 2012*). The eight GlcNAc binding sites on the WGA dimer (*Schwefel et al., 2010*) suggest that it likely inhibited rotational mobility by non-covalently 'crosslinking' the FG-network. Considering that WGA affected the rotational mobility of mEos3 probes that were located both in the central pore and on the cytoplasmic and nucleoplasmic sides (*Figure 7—figure supplement 1*), the most parsimonious conclusion is that WGA binding sites are found throughout the FG-network. This conclusion is also supported by the combined PALM and p-PALM data (*Figure 8*). However, this conclusion that WGA binding sites are located throughout the FG-network is inconsistent with previous dSTORM microscopy studies that localized WGA to an ~40 nm diameter ring near the scaffold of the central pore (*Löschberger et al., 2012*). Electron microscopy using WGA-gold revealed a similar picture to the dSTORM study, although more central localizations were also revealed (*Akey and Goldfarb, 1989*). It is possible that freezing or fixation influences the distribution of WGA-binding sites in these previous studies, which could explain the apparent conflict with the p-PALM data collected here on functional pores. However, there is an alternate interpretation. WGA could significantly increase Var$(p)_{cir}$ by binding to a distinct region of the FG-network, and strongly influencing the rotational mobility of the sub-population of probes in the neighborhood of these binding sites. In a highly interconnected network, such as a hydrogel or NTR/FG-polypeptide mixed network, WGA binding in one localized spatial region could influence more distant regions of the FG network via long-range allosteric-type effects. In this way, binding of WGA to one or more discrete spatial locations could influence the rotational motion of a probe throughout the FG-network. This interpretation is consistent with the WGA localization data obtained via dSTORM and electron microscopy (*Akey and Goldfarb, 1989*; *Löschberger et al., 2012*). In the case of the mEos3 probe on the N-terminus of Pom121, whose rotational mobility was also significantly reduced by the WGA, it seems unlikely that WGA-binding interactions within the central pore could be transmitted across the NE membrane and influence the rotational mobility of a probe within the perinuclear space. For this reason, we have concluded that the N-terminus of Pom121 is likely to reside within the central pore (*Figure 4*).

Our results also shed light on the NTR distribution within the NPC. Nearly a hundred molecules of Imp β1 are bound within each NPC during steady-state (*Lowe et al., 2015*; *Paradise et al., 2007*; *Tokunaga et al., 2008*), consistent with the finding that NTRs have high affinities for FG-polypeptides (summarized in [*Tetenbaum-Novatt et al., 2012*]). A high number of NTRs within the FG-network increases macromolecular crowding, which is expected to influence the structural and functional properties of the FG-network (*Kapinos et al., 2014*; *Lowe et al., 2015*; *Schleicher et al., 2014*; *Vovk et al., 2016*; *Wagner et al., 2015*). In particular, the NTR-centric model postulates that NTR/FG-polypeptide effective affinities are higher nearest the NPC scaffold, and significantly weaker in the center of the pore, thus enabling rapid transport only through a narrow channel (~10–20 nm diameter) in the center of the ~50 diameter pore (*Kapinos et al., 2014*; *Schleicher et al., 2014*; *Wagner et al., 2015*). This model therefore predicts significantly higher macromolecular crowding near the scaffold anchor domain of FG-polypeptides. This hypothesis was directly tested via the p-PALM measurements on the mEos3 probe placed at different locations within the Nup98 FG-polypeptide, which support the hypothesis that crowding is indeed higher near the Nup98 anchor domain (*Figure 5*). Considering mixed populations, the average weighted rotational diffusion constant ($D_{r,ave}$) for the probes on mEos3-Nup98 and mEos3-$^{700mid}$Nup98 were ~2800 and ~910 rad$^2$/s, respectively (*Figure 6—figure supplement 1A C*). Similarly, the $D_{r,ave}$ for the probes on Pom121-mEos3 and mEos3-Pom121 were ~2500 and ~850 rad$^2$/s, respectively (*Figure 6—figure supplement 2*). Therefore, the rotational mobility data for both Pom121 and Nup98 suggest higher macromolecular crowding near the NPC scaffold than at the tips of the FG-polypeptides, consistent with the NTR-centric model. For comparison, in the absence of molecular crowding effects, $D_r \approx 1000$ rad$^2$/s for the mEos3 probe would correspond to a viscosity of ~$10^4$ cP. Notably, the probes on both mEos3-$^{700mid}$Nup98 and mEos3-Pom121 experience multiple environments distinguished by at least two distinct rotational mobilities (*Figure 6—figure supplements 1,2*). Since the mEos3 probes in both of these constructs are attached near their respective anchor domains, and thus cannot migrate to spatially distinct sites within the FG-network, the local environment must be heterogeneous within an individual NPC, or with respect to different NPCs. Consequently, crowding near the NPC scaffold is somewhat heterogeneous.

High time-resolution super-resolution methods on functional NPCs in unfixed cells will continue to be instrumental in deciphering the complex, amorphous biomaterial that is the FG-network. We demonstrated here that the p-PALM method allows examination of rotational mobility over a range of at least 6 orders of magnitude. This range can be tuned by both acquisition conditions and experimental design, offering significant advantages over bulk measurements of fluorescence anisotropy. The results have allowed us to infer the local binding interactions and molecular crowding within NPCs, and have elucidated multiple aspects of the structural and dynamic complexity of the FG-network. While dynamics are an essential feature of the FG-network, enabling both rapid transport and dynamic maintenance of the permeability barrier, the extent to which newly identified heterogeneities play a role in functional properties of the NPC remains to be explored. While we expect that p-PALM will enable further dissection of the intricacies of the FG-network, it is also well-suited for probing the nanoscale structure of other dense molecular aggregates, such as the poorly understood organization of numerous nucleoplasmic and cytoplasmic membrane-less compartments ('bodies'), for example, nucleoli, stress granules, and RNA and protein processing bodies (*Aumiller et al., 2014*; *Mitrea and Kriwacki, 2016*). These bodies typically contain high concentrations of proteins, and often nucleic acids, and their high densities promote phase separation. These highly crowded environments are difficult to probe because of their rapid dynamics, and often, their small size (<1 μm) (*Aumiller et al., 2014*; *Mitrea and Kriwacki, 2016*), and thus, the high time- and super-resolution capability of PALM and the molecular crowding sensitivity of p-PALM provide an important novel tool.

## Materials and methods

### Experimental methods

#### Human cell lines

HeLa cells (authenticated via STR profiling by ATCC) were cultured in Dulbecco's Modified Eagle Medium (GIBCO, Invitrogen, Carlsbad, CA) supplemented with 4.5 g/L glucose, 862 mg/L Gluta-

MAX-I, 15 mg/mL phenol red, 100 U/mL penicillin, 100 µg/mL streptomycin, and 10% (v/v) fetal bovine serum (GIBCO, Invitrogen, Carlsbad, CA). Cells were transfected with expression plasmids using Lipofectamine 2000 according to the manufacturer's instructions (Invitrogen). For Pom121-mEos3, mEos3-Pom121, mEos3-Nup98, and RanGAP-mEos3, a stable cell line was generated from a single cell clone. In all other cases, cells were split ~24 hr after transient transfections, and were examined ~24 hr after splitting. Cell lines were occasionally tested for mycoplasma contamination.

## Plasmids

Schematics of the mEos3 fusion proteins produced by the following plasmids can be found in *Figure 1—figure supplement 2*. Protein expressing inserts of all plasmids were confirmed by DNA sequencing.

*mEos3* – mEos3.1 was PCR amplified from the mEos3.1 N1 vector (*Zhang et al., 2012*) (gift of Dr. Tijana Jovanovic-Talisman, City of Hope, Duarte, CA) using forward primer 5′-GTCGCTAGCAGTGCGATTAAGCCAGACATGAAG-3′ and reverse primer 5′-CCAGAATTCTTATCGTCTGGCATTGTCAGGCAATCC-3′. The product was digested with Nhe1/EcoR1 (here and elsewhere, restriction sites within primers are underlined) and ligated into pRSETA-mEos2 (gift of Dr. Jie Xiao, Johns Hopkins University, Baltimore, MD) digested with the same enzymes, yielding plasmid pRSETA-mEos3, which produces N-terminally 6xHis-tagged mEos3.

*Pom121-mEos3* – mEos3.1 was PCR amplified from the mEos3.1 N1 vector using forward primer 5′-TGGCAATTGGGAGGAAGTGCGATTAAGCCAGACATG-3′ and reverse primer 5′-CTAACGCGTTTATCGTCTGGCATTGTCAGGCAATCC-3′. The product was digested with Mfe1/Mlu1 and ligated into plasmid peGFP-rPom121 (gift of Dr. Jan Ellenberg, EMBL, Heidelberg) digested with the same enzymes, yielding plasmid peGFP-rPom121-mEos3. Rat Pom121 was PCR amplified from the plasmid eGFP-rPom121 using forward primer 5′-TTTGCTAGCATGTCTCCGGCGGCTGCGGC-3′ and reverse primer 5′-GGGCAATTGTAACTTCTTGCGGGTGTGCTGCCTTCG-3′, which mutates the stop codon TTA on Pom121 to TTA. The rPom121 PCR product was digested with Nhe1/Mfe1 and ligated into peGFP-rPom121-mEos3 digested with the same enzymes, yielding plasmid prPom121-mEos3, which produces Pom121-mEos3.

*mEos3-Pom121* – mEos3.1 was PCR amplified from the mEos3.1 N1 vector using forward primer 5′-GGCGCTAGCATGAGTGCGATTAAGCCAGAC-3′ and reverse primer 5′-CGGAGATCTTCGTCTGGCATTGTCAGGCAATC-3′, which removes the stop codon on mEos3. The product was digested with Nhe1/Bgl2 and ligated into peGFP-rPom121 digested with the same enzymes, yielding plasmid pmEos3-rPom121(long linker). To remove the long linker between the mEos3 and rPom121 domains, rPom121 was PCR amplified from the plasmid peGFP-rPom121 using forward primer 5′-TTTAGATCTTCTCCGGCGGCTGCGGCGGCTGAC-3′ and reverse primer 5′-CCCACGCGTTTACTTCTTGCGGGTGTGCTGCC-3′. The PCR product was digested with Bgl2/Mlu1 and ligated into pmEos3-rPom121(long linker) digested with the same enzymes, yielding plasmid pmEos3-rPom121, which produces mEos3-Pom121.

*mEos3-Nup153* – Human Nup153 was PCR amplified from the plasmid peGFP3-Nup153 (gift of Jan Ellenberg, EMBL, Heidelberg) using forward primer 5′-TTAAGATCTGCCTCAGGAGCCGGAGGAGTCG-3′ and reverse primer 5′-CGGACGCGTTTATTTCCTGCGTCTAACAGCAGTC-3′. The product was digested with Bgl2/Mlu1 and ligated into plasmid pmEos3-Pom121 digested with the same enzymes, yielding plasmid pmEos3-Nup153, which produces mEos3-Nup153.

*Nup153-mEos3* – Human Nup153 was PCR amplified from the plasmid peGFP3-Nup153 using forward primer 5′-TATGCTAGCATGGCCTCAGGAGCCGGAGGAGTCG-3′ and reverse primer 5′-GGGACGCGTTTTCCTGCGTCTAACAGCAGTCTTTATCTTG-3′. The product was digested with Nhe1/Mlu1 and ligated into plasmid prPom121-mEos3 digested with the same enzymes, yielding plasmid pNup153. Then, mEos3.1 was PCR amplified from the mEos3.1 N1 vector using forward primer 5′- TTTACGCGTGGAGGAAGTGCGATTAAGCCAGACATG-3′ and reverse primer 5′-CTAACGCGTTTATCGTCTGGCATTGTCAGGCAATCC-3′, which includes a stop codon at the end of the coding sequence for mEos3. Plasmid pNup153 was digested with Mlu1 and the digested product was dephosphorylated by shrimp alkaline phosphatase (rSAP, New England Biolabs, Ipswich, MA). The mEos3.1 PCR product was digested with Mlu1, and ligated into the dephosphorylated pNup153 fragment, yielding plasmid pNup153-mEos3, which produces Nup153-mEos3.

*mEos3-Nup98* – Human Nup98 was PCR amplified from the plasmid peGFP-Nup98 (gift of Jan Ellenberg, EMBL, Heidelberg) using forward primer 5′-CCG<u>AGATCT</u>TTTAACAAATCATTTGGAA-CACCCTTTGG-3′ and reverse primer 5′-TAT<u>ACGCGT</u>TCACTGTCCTTTTTTCTCTACCTGAG-3′. The product was digested with Bgl2/Mlu1 and ligated into plasmid pmEos3-Pom121 digested with the same enzymes, yielding plasmid pmEos3-Nup98, which produces mEos3-Nup98. Plasmid pmEos2-Nup98, which produces mEos2-Nup98, was made identically.

*Mutant Versions of mEos3-Nup98* – Forward primer 5′-TTT<u>AGATCT</u>GCACAAAATAAACCAAC TGGCTTTGGC-3′ and reverse primer 5′-TAT<u>ACGCGT</u>TCACTGTCC TTTTTTCTCTACCTGAG-3′ were used to obtain a PCR fragment of Nup98 encoding amino acids 110–920, which was digested with Bgl2/Mfe1 and then ligated into plasmid pmEos3-Nup98 digested with the same enzymes, yielding plasmid pmEos3-[110tip]Nup98, which produces mEos3-[110tip]Nup98. Forward primer 5′-CCC<u>GCTAGC</u>ATGTTTAACAAATCATTTGGAACACCC-3′ and reverse primer 5′-TTT<u>GCTAG-C</u>AAAGGCATTGTTTTGGGATGAGAAGAG-3′ were used to obtain a PCR fragment of Nup98 encoding amino acids 1–109, which was digested with Nhe1, dephosphorylated as described above, and then ligated into plasmid pmEos3-[110tip]Nup98 digested with the same enzyme, yielding plasmid pmEos3-[110mid]Nup98, which produces mEos3-[110mid]Nup98. The 400tip, 400mid, 500tip, 500mid, 700tip, and 700mid (see *Figure 1—figure supplement 2* for nomenclature) mutant expression plasmids were constructed in a similar manner.

*RanGAP-mEos3* – Mouse RanGAP was PCR amplified from the plasmid pET11d-RanGAP (gift of Jan Ellenberg, EMBL, Heidelberg) using forward primer 5′-TTG<u>GCTAGC</u>ATGGCCTCTGAAGACA TTGCC-3′ and reverse primer 5′-GGG<u>CAATTG</u>GATGTTGTATAGCGTCTGCAGCAG-3′. The product was digested with Nhe1/Mfe1 and ligated into plasmid prPom121-mEos3 digested with the same enzymes, yielding plasmid pRanGAP-mEos3, which produces RanGAP-mEos3.

## Protein purification

The mEos3, Imp β1 (*Kutay et al., 1997b*), and Imp β2 (*Izaurralde et al., 1997*) proteins all contain a 6xHis-tag and were purified by NiNTA and size exclusion chromatography. Plasmids were transformed into *Escherichia coli* BL21(DE3), and protein production (1 L total culture) was induced with 1 mM isopropyl β-D-1-thiogalactopyranoside (IPTG). After overnight growth at 18°C, cells were centrifuged at 5000 $g$ for 10 min at 4°C. Pellets (~2 g) were resuspended on ice in 20 mM Tris, 1 M NaCl, 10 mM imidazole, 2 mM β-mercaptoethanol, pH 8.0 with protease inhibitors (1 mM phenylmethylsulfonyl fluoride, 2 μg/mL pepstatin, 2 μg/mL leupeptin, and 20 μg/mL soybean trypsin inhibitor). Resuspended cells were lysed via French press (3X at 1000 psi). The lysate was centrifuged at 50,000 $g$ for 15 min. The supernatant was added to a 1 mL Ni-NTA column (Qiagen, Germantown, MD). The $Ni^{2+}$ beads were washed with 40 mL 20 mM Tris, 1 M NaCl, 10 mM imidazole, 0.1% Triton X-100, pH 8.0 with protease inhibitors followed by 40 mL 20 mM Tris, 50 mM NaCl, 20 mM imidazole, pH 8.0 with protease inhibitors. Proteins were eluted with 10 mM Tris, 250 mM imidazole, 50 mM NaCl, pH 8.0. Major protein fractions were combined, concentrated with Microsep 10K or 30K Omega centrifuge filters (Pall Corp., NY), and then further purified by size-exclusion chromatography (Superdex 200; GE Healthcare, Wauwatosa, WI) using 20 mM Hepes, 110 mM KOAc, 5 mM NaOAc, 2 mM MgOAc, 1 mM EGTA, pH 7.3.

## Cell permeabilization

Cells were permeabilized and prepared for microscopy as previously described (*Izaurralde et al., 1997*; *Lyman et al., 2002*; *Yang et al., 2004*). In short, HeLa cells were grown on coverslips overnight, and an ~20 μL flow chamber was constructed from high-vacuum grease and a top coverslip. Cells were permeabilized by incubation with 40 mg/mL digitonin in import buffer (20 mM Hepes, 110 mM KOAc, 5 mM NaOAc, 2 mM MgOAc, 1 mM EGTA, 1 mM dithiothreitol, pH 7.3) for 2 min. Permeabilized cells were washed twice with import buffer containing 1.5% (w/v) polyvinylpyrrolidone (~360,000 g/mol). When used, WGA (Vector Laboratories, CA), Imp β1, or Imp β2 were incubated with cells for 10 min before imaging.

## Microscopy

Cells were imaged using a Zeiss 200M inverted microscope, equipped with an alpha plan-apochromat 100X, 1.46 NA oil-immersion objective (Zeiss). A 405 nm laser (100 mW, CUBE, Coherent, Santa

Clara, CA) was used to activate the fluorescent protein mEos3 (30–35 W/cm$^2$). The green fluorescence of mEos3 was obtained with an ArKr mixed-gas ion laser (2.5 W all lines, Stabilite 2018-RM, Spectra-Physics, Mountain View, CA) at 488 nm and the orange fluorescence of activated mEos3 was obtained with a solid-state laser (150 mW, Excelsior One, Newport, Santa Clara, CA) at 561 nm with an excitation density of 2 kW/cm$^2$. Both activation and excitation beams passed through a λ/4 wave plate (ThorLabs, Newton, NJ), generating circularly polarized light. A λ/2 wave plate was used to rotate the angle of linear polarization of the excitation lasers. The sample was illuminated via narrow-field epifluorescence (*Yang et al., 2004*), that is, a 300 µm pinhole was placed within a specimen-conjugate plane in the activation/excitation beam path, thereby restricting the specimen illumination area to ~7 µm diameter. After passing through a quad-bandpass filter (FF01-446/523/600/677-25, Semrock, Rochester, NY), the fluorescence emission was collected with an EMCCD camera (Evolve 128, Photometrics, Tucson, AZ). Image acquisition was controlled by MetaMorph (Molecular Devices, Sunnyvale, CA).

## Single molecule localization

Single mEos3 molecules were localized using an algorithm written in Matlab (The MathWorks, Inc., Natick, MA). Fluorescent spots were fit by a symmetric two-dimensional (2D) Gaussian function, whose center was assumed to be the particle's position. Particles in consecutive frames were considered to belong to the same trajectory when they were within a user-defined distance $r$. Considering the NPC size, we set $r$ = 200 nm.

## Image alignment matrix

Coverslip adsorbed 1 µM TetraSpeck microspheres were used for image alignment. Spot centers were determined by 2D Gaussian fitting. The coordinates of $n$ spots were summarized as:

$$X = (x_1\ x_2 \cdots x_n) \qquad X' = (x'_1\ x'_2 \cdots x'_n)$$
$$Y = (y_1\ y_2 \cdots y_n) \qquad Y' = (y'_1\ y'_2 \cdots y'_n)$$

where $x_1, x_2, …, x_n$ and $y_1, y_2, …, y_n$ are the coordinates of the spot centers in the $p$-polarization channel, and the primed values correspond to the coordinates of the spot centers in the $s$-polarization channel. ($X$, $Y$) and ($X'$, $Y'$) are related as follows:

$$\begin{bmatrix} X \\ Y \end{bmatrix} = f M_r \begin{bmatrix} X' \\ Y' \end{bmatrix} + B = f \begin{bmatrix} \cos\Omega & \sin\Omega \\ \sin\Omega & \cos\Omega \end{bmatrix} \begin{bmatrix} X' \\ Y' \end{bmatrix} + \begin{bmatrix} b_x \\ b_y \end{bmatrix} \qquad (1)$$

where $M_r$ is a rotation, $f$ is zoom factor, and $B$ is a translation. Fit parameters were determined by using the Matlab non-linear least squares fitting function 'lsqcurvefit', yielding $\Omega \approx 0$ and $f \approx 1$. Once $M_r, f,$ and $B$ were determined, the expected $(x_i, y_i)$ values were compared with the experimental $(x_i, y_i)$ values. The standard deviation of the differences, which was considered to be the alignment precision in both $x$ and $y$, was determined to be ~11 nm for each coordinate.

## PALM imaging

Samples were illuminated continuously with both 405 nm (activation beam) and 561 nm (measurement beam). Since the number of inactive mEos3 molecules decreased during data acquisition, the intensity of the activation laser was increased from 30 W/cm$^2$ to 35 W/cm$^2$ over forty 1000-frame movies. As a measure of the rapid photocycling in our experiments, ~80% of the mEos3 molecules in an imaging field were activated and photobleached in ~60 s at an imaging speed of 100 Hz.

## Localization precision

The localization precision in both the $x$ and $y$ dimensions can be estimated by (*Mortensen et al., 2010*):

$$\sigma_{x,y}^2 = 2\left[ \left(\frac{4}{3}\right)^2 \frac{s^2 + a^2/12}{N} + \frac{8\pi b^2 (s^2 + a^2/12)^2}{a^2 N^2} \right] \qquad (2)$$

where $s$ is the standard deviation of the Gaussian fit (~140 nm), $a$ is the effective pixel size (240 nm),

$b$ is the background noise (~3 photons/pixel), and $N$ is the total number of photons collected in the spot. *Equation 2* yielded an average static localization precision of $\sigma_{x,y}$ = ~17 nm for single mEos3 molecules in PALM experiments ($N \approx 350$ photons).

## The position of the nuclear envelope (NE) in PALM experiments

Brightfield images of the NE were taken at the beginning of each movie (1000 frames, 10 s duration), and the NE position was determined essentially as described earlier (*Yang and Musser, 2006b*). The pixel intensities within a row across the NE were fit with a 1D Gaussian function. Peak positions from rows covering the useful area were fit with a cubic function, which was considered to trace the NE.

## PALM fluorescent spot alignment and overlay

Clusters of three or more fluorescent spots with a maximal distance from their centroid of 200 nm were considered to arise from a single NPC. Inter-cluster distances were >400 nm. A normal to the NE that passed through a cluster centroid was defined as the transport axis of an NPC. Individual NPC transport axes were aligned (translated and rotated) to overlay fluorescent clusters.

## PALM 2D particle density maps

Origin 7 (OriginLab, Northampton, MA) was used to convert the $x$- and $y$-coordinates of aligned and overlaid fluorescent spots into a 20 × 20 matrix with a bin size of 20 nm. Contours were plotted in intervals of 12.5% of the maximum bin.

Two major sources of error contribute to the broad distributions of PALM localizations along the transport axis ($\sigma > 80$ nm: *Figure 7*). The error in the NE position ($\sigma_{NE}$) is likely substantial due to NE spatial fluctuations, a non-smooth path of the NE, and heterogeneity near the NE, which affects the accuracy of the bright-field NE localization algorithm. Since previous work has demonstrated resolution of activities occurring on opposite sides of the NPC with particle distribution widths of <50 nm (which includes localization error) (*Sun et al., 2013*; *Sun et al., 2008*), we estimate an upper limit of ~45 nm for $\sigma_{NE}$. The second major source of error is the particle localization error ($\sigma_{x,y}$ = ~17 nm; see Localization Precision section). While other sources of error exist (*Musser and Grünwald, 2016*), $\sigma_{NE}$ and $\sigma_{x,y}$ dominate and yet are insufficient to explain the broad particle distributions along the transport axis. The most likely additional major contributors to the broad PALM distribution maps are multiple anchoring sites for the labeled proteins and motion of the FG-polypeptides themselves. Higher resolution data is required for more precise conclusions on FG-polypeptide translational mobility and more refined maps of FG-polypeptide distributions and anchoring sites.

## Polarization PALM (p-PALM)

For p-PALM, the emission light was separated with a 50% polarizing beam splitter cube (PBS201, ThorLabs, Newton, NJ) mounted in an Optosplit III beamsplitter (Cairn Research, Kent, UK). The two polarization components were imaged simultaneously on the two halves of the EMCCD camera. A system-dependent factor, $g$, corrects for differences in photon collection efficiency by the $p$- and $s$-detection channels, and must be empirically determined to calculate the polarization according to:

$$p = \frac{(gI_p - I_s)}{(gI_p + I_s)} \tag{3}$$

where $I_p$ and $I_s$ are the intensities measured in the two detection channels. To estimate $g$, we measured the intensities of mEos3 molecules in 92% glycerol, which rapidly rotate on the data collection timescale (*Figure 3—figure supplement 1*), and therefore, $<p>_{cir}$ is assumed to be 0. Under these conditions $g = <I_s/I_p> = 0.92$.

## Polarization histograms

Except for the data collected in 92% glycerol (see *Figure 3—figure supplement 1*), only molecules that lasted for three or more frames were used for polarization analysis. Polarization values were only calculated for the second frame of each trajectory. This approach was designed to only include intensities that reflected photon emission over the entire frame (i.e., ensuring that photoactivation

or photobleaching did not occur during data collection) and to weight each trajectory/molecule equally. Both of these constraints are essential to accurately estimate rotational mobility. Note that pooling p-PALM images would increase the integration time and the number of photons collected, both of which influence the measurement (**Figure 2—figure supplements 2A,5**). Pixel-dependent background intensities were determined by averaging a 1000-frame movie collected at the end of the experiment after complete photobleaching of mEos3.

## Rotational random walk simulations

### Overview of the problem

For a single fluorophore molecule, the photons collected in a single image correspond to hundreds of excitation and emission cycles, during and between each of which the probe might rotate. To decipher how polarization measurements are affected by the various conditions that could occur in p-PALM experiments, we estimated the photons collected (intensities measured) in the *p*- and *s*-polarization channels using rotational random walk simulations. Rotational random walk trajectories were simulated using Microsoft Excel with the RiskAMP Monte Carlo Simulation Engine (https://www.riskamp.com).

A freely diffusing spherical particle has identical rotational diffusion constants for the molecule's three principle rotational axes. Such a particle reasonably approximates most globular proteins (**Loman et al., 2010**). However, for an mEos3 molecule tethered to an FG-polypeptide, restrictions on rotational mobility due to the tether point can be expected. mEos3 has a β-barrel structure (**Zhang et al., 2012**). It was tethered to FG-polypeptides via its N- or C-terminus (or both), each of which are at the bottom of this β-barrel. For simplicity, we assumed that the tether point was on the rotational *z*-axis, and that this was coincident with the β-barrel axis. Restricted movement of the tether point thus results in $D_z > D_x \approx D_y$ (**Appendix 1—figure 1**). We assumed that the excitation and emission transition dipole moments of mEos3 are parallel, which is the case for GFP and many fluorophores (**Ha et al., 1999**; **Inoué et al., 2002**), and therefore, we usually more simply refer to these as the transition dipole. The transition dipole was assumed to be in the *yz*-plane of the molecule, and, since the transition dipole of GFP is ~60° from the β-barrel axis (**Inoué et al., 2002**), we assumed this to be approximately true for mEos3 as well (**Zhang et al., 2012**). **Appendix 1—figure 2** demonstrates the effects of varying the angle (γ) between the transition dipole and the rotational *z*-axis.

### Summary of rotational random walk simulation algorithm

The initial output of the rotational random walk simulations was the number of photons collected in each of the two polarization channels. The approach is briefly summarized here – details follow in subsequent sections. The dipole's initial orientation was chosen randomly. For each time step, the dipole was rotated around its three principle rotational axes via an angular step randomly chosen from a normal distribution defined by its rotational diffusion constant. Three decisions were then used to determine if a photon was collected, and if so, to which detection channel it went: (1) the excitation of the probe was stochastically determined based on its excitation probability, given the illumination ellipticity and the orientation of the dipole; (2) the probability of photon collection was determined based on the dipole's orientation and the solid angle subtended by the objective NA; and (3) if a photon was collected, it was partitioned into either the *p*- or *s*-channel depending on probabilities determined by the dipole's orientation. Most rotational random walk steps did not result in the collection of a photon. For circular excitation, the number of steps ($N_s$) for most simulations was 1400, leading to an average of ~352 photons collected under high $D_r$ conditions (**Figure 2—figure supplement 5**), which matches well with the average of ~300–400 total photons collected in the two polarization channels in p-PALM experiments (**Supplementary file 1**). Under these simulation conditions, an average of 341 total photons are expected based on the average excitation efficiency (2/3) and the photon collection efficiency of the objective (~36.5%). For linear excitation, $N_s = 2800$ was used to compensate for the lower average excitation efficiency (1/3). Thresholding (see later) is responsible for the higher than expected average total photons collected, particularly under low $D_r$ conditions (**Figure 2—figure supplement 5**).

## Coordinate systems and rotation transformations

For the laboratory frame, we assumed a spherical coordinate system where φ describes the angle from the z-axis (optical axis), and θ describes rotation around the z-axis from the x-axis. Random walk steps consisted of three angular sub-steps, one each around the molecule's three principal rotation axes, which were randomly oriented at the beginning of each simulation. The molecule's coordinate system was continuously updated and output to the laboratory frame using vector cross product multiplication to determine perpendicular unit vectors. Rotations were calculated by quaternion multiplication as follows. A rotation of the unit dipole vector $a = a_x i + a_y j + a_z k$ by an angle $\phi$ around the unit rotation axis vector $u = u_x i + u_y j + u_z k$ was calculated as:

$$a' = q^{-1} a q \tag{4}$$

where $a'$ is the rotated vector and the quaternion ($q$) and its conjugate ($q^{-1}$) were given by,

$$q = \cos\frac{\phi}{2} + \left(u_x i + u_y j + u_z k\right)\sin\frac{\phi}{2} \tag{5}$$

$$q^{-1} = \cos\frac{\phi}{2} - \left(u_x i + u_y j + u_z k\right)\sin\frac{\phi}{2} \tag{6}$$

Angular sub-step sizes were randomly selected from a normal distribution centered around the starting position with a variance of $\phi^2 = 2D\tau$ (with $D = D_x$, $D_y$, or $D_z$, as required for the rotational axis), where $\tau$ is the duration of each angular step, or $\phi^2 = 2D\tau_m$, where $\tau_m$ is time between excitation and emission of the molecule. The $\tau_m$ at each step was randomly chosen from an exponential decay defined by the fluorescence lifetime ($\tau_F$) of mEos3, which was assumed to be 3.5 ns (**Adam et al., 2011**). Each rotational random walk cycle included an excitation step, rotation during the fluorescence lifetime (three independent sub-steps defined by $D_x$, $D_y$, and $D_z$), and rotation after the photon was emitted (three independent sub-steps). Since $t$ is the image integration time, $\tau = t/N_s$.

## Excitation probabilities and emission intensities

Excitation probability is proportional to the square of the magnitude of the electric field along the direction of the transition dipole (**Forkey et al., 2000**). Therefore, since we used a narrow-field excitation approach (i.e. with a very low NA for excitation), the excitation probability at each time step was set to $(\sin^2\varphi)(\cos^2\theta + (1/\varepsilon)\sin^2\theta)$, where the ellipticity is defined by $\varepsilon = E_x/E_y$, and where $E_x$ and $E_y$ correspond to the average magnitudes of the electric fields along the x- and y-laboratory axes. Typical values used in the simulations were $\varepsilon = 1$ (circularly polarized light) and $\varepsilon = 100$ (linearly polarized light).

For a fixed unit dipole with laboratory frame components (x, y, z), the emission intensities collected in the p- and s-channels are:

$$i_p = i_{tot}\left(K_1 x^2 + K_2 y^2 + K_3 z^2\right) \tag{7}$$

$$i_s = i_{tot}\left(K_2 x^2 + K_1 y^2 + K_3 z^2\right) \tag{8}$$

where

$$K_1 = \frac{3}{32}\left(5 - 3\cos\left(\theta_{obj}\right) - \cos^2\left(\theta_{obj}\right) - \cos^3\left(\theta_{obj}\right)\right) \tag{9}$$

$$K_3 = \frac{1}{32}\left(1 - 3\cos\left(\theta_{obj}\right) + 3\cos^3\left(\theta_{obj}\right)\cos^3\left(\theta_{obj}\right)\right) \tag{10}$$

$$K_3 = \frac{1}{8}\left(2 - 3\cos\left(\theta_{obj}\right) + \cos^3\left(\theta_{obj}\right)\right) \tag{11}$$

are constants determined from Axelrod's expressions (**Axelrod, 1979**) normalized such that $i_p + i_s = i_{tot}$ for $\theta_{obj} = 180°$ (all light collected) (**Ha et al., 1999**). Assuming an angular semiaperture of $\theta_{obj} = 74.1°$ estimated from an immersion oil index of refraction of $n = 1.518$ and NA = $n \sin(\theta_{obj})$

=1.46, these constants were calculated as $K_1 = 0.38$, $K_2 = 0.012$, and $K_3 = 0.15$ (but see **Appendix 1—figure 4**). For values of $\theta_{obj} < 180°$, $i_p + i_s + i_e = i_{tot}$ where $i_e$ is the intensity of photons that escaped detection. Thus, the probability that an emitted photon escapes detection is:

$$P_e = \frac{i_e}{i_{tot}} = 1 - \frac{i_p}{i_{tot}} - \frac{i_s}{i_{tot}} \tag{12}$$

and the probability that the photon is collected is $1 - P_e$. If a photon is collected, the probability that the photon is partitioned into either the p- or s-channel is:

$$P_p = \frac{i_p}{i_p + i_s} \tag{13}$$

$$P_s = \frac{i_s}{i_p + i_s} = 1 - P_p \tag{14}$$

Note that $i_{tot}$ ends up being simply a scaling factor that disappears upon calculation of these probabilities. Note also that the high NA implies that photons can be collected in either the p- or s-channels from molecules oriented with a z-axis component, which reduces the number of measured polarization values near the ±1 limits (**Figure 2**).

*Equation 12* was additionally verified using simulations in which it was determined if the propagation direction of the emitted photon allowed it to be captured based on $\theta_{obj}$. The photon's propagation direction was randomly chosen assuming that the propagation direction is proportional to $\sin^2\varphi_k$ (**Forkey et al., 2000**), where $\varphi_k$ is the angle between the transition dipole and the propagation direction. Identical results were obtained.

## Broadening intensity distributions and addition of background noise

In order that the simulation results more accurately reflected experimental intensity distributions and noisy single molecule data, emission intensities based on photon counts were broadened and background noise was added. Whereas histograms of total photon intensities from the simulations were Gaussian under high rotational mobility conditions, experimental intensity histograms were approximately three-fold wider and were better described by log-normal distributions (e.g. **Figure 6E**), consistent with previous observations (**Mutch et al., 2007**). The broader experimental intensity distributions likely arise from a variety of factors, including the amplification noise of the EMCCD camera (**Chao et al., 2013**), differential focusing, irregularities within different light paths (permeabilized cells provide complex scattering and refractive index changes), as well as dirt and aberrations/variations in the optics and pixel quantum efficiencies (**Mutch et al., 2007**). Conformational differences of the probe that affect photon output are also possible. The simulated intensity distributions were therefore broadened by random selection of intensities from log-normal distributions with scale parameter $\mu^* =$ the total photons collected at each time step and shape parameter $\sigma^* = 0.24$. The value of the shape parameter was guided by fits to experimental intensity distributions and agreement of the final simulation results to experiment. The value of $\sigma^*$ has essentially no effect on $<p>$ and Var(p) (**Figure 2—figure supplement 4**). Log-normal distributions have an advantage over normal distributions for the purpose described here, particularly for lower intensities, since negative values cannot occur. Background noise was added to the p- and s-intensities recalculated from the new total intensities, partitioned according to the original photon numbers. Background noise (in photons) was normally distributed with $\sigma_p = 15$ and $\sigma_s = 15$ for circular excitation, and $\sigma_p = 22$ and $\sigma_s = 15$ for linear excitation, which are the average noise levels in p-PALM experiments over the $5 \times 5$ pixel regions of interest used to obtain fluorescence intensities. This approach yielded intensity distributions, ps-photon scatterplots, and polarization histograms that resembled the experimental data (**Figure 6**).

## Thresholding

A threshold was implemented to distinguish single molecule signals from background noise in experimental measurements. For rapidly rotating particles, distinguishing single molecule signals from background was fairly straightforward. However, slowly rotating particles have strongly preferred transition dipole orientations, and therefore, intensity histograms were broader for such molecules,

as expected, since excitation probabilities and emission intensities depend on dipole orientation. In particular, dipoles with a strong z-component yielded fewer emission photons primarily due to poor excitation efficiency, and therefore, molecules with such preferred orientations were not reliably detected in our experiments. Our criteria for spot selection was ≥100 photons in either the p- or s-detection channel.

To ensure more accurate modeling of the experimental data, a 100 photon threshold was also implemented for the simulation results. When using linear excitation, this threshold eliminates ~50% of the p values < 0 when $D_r$ is low (Regime V; compare *Figure 2B* and *Figure 2—figure supplement 1B*). Consequently, under these conditions, a 100 photon threshold increases $<p>_{lin}$ from ~0 to ~0.24 (*Figure 2—figure supplement 4A*). This is a major change, and indicates an extreme sensitivity to imaging and analysis parameters. In contrast, Var(p)$_{lin}$ and Var(p)$_{cir}$ values are less sensitive to this threshold for all values of $D_r$ (*Figure 2—figure supplement 4B C*). Notably, however, Var(p)$_{cir}$ has a wider dynamic range than Var(p)$_{lin}$ at low $D_r$ values, indicating greater sensitivity to changes in rotational mobility. For circular excitation and small $D_r$ values, the threshold eliminates ~20% of the simulation values, which predominantly increases the low $D_r$ asymptote in Var(p)$_{cir}$ plots. Under these conditions, the thresholding selects against particles oriented with a strong z-component, for which x- and y-intensities are similar and low, leading to the selective elimination of p values near zero (see *Figure 2D*) and an increase in Var(p)$_{cir}$. Background noise also contributes to increasing this asymptotic value, although noise predominantly influences the high $D_r$ asymptote (*Figure 2—figure supplement 4C*). In contrast, the 100 photon threshold does not affect the high $D_r$ asymptote since at high $D_r$ values all measurements are above the threshold. Notably, thresholding has little influence in most of the region where Var(p)$_{cir}$ is most sensitive to $D_r$ (*Figure 2—figure supplement 4C*). These simulations indicate that measuring $<p>_{lin}$ is best for rotational diffusion regimes I-III and measuring Var(p)$_{cir}$ is best for regimes III-V.

## Single molecule vs. bulk results

Single molecule and bulk fluorescence data yield fundamentally different information under conditions of low rotational mobility. This is directly observed in *Figure 2A*, where the single molecule and ensemble polarization results are compared for linear excitation. The emission intensities collected in the p- and s-polarization channels, $I_p$ and $I_s$, are mixtures of the intensities emitted along the x-, y-, and z-axes of the laboratory frame, $I_x$, $I_y$, and $I_z$, respectively:

$$I_p = K_1 I_x + K_2 I_y + K_3 I_z \tag{15}$$

$$I_s = K_2 I_x + K_1 I_y + K_3 I_z \tag{16}$$

where $K_1$, $K_2$, and $K_3$ are defined in *Equations 9-11*. For x-polarized linear excitation, $I_y = I_z$, and therefore:

$$I_x = I_\parallel = \frac{(K_1 + K_3)I_p - (K_2 + K_3)I_s}{(K_1 - K_2)(K_1 + K_2 + K_3)} \tag{17}$$

$$I_y = I_z = I_\perp = \frac{K_1 I_s - K_2 I_p}{(K_1 - K_2)(K_1 + K_2 + K_3)} \tag{18}$$

Thus, the corrected polarization ($p_c$) and anisotropy ($r_c$) values calculated for individual molecules in p-PALM images are given by:

$$p_c = \frac{I_\parallel - I_\perp}{I_\parallel + I_\perp} = \frac{(K_1 + K_2 + K_3)(I_p - I_s)}{(K_1 - K_2 + K_3)I_p + (K_1 - K_2 - K_3)I_s} \tag{19}$$

$$r_c = \frac{I_\parallel - I_\perp}{I_\parallel + 2I_\perp} = \frac{(K_1 + K_2 + K_3)(I_p - I_s)}{(K_1 - 2K_2 + K_3)I_p + (2K_1 - K_2 - K_3)I_s} \tag{20}$$

Using these expressions, the corrected values for the mean polarization ($<p>_{lin,c}$) and the mean anisotropy ($<r>_{lin,c}$) measured in p-PALM experiments are given by:

$$<p>_{lin,c} = \frac{1}{N}\sum_{k=1}^{N} p_c \tag{21}$$

$$<r>_{lin,c} = \frac{1}{N}\sum_{k=1}^{N} r_c \tag{22}$$

and are plotted in *Figure 2A* and *Figure 2—figure supplement 3*, respectively. The bulk polarization ($p_{bulk}$) and anisotropy ($r_{bulk}$) were estimated by summing the intensities from 10,000 different molecules with random initial orientations:

$$p_{bulk} = \frac{\sum_k I_\parallel - \sum_k I_\perp}{\sum_k I_\parallel + \sum_k I_\perp} \tag{23}$$

$$r_{bulk} = \frac{\sum_k I_\parallel - \sum_k I_\perp}{\sum_k I_\parallel + 2\sum_k I_\perp} \tag{24}$$

These expressions yield low $D_r$ limiting values of 0.5 and 0.4, respectively (*Figure 2A* and *Figure 2—figure supplement 3*), as expected (*Lakowicz, 2006*). The bulk and p-PALM results differ for two reasons (discussed more fully in the text): (1) the microscope objective mixes polarizations (*Equations 15 and 16*); and (2) the values calculated from single molecule data weights each molecule identically (i.e. the polarization/anisotropy of each molecule counts the same no matter how many photons are emitted), whereas the values corresponding to bulk conditions weights the contribution of each molecule to the measured polarization/anisotropy (a single measurement) depending on the number of photons emitted. The bulk polarization and anisotropy under circular excitation are always 0.

Under single molecule conditions, the variance in the measured polarization values, Var($p$), provides a measure of the width of polarization histograms (see *Figure 2*). The variance was calculated as Var($p$) = $<p^2> - (<p>)^2$, where $<p^2>$ is the average square of the measured polarization values. Theoretically, using circular excitation, Var($p$) = $<p^2>$ since $<p>$ = 0 for all $D_r$ values. Consequently, since in most cases we obtained $<p>_{cir} \approx 0$ (see *Supplementary file 1*), we assumed that Var($p$)$_{cir}$ was equivalent to the experimentally determined $<p^2>_{cir}$. As defined here, Var($p$) does not exist under bulk conditions since the information from all individual molecules is integrated by the measurement into a single value. This fundamentally explains why circular polarized excitation provides rotational mobility information under slow rotation conditions in a single molecule p-PALM experiment, whereas no information is obtained in a corresponding bulk experiment.

## Inferring $D_r$ from Var($p$)$_{cir}$

A general analytical solution for the Var($p$)$_{cir}$ dependence on the rotational diffusion constants is a complex problem. A few special cases are described in the Appendix. Guided by these results, it became clear that Pade-like approximations could be used for curve fitting, which enables extraction of rotational diffusion constants from measurements of Var($p$)$_{cir}$. Here, we summarize our findings from a range of simulations where $D_x$, $D_y$, $D_z$, $t$, $\gamma$, background noise and threshold values were varied.

The three principle rotational axes, $x$, $y$, and $z$, of a particle are characterized by the rotational diffusion constants $D_x$, $D_y$, and $D_z$, respectively, with an average rotational diffusion constant given by $D_r = (D_x + D_y + D_z)/3$. For a freely diffusing sphere, $D_x = D_y = D_z = D_r$. For a sphere whose rotational motion is restricted via a tether point on its rotational $z$-axis, $D_x = D_y$, the average perpendicular rotational diffusion constant is $D_\perp = D_{xy} = (D_x + D_y)/2$, and $D_r = (2D_\perp + D_z)/3$. $D_{xz}$ and $D_{yz}$ are similarly defined. For a particle that is tethered via multiple constraints, is highly geometrically asymmetrical, or has its rotational diffusion differentially constrained by other means, $D_x \neq D_y \neq D_z$.

For a particle behaving similar to a freely diffusing sphere (i.e., $D_x \approx D_y \approx D_z$), we find that (e.g., see *Appendix 1—figure 3*):

$$\text{Var}(p)_{cir} = \frac{(1 + a_1 D_r t)}{\beta + a_2 D_r t + a_3 (D_r t)^2} + \alpha \tag{25}$$

where $\alpha$, $\beta$, and $a_{1\text{-}3}$ are fit parameters that depend on $\gamma$, noise, the average number of photons collected, and the threshold. The value of $\alpha$ is determined by the high mobility limit ($D_r t \rightarrow \infty$; lower asymptote), and is the same regardless of the relationship between the rotational diffusion constants. The value of $\beta$ (with a small contribution from $\alpha$) determines the low mobility limit ($D_r t \rightarrow 0$; upper asymptote). In most cases, $a_1 \approx 1$ works well. Using the quadratic equation, $D_r$ is obtained from the positive root of *Equation 25*:

$$D_r = \frac{1}{2 a_3 t} \left[ \frac{a_1}{\text{Var}(p)_{cir} - \alpha} - a_2 + \sqrt{(a_2^2 - 4 a_3 \beta) + \frac{4 a_3 - 2 a_1 a_2}{\text{Var}(p)_{cir} - \alpha} + \left( \frac{a_1}{\text{Var}(p)_{cir} - \alpha} \right)^2} \right] \tag{26}$$

which allows calculation of $D_r$ from a Var$(p)_{cir}$ measurement.

In more complex situations (e.g., $D_x \approx D_y \ll D_z$ or $D_x \ll D_y \ll D_z$), we find that Var$(p)_{cir}$ is well-approximated by:

$$\text{Var}(p)_{cir} \approx (1 + a_1 D_r t) \left[ \frac{1}{\beta_1 + a_2 D_r t + a_3 (D_r t)^2} + \frac{1}{\beta_2 + a_4 D_r t + a_5 (D_r t)^2} \right] + \alpha \tag{27}$$

The $\beta_2$ value determines an intermediate plateau (e.g., see *Appendix 1—figure 3*). Parameters from simpler situations can be used as a guide to fit data from more complex situations and often work well, but the parameter values are not strictly the same. We emphasize that *Equations 25 and 27* are empirical solutions used to obtain smooth fits of simulation data. As far as we know, the constants $a_{1\text{-}5}$ do not have a physical interpretation.

## Error in Var$(p)_{cir}$ measurements

The simulation results were used to estimate the number of experimental data points required to obtain a reasonable estimate of Var$(p)$ and the error in the measurement. Datasets of 500–2000 measurements were randomly divided into 4 equivalently sized datasets, and means and standard deviations (SDs) were calculated. As shown in *Figure 2—figure supplement 6*, an experimental Var$(p)_{cir}$ value obtained from ~2000 (4 × 500) measurements differs from the 'true' value by an average of ~2–3%. The 'true' value was included in the range defined by the mean ±SD approximately 85% of the time, and thus this range defines the 85% confidence interval. Errors (SDs) for all experimental $<p>$ and Var$(p)$ values were calculated in this manner (from 4 equivalently sized datasets), and are summarized in *Supplementary file 1*.

# Acknowledgements

We thank Jan Ellenberg for Nup98 and Pom121 expression plasmids, Jie Xiao for the mEos2 plasmid, and Tijana Jovanovic-Talisman for the mEos3 plasmid and for thoughtful discussions.

# Additional information

### Funding

| Funder | Grant reference number | Author |
|---|---|---|
| National Institutes of Health | GM084062 | Siegfried M Musser |
| Welch Foundation | BE-1541 | Siegfried M Musser |
| Canadian National Science and Engineering Research Council | RGPIN-2016-06591 | Anton Zilman |

The funders had no role in study design, data collection and interpretation, or the decision to submit the work for publication.

## Author contributions

Guo Fu, Conceptualization, Data curation, Formal analysis, Investigation, Visualization, Methodology, Writing—original draft, Writing—review and editing; Li-Chun Tu, Methodology, Writing—review and editing; Anton Zilman, Conceptualization, Formal analysis, Funding acquisition, Visualization, Methodology, Writing—review and editing; Siegfried M Musser, Conceptualization, Formal analysis, Supervision, Funding acquisition, Visualization, Methodology, Project administration, Writing—review and editing

## Author ORCIDs

Anton Zilman ⑩ http://orcid.org/0000-0002-8523-6703
Siegfried M Musser ⑩ https://orcid.org/0000-0002-7793-2557

## Decision letter and Author response

Decision letter https://doi.org/10.7554/eLife.28716.046
Author response https://doi.org/10.7554/eLife.28716.047

## Additional files

### Supplementary files

- Supplementary file 1. Table S1: $<p>_{cir}$ and $<p^2>_{cir}$ Values, Table S2: $<p>_{lin}$ and $Var(p)_{lin}$ Values.
DOI: https://doi.org/10.7554/eLife.28716.033

- Transparent reporting form
DOI: https://doi.org/10.7554/eLife.28716.034

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

## Appendix 1

DOI: https://doi.org/10.7554/eLife.28716.035

### Additional considerations for interpreting p-PALM data

#### Rotational anisotropy

A probe's rotational mobility is determined by its three principle rotational diffusion constants $D_x$, $D_y$, and $D_z$, which are related to the average rotational diffusion constant by $D_r = (D_x + D_y + D_z)/3$. The relationship between the values of these rotational diffusion constants can have a significant effect on the $<p>$ and Var($p$) obtained for a given $D_r$ value. A critical parameter is the angle ($\gamma$) between the transition dipole and the major (dominant) rotational axis. Values of $\gamma$ far from the magic angle of 54.7° (**Axelrod, 1989**) have a substantial effect on p-PALM measurements under highly anisotropic conditions (**Appendix 1—figures 1–3**).

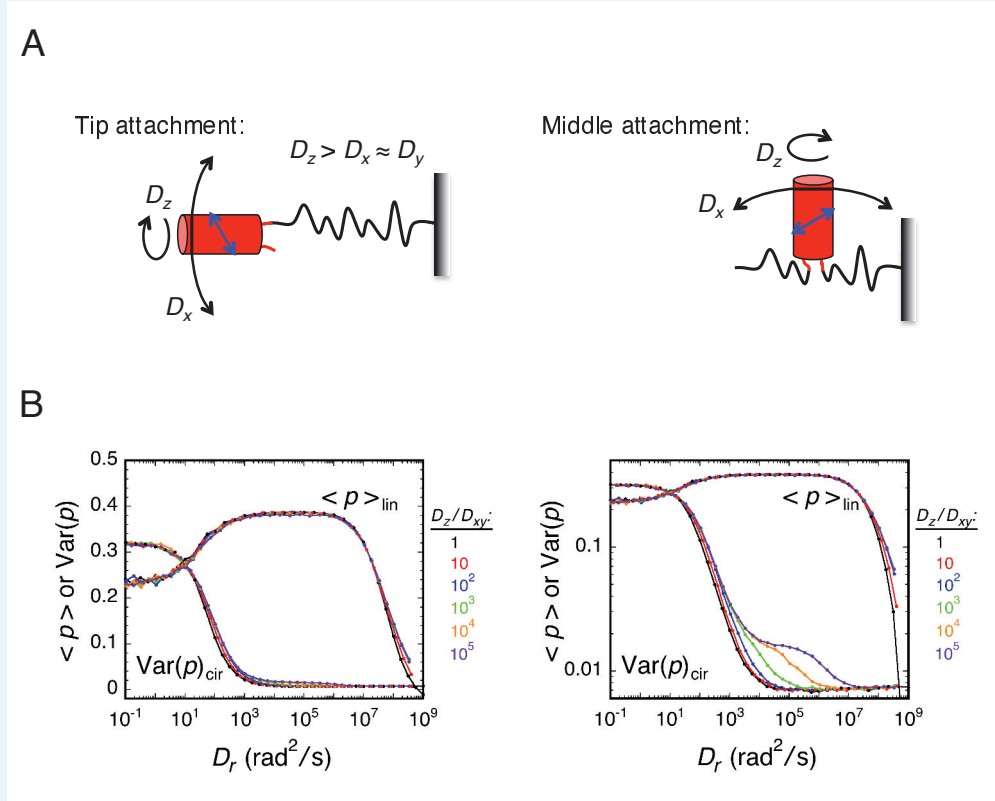

**Appendix 1—figure 1.** Effect of rotational anisotropy, expressed through the $D_z/D_{xy}$ ratio, on $<p>_\text{lin}$ and Var($p$)$_\text{cir}$. (**A**) Rotational modes for an mEos3 protein attached to the tip or middle of an FG-polypeptide. For 'tip' attachment, free rotation of mEos3 (*red*) is expected around the bonds in the tether (polypeptide link). 'Middle' attachment could potentially inhibit *z*-axis rotation because this would require the linked FG-polypeptides to twist in a coordinated fashion. Rotational motion about the three rotational axes could potentially be differentially influenced by local crowding and the motion of the FG-polypeptide domain(s) to which the probe is attached. The transition dipole (*blue*) is assumed to be within the *yz*-plane at an angle of ~60° to the rotational *z*-axis (assumed to be approximately equivalent to the β-barrel axis). (**B**) Effect of the $D_z/D_{xy}$ ratio. Rotational random walk simulations were used to estimate $<p>_\text{lin}$ and Var($p$)$_\text{cir}$ for different $D_z/D_{xy}$ ratios, assuming $t$ = 10 ms. Under these conditions, the effect of the $D_z/D_{xy}$ ratio is relatively minor (but see **Appendix 1—figures 2** and **3**). The panel on the *right* shows the same information as the one on the *left*, except that the ordinate is log-scaled to more easily reveal the biphasic relationship of Var($p$)$_\text{cir}$ with $D_r$ at high $D_z/D_{xy}$ ratios.

DOI: https://doi.org/10.7554/eLife.28716.036

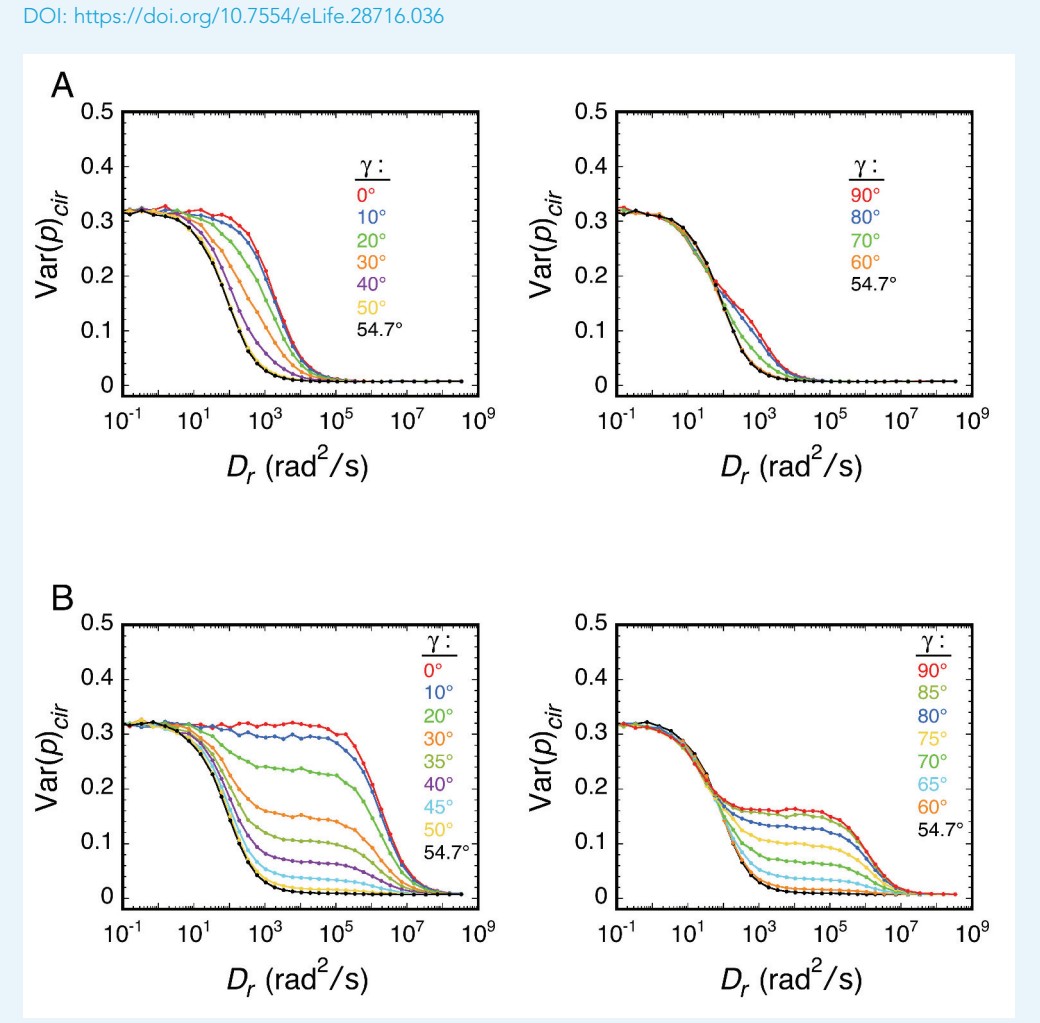

**Appendix 1—figure 2.** Effect of $\gamma$ on the relationship between Var$(p)_{cir}$ and $D_r$. Rotational random walk simulations were used to estimate Var$(p)_{cir}$ for different $\gamma$ values ($t$ = 10 ms). (**A**) $D_z/D_{xy}$ = 100. (**B**) $D_z/D_{xy}$ = 100,000.

DOI: https://doi.org/10.7554/eLife.28716.037

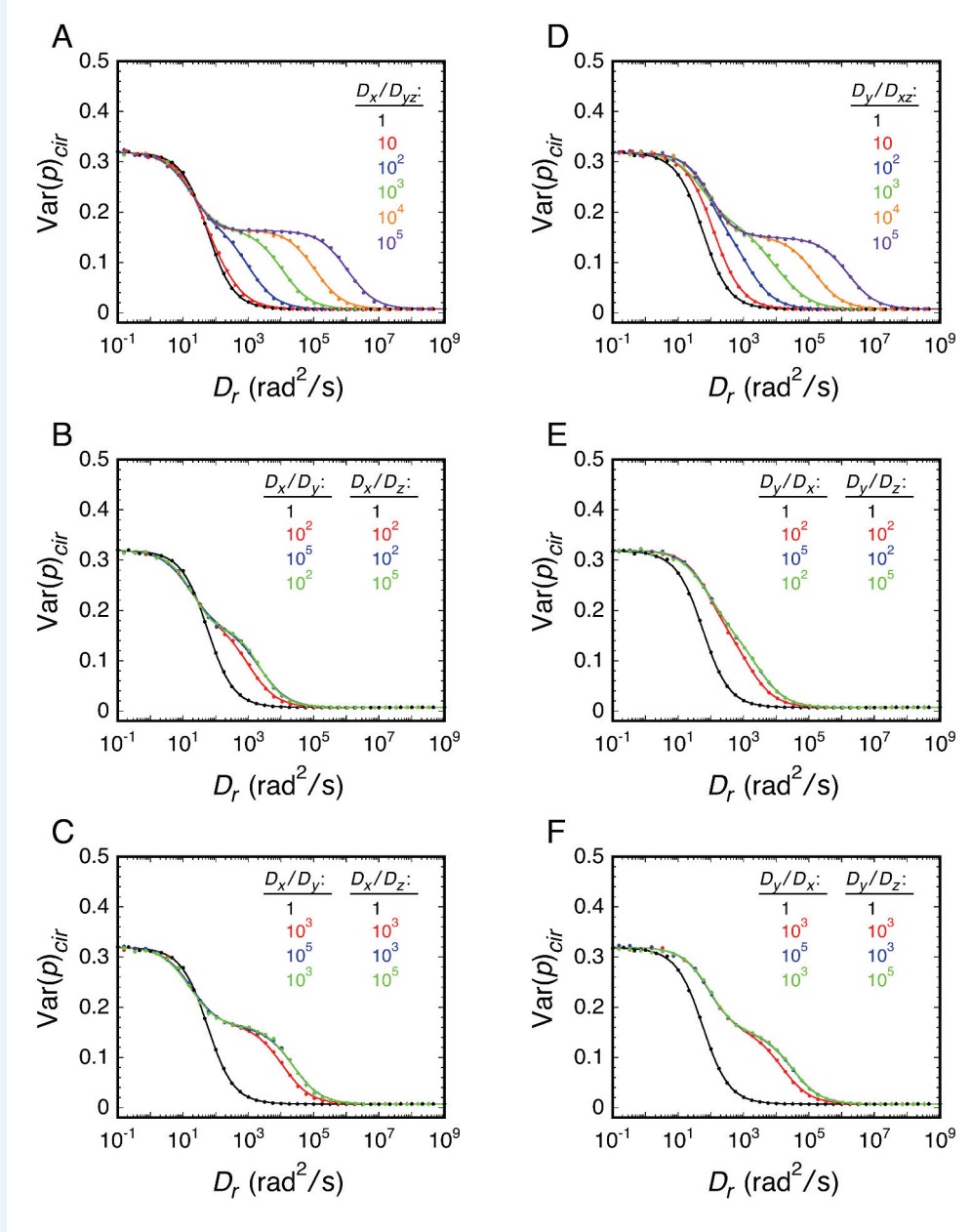

**Appendix 1—figure 3.** Effect of rotational diffusion constant anisotropy on the relationship between Var($p$)$_{cir}$ and $D_r$. For all results in this figure, $\gamma = 60°$ and the transition dipole was in the $yz$-plane. (**A**) Effect of the $D_x/D_{yz}$ ratio, where $D_{yz} = D_y = D_z$. (**B and C**) Effect of $D_x > D_y$, $D_x > D_z$, and $D_y \neq D_z$. Since the transition dipole was in the $yz$-plane, $D_x = 100D_y = 100D_z$ is equivalent to $\gamma = 90°$ in **Appendix 1—figure 2A**. (**D**) Effect of the $D_y/D_{xz}$ ratio, where $D_{xz} = D_x = D_z$. (**E and F**) Effect of $D_y > D_x$, $D_y > D_z$, and $D_x \neq D_z$. Since the transition dipole was in the $yz$-plane 60° from the $z$-axis, $D_y = 100D_x = 100D_z$ is equivalent to $\gamma = 30°$ in **Appendix 1—figure 2A**. In all cases, $D_r = (D_x + D_y + D_z)/3$ and $t = 10$ ms. Results were fit using **Equations 25 and 27**. In (**B**), (**C**), (**E**), and (**F**), the *green* and *blue* curves overlap. The large effects of diffusion constant anisotropy shown here contrast with the small effects shown in **Appendix 1—figure 1**. The latter results from the assumption that $\gamma = 60°$, which is near the magic angle of 54.7° (see **Appendix 1—figure 2**).

DOI: https://doi.org/10.7554/eLife.28716.038

In most of the reported experiments, the mEos3 probe was tethered to FG-polypeptides via its N- or C-terminus, both of which are at the bottom of its β-barrel structure (*Zhang et al., 2012*). Consequently, we assumed that the tethered probe's rotational z-axis was well-approximated by its β-barrel axis. The transition dipole of GFP subtends an ~60° angle from the β-barrel axis (*Inoué et al., 2002*), and we assumed that this is approximately true for mEos3 (*Zhang et al., 2012*). With these constraints, if the principal rotational diffusion constants were non-identical, we assumed that z-axis rotation of the probe at the tip of an FG-polypeptide would be the most facile (*Appendix 1—figure 1A*). Rotational random walk simulations for various $D_z/D_{xy}$ ratios assuming $D_z > D_{xy}$ (where $D_\perp \equiv D_{xy} = D_x = D_y$) indicated that for γ = ~60° (i.e, near the magic angle), the $D_z/D_{xy}$ ratio has little detectable influence on the relationship between $<p>_{lin}$ or $Var(p)_{cir}$ and $D_r$ (*Appendix 1—figure 1B,C*). These simulations thus suggest that the mEos3 probe on tip-labeled FG-polypeptides will yield p-PALM data similar to that expected for a free particle.

## Numerical aperture (NA)

For single molecule fluorescence experiments, a high NA objective is typically desired to maximize photon collection efficiency. However, the ranges of $<p>$ and $Var(p)$ obtained in p-PALM experiments depends on the NA, and both are broader at lower NA values (*Appendix 1—figure 4*). A major factor that leads to a reduction in the nominal NA is the spherical aberration that results from a refractive index mismatch, which occurs when using an oil immersion lens for an aqueous sample (*Appendix 1—figure 5*) (*Mondal and Disaspro, 2014*). An additional, albeit minor, factor reducing the NA can be a high scattering sample, which preferentially results in the loss of photons emitted at high angles relative to the transport axis due to the longer path in the scattering medium (*Theer and Denk, 2006*). Both of these factors contribute to reducing the NA in the reported p-PALM experiments.

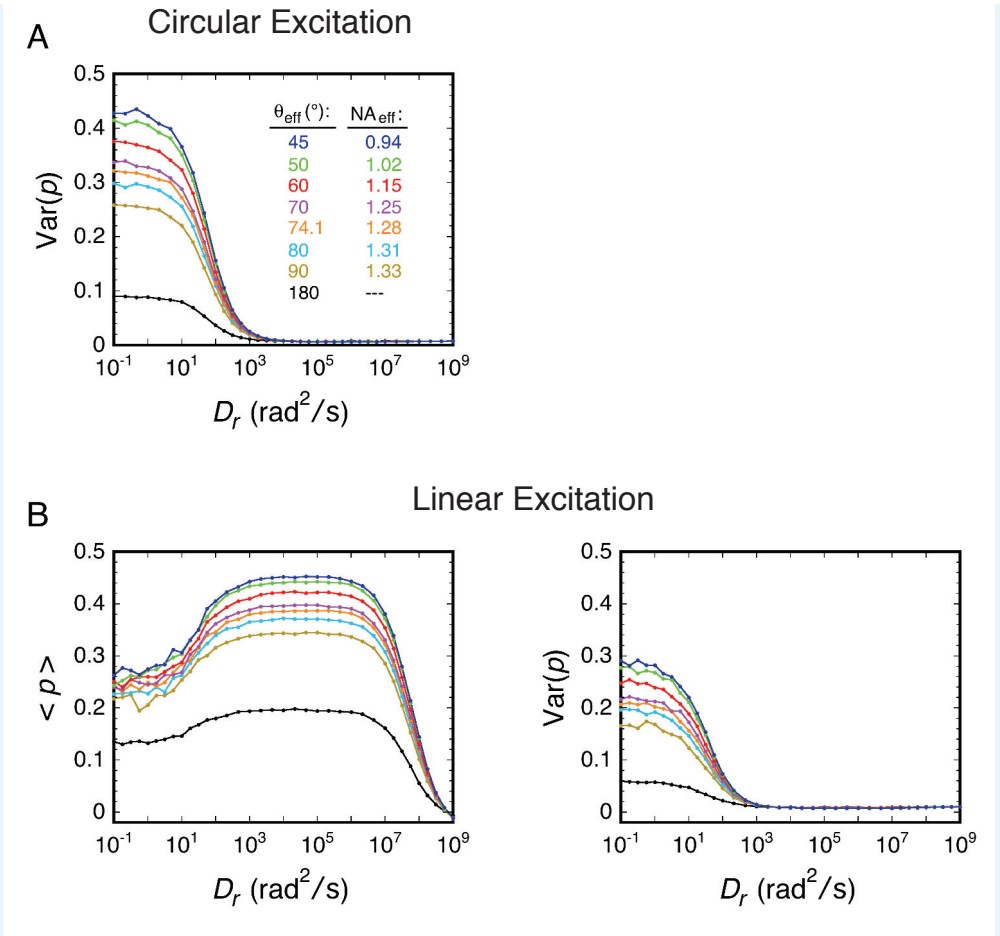

**Appendix 1—figure 4.** Effects of the numerical aperture. (**A**) Effect of the NA on Var(p)$_{cir}$. (**B**) Effect of the NA on $<p>_{lin}$ and Var(p)$_{lin}$. For most of the rotational random walk simulations reported in this paper, we have assumed an angular semi-aperture of $\theta_{obj}$ = 74.1°, where $\theta_{obj}$ is the half-angle of the cone of light recovered by the microscope objective. This $\theta_{obj}$ was calculated in the normal way (**Ha et al., 1999**), assuming an immersion oil refractive index of $n$ = 1.518 and NA = $n$ sin($\theta_{obj}$) where NA = 1.46 is the value inscribed on the objective. This approach assumes that the sample is embedded in a medium with refractive index equivalent to that of immersion oil. The NA$_{eff}$ for an aqueous sample depends on a variety of factors, including distance from the surface, spherical aberrations, and scattering. Our p-PALM data are most consistent with $\theta_{eff}$ ≈ 50–60° (see text for details). In the figure, NA$_{eff}$ = 1.33 sin($\theta_{eff}$), where $n$ = 1.33 is the refractive index for water. $\theta_{eff}$ = 180° describes a perfect objective (all light collected), and is compared with an analytical calculation in **Appendix 2—figure 3** (see **Appendix 2**). $\theta_{eff}$ = 90° represents the theoretical maximum with current objective designs. The color key is the same for all graphs.

DOI: https://doi.org/10.7554/eLife.28716.039

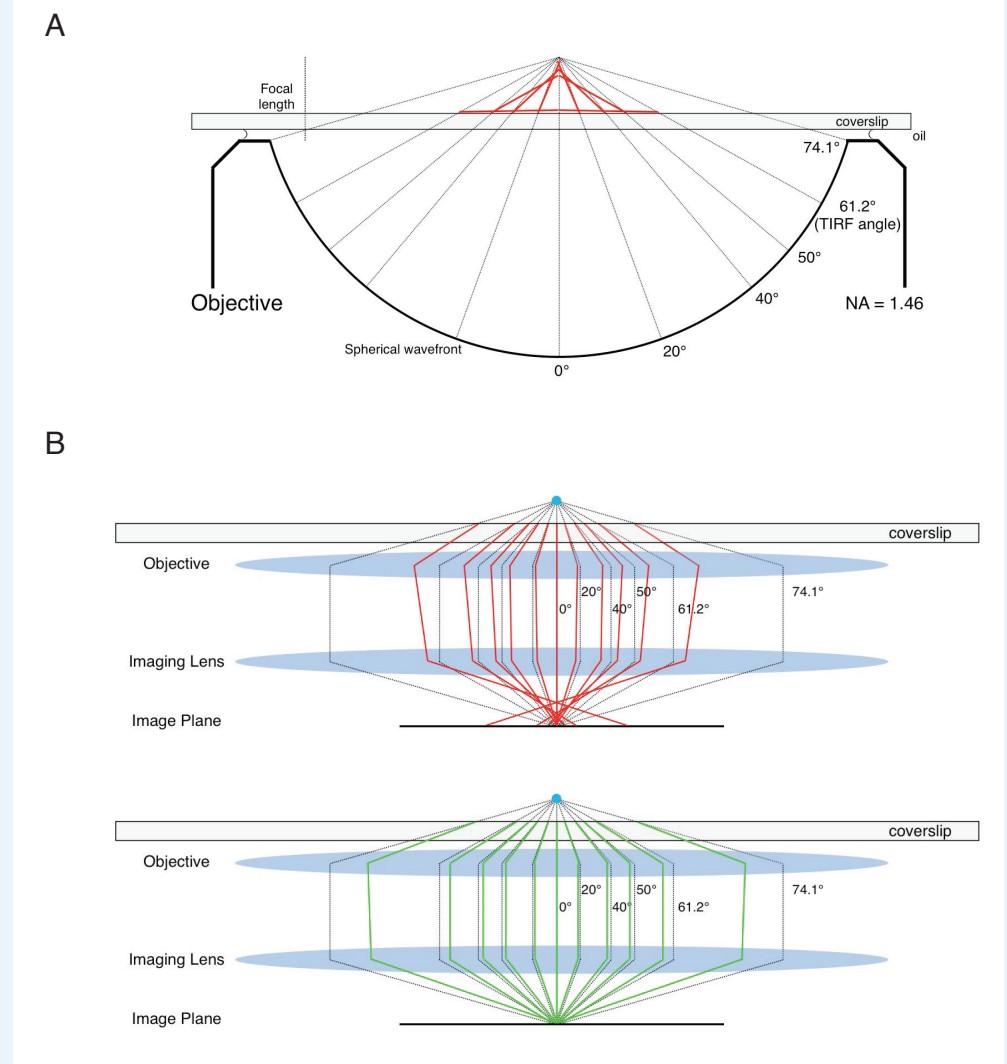

**Appendix 1—figure 5.** Effect of spherical aberration on $NA_{eff}$. Refractive index mismatch causes spherical aberration. (**A**) Effect of refractive index mismatch on the point spread function (PSF). When an oil-immersion objective is used to illuminate a sample in identical refractive index medium ($n = 1.518$), all rays converge upon a single focal point (*dashed lines*). When illuminating an aqueous sample, the rays are refracted (*red*), and the PSF is elongated along the optical axis. (**B**) Effect of refractive index mismatch on image quality. (*top*) Imaging an aqueous sample ($n = 1.33$) with an oil-immersion objective leads to a poorly focused image with loss of photons emitted at high angles to the optical axis, thus reducing the $NA_{eff}$ (*refracted rays in red*). (*bottom*) Imaging a sample in glycerol ($n = 1.47$) with the same objective leads to a more tightly focused image (*refracted rays in green*). If the objective is corrected for spherical aberrations introduced by aqueous samples, the opposite will be true – the PSF will be smaller for aqueous samples and a sample in glycerol will be poorly focused and have a low $NA_{eff}$.

DOI: https://doi.org/10.7554/eLife.28716.040

The influence of refractive index mismatch/spherical aberration on p-PALM measurements became apparent in control experiments on freely diffusing mEos3 in glycerol solution. Under our experimental conditions, translational motion was too fast to obtain p-PALM data on freely diffusing mEos3 in normal buffer. However, in 92% glycerol, the translational motion of freely diffusing mEos3 was significantly reduced and p-PALM experiments yielded $<p>_{lin} = 0.47–0.50$ (*Supplementary file 1*; *Figure 3—figure supplement 1*). Notably, the estimated $D_r$ of $\sim 3 \times 10^4$ rad$^2$/s (Stokes-Einstein-Debye relation; [*Loman et al., 2010*]) in this

high viscosity solution ($\eta \approx$ 380 cP; [**Lide, 1998**]) is in rotational mobility Regime III, indicating that $\theta_{eff} < 45°$ ($NA_{eff} < 0.94$). Thus, the $NA_{eff}$ was lower when using a 92% glycerol solution ($n = 1.46$; $\theta_{eff} < 45°$) than an aqueous sample ($n = 1.33$, $\theta_{eff} \approx$ 50–60°). This was surprising since spherical aberration should be worse for an aqueous sample than a glycerol sample when using an oil immersion objective due to a larger refractive index mismatch (**Appendix 1—figure 5**). Additional aberrations may have been introduced by the dual-polarization imaging system (see **Materials and methods**).

The range of $<p>_{lin}$ for mEos3 within the NPC was ~0.41–0.46 for the various conditions tested (**Supplementary file 1**). This range is higher than that in Region III of **Figure 2A** (~0.39). Thresholding, background noise, and ellipticity were not able to explain these differences (**Figure 2—figure supplements 4,7**), but a reduced NA can. Our linear excitation data (**Supplementary file 1**) are consistent with an $NA_{eff} \approx$ 1.02–1.15 in water ($\theta_{eff} \approx$ 50–60°) (**Appendix 1—figure 4**), which is substantially below the nominal value of the NA = 1.46 ($\theta_{obj}$ = 74.1°). The reduced NA is likely a consequence of spherical aberration with a minor contribution from scattering in the sample.

Most rotational walk simulations reported in this paper assume $\theta_{obj}$ = 74.1°, which corresponds to a non-scattering sample medium and aberration-free imaging with a sample index of refraction matching the immersion oil. These simulations are valid for determining the role of various parameters. However, since simulations with $\theta_{eff}$ = 50–60° more closely model the experimental conditions on permeabilized cells, they were used for more precise interpretation of the results (see Mixed populations section in main text). Importantly, our major conclusions are not affected by a moderate uncertainty in the $NA_{eff}$.

## Sensitivity to binding interactions

The sensitivity of the p-PALM approach to binding interactions became apparent when mEos2 (**McKinney et al., 2009**) was used as a probe. We observed significantly reduced rotational mobility for the probes on mEos2-Nup98 relative to mEos3-Nup98 (**Appendix 1—figure 6**), consistent with the finding that mEos2 forms dimers (**Hoi et al., 2010**; **McKinney et al., 2009**). Dimerization of mEos2 was likely promoted by the high local concentration of such probes within the FG-network. Since the mEos3 probe is a true monomer (**Zhang et al., 2012**), the reduced rotational mobilities reported here under certain conditions requires an alternate interpretation. As we have argued in this paper, molecular crowding is a reasonable explanation for the low rotational mobilities observed for probes in the FG-network of NPCs under some conditions. Nonetheless, the mEos2 results indicate that p-PALM can be used to directly monitor the binding of fluorescent molecule to a binding partner.

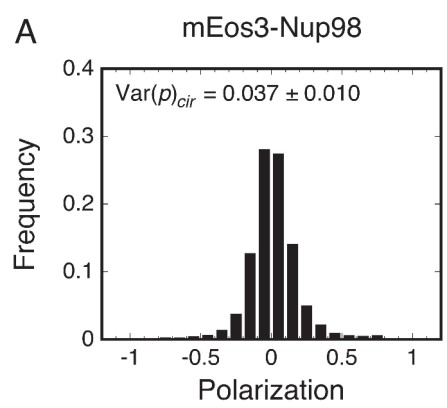 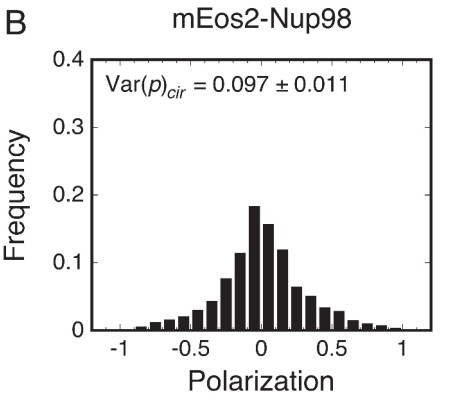

**Appendix 1—figure 6.** Reduced rotational mobility of the mEos2 probe. p-PALM measurements were made on mEos3 (**A**) and mEos2 (**B**) attached to the N-terminus of Nup98. The rotational mobility of mEos2 was significantly lower, likely due to the tendency of mEos2 to dimerize (**Hoi et al., 2010**; **McKinney et al., 2009**).

DOI: https://doi.org/10.7554/eLife.28716.041

## Mixed populations

Since the p-PALM parameters $<p>$ and Var($p$) are averages over the entire population within the dataset, the rotational mobility behavior of individual sub-populations in more complex samples may not be accurately reflected in these values. In the main text, one example of a mixed population sample (containing molecules with at least two distinct rotational mobilities) was discussed (*Figure 6*). Here, we further discuss this example and identify additional datasets containing mixed populations.

### Higher than expected Var($p$)$_{cir}$ values for tip-labeled FG-polypeptides

High mobility was expected for probes at the end of long FG-polypeptides, as these were presumed to be positioned at or near the periphery of the FG-network, and thus, in a relatively open environment (sparse polypeptide distribution). Since $D_r$ for mEos3 free in solution is $\sim 10^7$ rad$^2$/s (calculated for a sphere [*Loman et al., 2010*]), a high rotational mobility should yield Var($p$)$_{cir}$ = $\sim 0.01$–$0.02$ ($D_r > \sim 10^3$ rad$^2$/s; Regimes II-III of *Figure 2B*). Unexpectedly, the Var($p$)$_{cir}$ for tip-labeled FG-polypeptides under wildtype conditions was $\sim 0.04$–$0.06$ (*Supplementary file 1*). While the higher than expected Var($p$)$_{cir}$ values could be a reflection of an unexpectedly low rotational mobility (Regime IV), we explored whether there might be an alternate explanation. We first considered whether the asymptote at high $D_r$ values in *Figure 2B* is too low due to an error in the assumed threshold, camera noise, photons collected, the NA$_{eff}$, or the ellipticity. However, reasonable values for these parameters are unable to increase the high $D_r$ asymptote to reach the experimental values (*Figure 2—figure supplements 4,5,7*, and *Appendix 1—figure 4*).

Notably, the broader 'wings' in the polarization histograms for Pom121-mEos3 obtained under wildtype and conditions (*Figure 3*) were not obtained for the simulations on homogeneous populations (*Figure 2*). These broad 'wings' are consistent with a small sub-population of rotationally constrained molecules (low $D_r$ values) that yield high $|p|$ values. Thus, we explored whether a simulated mixture of probes with $D_r$ values $< \sim 10^3$ rad$^2$/s and $> \sim 10^3$ rad$^2$/s could simultaneously yield a Var($p$)$_{cir}$, polarization histogram, and photon scatterplot consistent with the data. This was indeed possible, as is demonstrated by the results of such an analysis (*Figure 6*). We therefore conclude that the higher than expected Var($p$)$_{cir}$ value for Pom121-mEos3 under wildtype conditions arises from sub-populations of probe molecules with distinct rotational mobilities, which are likely generated by at least two distinct environments. We therefore anticipated that the higher than expected Var($p$)$_{cir}$ values for other tip-labeled FG-polypeptides could be similarly explained.

### Rotational mobility near the NPC scaffold

We next examined whether probes positioned near the NPC scaffold also exhibited multiple rotational mobilities. The significantly shorter FG-polypeptide segment between the probe and the anchor domain prevents a wide spatial distribution of the probe, which could potentially confine the probe to a more homogeneous environment. Alternately, multiple anchoring sites or different environments in different NPCs could result in distinct rotational mobilities.

When attached to two different proteins, mEos3 probes attached near the NPC scaffold exhibited multiple distinct rotational mobilities. We first examined the rotational mobility of the probe on mEos3-$^{700mid}$Nup98, which was attached 12 residues away from the APD anchor domain (*Figure 5A*). The photon scatterplot for mEos3-$^{700mid}$Nup98 is consistent with a mixed population hypothesis (*Figure 6—figure supplement 1C*). As a control, the photon scatterplot of mEos3-Nup98 is also consistent with a mixed population hypothesis, although the percentage of the low rotational mobility sub-population appears significantly lower (*Figure 6—figure supplement 1A*). We also examined the rotational mobility of mEos3 attached to the N-terminus of Pom121, which is near the membrane anchor domain (*Figure 4*), and found that the photon scatterplot is consistent with a mixed population hypothesis, similar to the C-terminally labeled Pom121 (*Figure 6—figure supplement 2*). These data indicate the probes attached near the NPC scaffold on Nup98 and Pom121

experienced multiple environments, though it remains unclear whether this was a consequence of local heterogeneity near the anchoring sites within a single NPC, or whether the FG-networks of different NPCs had different compositions and/or structures (e.g., due to different NTR occupancies).

Since the probe was in the middle of an FG-polypeptide in the mEos3-$^{700mid}$Nup98 experiments, its rotational mobility behavior could potentially have been very different from that of tip-labeled mutants due to the two attachment points (**Appendix 1—figure 1A**). Therefore, it was important to examine the assumptions that the rotational anisotropy was low and that $\gamma \approx$ ~60°. Rotational random walk simulations where the $D_x/D_{yz}$ and $D_y/D_{xz}$ ratios were varied (**Appendix 1—figure 2**) revealed that values of $\gamma$ far from the magic angle of 54.7° (**Axelrod, 1989**) have a substantial effect on p-PALM measurements under highly anisotropic conditions. Allowing $\gamma$ to vary, we found that $\gamma =$ ~35–40° or ~70–75° at a high $D_z/D_{xy}$ ratio (**Appendix 1—figure 2**) yields Var(p)$_{cir}$ values consistent with that observed for mEos3-$^{700mid}$Nup98 (**Figure 5B**). However, the experimental polarization histograms are inconsistent with these parameters (**Figure 6—figure supplement 4**). Therefore, we conclude that the relatively high Var(p)$_{cir}$ value for the probe on mEos3-$^{700mid}$Nup98 is indeed a consequence of probe sub-populations with distinct $D_r$ values, and not due to highly anisotropic rotational mobility behavior.

## Increasing Var(p)$_{cir}$ by increasing the low mobility fraction in mixed populations.

The increased Var(p)$_{cir}$ for the mEos3 probe on mEos3-$^{700mid}$Nup98 relative to mEos3-Nup98 (0.12 vs 0.037; **Supplementary file 1**) coincided with an increase in the fraction of the sub-population with lower rotational mobility (**Figure 6—figure supplement 1**). We therefore explored whether other increases in Var(p)$_{cir}$ could be explained by an increase in the fraction of a lower mobility sub-population for a variety of other conditions. Photon scatterplots for the Nup98 middle mutants support the hypothesis that the observed increase in Var(p)$_{cir}$ as the probe was moved toward the APD anchor domain was indeed a consequence of an increase in the fraction of a lower mobility sub-population under both wildtype and +Imp β1 conditions (**Figure 6—figure supplement 5**). The substantial increases in Var(p)$_{cir}$ by WGA also are consistent with an increase in the low mobility fraction, though the high mobility fraction also appears to have reduced mobility (**Figure 6—figure supplements 6–8**). In summary, we consider it likely that most, if not all, of the high Var(p)$_{cir}$ values arose from mixed populations, one sub-population of which had a relatively low rotational mobility ($<10^3$ rad$^2$/s).

## Appendix 2

DOI: https://doi.org/10.7554/eLife.28716.042

# Analytical solutions for the Var(p)$_{cir}$ dependence on rotational diffusion

## Definition of *p*

The <p> and Var(p) values from rotational random walk simulations for a variety of conditions were obtained by running the simulations many times for individual molecules and averaging the results. In order to augment and verify the simulations, and guide the choice of the functional form of fits to the simulation data (**Equations 25 and 27**), we performed analytical calculations for a few special situations under circularly polarized illumination. For simplicity, we have dropped the 'cir' subscripts on <p> and Var(p) in this appendix.

Spherical coordinates were defined as:

$$x = \cos\theta \sin\varphi$$
$$y = \sin\theta \sin\varphi$$
$$z = \cos\varphi$$

Using **Equation 7** (**Materials and methods**) and including the excitation probability, the intensity collected from *m* dipole orientations under high photon flux is:

$$I_p = i_{tot} \sum_{k=1}^{m} (K_1 x_k^2 + K_2 y_k^2 + K_3 z_k^2)(\sin^2 \varphi_k)(\cos^2 \theta_k + (1/\varepsilon)\sin^2 \theta_k) \tag{28}$$

where ε is the ellipticity, as defined earlier (**Materials and methods**). For all of the derivations in this appendix, we have assumed circular excitation (ε = 1) and that the dipole does not rotate between excitation and emission ($D_r < \sim 10^6$ rad$^2$/s). Consequently,

$$I_p = i_{tot} \sum_{k=1}^{m} (K_1 x_k^2 + K_2 y_k^2 + K_3 z_k^2)(\sin^2 \varphi_k) \tag{29a}$$

$$I_p = i_{tot} \sum_{k=1}^{m} (K_1 \cos^2 \theta_k \sin^4 \varphi_k + K_2 \sin^2 \theta_k \sin^4 \varphi_k + K_3 \cos^2 \varphi_k \sin^2 \varphi_k) \tag{29b}$$

and similarly,

$$I_s = i_{tot} \sum_{k=1}^{m} (K_2 x_k^2 + K_1 y_k^2 + K_3 z_k^2)(\sin^2 \varphi_k) \tag{30a}$$

$$I_s = i_{tot} \sum_{k=1}^{m} (K_2 \cos^2 \theta_k \sin^4 \varphi_k + K_1 \sin^2 \theta_k \sin^4 \varphi_k + K_3 \cos^2 \varphi_k \sin^2 \varphi_k) \tag{30b}$$

Substituting the expressions above into the definition of polarization in **Equation 3** (**Materials and methods**) and assuming that *g* = 1, the polarization *p* measured for one molecule in elapsed time *t* for *m* dipole orientations (analogous to the rotational walk steps) is given by:

$$p = \frac{I_p - I_s}{I_p + I_s} = \frac{\sum_{k=1}^{m}(K_1 - K_2)\sin^4 \varphi_k \cos 2\theta_k}{\sum_{k=1}^{m}\left[(K_1 + K_2)\sin^4 \varphi_k + 2K_3 \sin^2 \varphi_k \cos^2 \varphi_k\right]} = \frac{A \sum_{k=1}^{m}\sin^4 \varphi_k \cos 2\theta_k}{\sum_{k=1}^{m}\left[(\sin^4 \varphi_k + B \sin^2 \varphi_k \cos^2 \varphi_k\right]} \tag{31}$$

where $A = (K_1 - K_2)/(K_1 + K_2)$ and $B = 2K_3/(K_1 + K_2)$. As discussed earlier, $K_1$, $K_2$, and $K_3$ depend on the NA of the microscope objective (**Equations 9–11**). For $\theta_{obj}$ = 74.1°, A = 0.939 and B = 0.765.

## Case 1: $D_r t = 0$

The upper limit of Var($p$) is attained when the particles are rotationally immobile, which corresponds to either short collection times or arrested rotational diffusion. This is formally defined as the limit where $D_r t \to 0$. Under these conditions, $\theta$ and $\varphi$ are constant for a given individual particle, and thus, *Equation 31* reduces to:

$$p = \frac{A\sin^2\varphi}{\sin^2\varphi + B\cos^2\varphi}\cos 2\theta = H(A,B,\varphi)\cos 2\theta \tag{32}$$

where the $\varphi$-dependence has been subsumed into the function $H$, which depends on $A$, $B$, and $\varphi$. Assuming an isotropic distribution of dipole orientations, the second moment of the polarization distribution, $<p^2>$, for a very large collection of such particles is obtained by averaging $p$ over all angles. Recalling that $<p> = 0$ and Var($p$) = $<p^2>$ for circular excitation (see **Materials and methods**, *Figure 2*, and *Equations 46 and 65*), we get:

$$\text{Var}(p)_{D_r t \to 0} = <p^2>_{D_r t \to 0} = \frac{1}{4\pi}\int_0^\pi\int_0^{2\pi}\left(\frac{A\sin^2\varphi\cos 2\theta}{\sin^2\varphi + B\cos^2\varphi}\right)^2\sin\varphi\, d\theta d\varphi \tag{33}$$

The integration over $\theta$ equates to $\pi$, which leads to:

$$\text{Var}(p)_{D_r t \to 0} = \frac{1}{4}\int_0^\pi\left(\frac{A\sin^2\varphi}{\sin^2\varphi + B\cos^2\varphi}\right)^2\sin\varphi\, d\varphi \tag{34}$$

As $A$ and $B$ are constant for a given experiment, numerically integrating using values for $\theta_{obj} = 74.1°$ yields:

$$\text{Var}(p)_{D_r t \to 0} = \frac{1}{4}(1.0117) = 0.2529 \tag{35}$$

Thus, for $D_r t \to 0$ (immobile) and $\alpha = 0$ (infinite photons), the value of $\beta$ in *Equation 25* is $\beta_0 = 1/0.2529 = 3.954$.

## Case 2: $\varphi$ = Constant (rotation on a circle)

### Definitions

In some cases, a dipole may be restricted to rotate around a single rotational axis. For example, a membrane protein could rotate around an axis normal to the membrane – the extent of lateral diffusion is irrelevant, as long as the movement is two-dimensional (no membrane curvature). The orientation of the dipole relative to the membrane normal is not restricted, but remains constant ($\varphi$ = constant). This situation corresponds to rotation around a circle, or $D_z > 0$ and $D_x = D_y = 0$. Under these conditions, *Equation 31* reduces to:

$$p = \frac{H(A,B,\varphi)}{m}\sum_{k=1}^m\cos 2\theta_k \tag{36}$$

where the factor $m$ in the denominator normalizes the result based on the number of different orientations of the dipole. This is analogous to the integral along the dipole trajectory,

$$p(t,\theta_0) = \frac{H(A,B,\varphi)}{t}\int_0^t dt'\cos[2\theta(t',\theta_0)] \tag{37}$$

where $\theta(t, \theta_0)$ is the value of $\theta$ at time $t$ given a starting angle of $\theta_0$. Note that $p(t, \theta_0)$ is a random variable that depends on the particle's entire angular trajectory. To evaluate its distribution, we need to know the time evolution of $\theta$.

Assuming that the dipole motion is an isotropic rotational random walk, the probability for a dipole starting at angle $\theta_0$ to be at an angle $\theta$ at time $t$ is given by:

$$q(\theta, t, \theta_0) = \sum_{ngt=gt-\infty}^{ngt=gt\infty} \frac{1}{\sqrt{4\pi D_z t}} \exp\left[-(\theta - \theta_0 + 2\pi n)^2/4D_z t\right] \tag{38}$$

where $n$ is an integer. This wrapped normal distribution is commonly approximated by a von Mises distribution (**Watson, 1982**):

$$q(\theta, t, \theta_0) = \frac{1}{2\pi I_0\left(\frac{1}{2D_z t}\right)} \exp\left[\frac{\cos(\theta - \theta_0)}{2D_z t}\right] \tag{39}$$

where the normalization factor

$$I_0\left(\frac{1}{2D_z t}\right) = \frac{1}{2\pi}\int_{-\pi}^{\pi} d\theta \exp\left[\frac{\cos(\theta - \theta_0)}{2D_z t}\right] \tag{40}$$

is the modified Bessel function of order 0. Denoting $Q(p, t, \theta_0)$ as the probability that the value of $p(t, \theta_0)$ at time $t$ is $p$, the mean value of $p(t, \theta_0)$ at time $t$ for a given $\theta_0$ is:

$$<p>(t, \theta_0) = \int_{-1}^{1} p\,dp \cdot Q(p, t, \theta_0) \tag{41}$$

Assuming a uniform (random) initial angle distribution, the probability to obtain some value $p$ is:

$$Q(p, t) = \frac{1}{2\pi}\int_{-\pi}^{\pi} d\theta_0 Q(p, t, \theta_0) \tag{42}$$

where $1/2\pi$ normalizes the integration over $\theta_0$. The average $p$ at time $t$ is therefore:

$$\begin{aligned}
<p>(t) &= \frac{1}{2\pi}\int_{-1}^{1} p\,dp \cdot Q(p, t) \\
&= \frac{1}{2\pi}\int_{-1}^{1} p\,dp \int_{-\pi}^{\pi} d\theta_0 Q(p, t, \theta_0) \\
&= \frac{1}{2\pi}\int_{-\pi}^{\pi} d\theta_0 \int_{-1}^{1} p\,dp \cdot Q(p, t, \theta_0) \\
<p>(t) &= \frac{1}{2\pi}\int_{-\pi}^{\pi} d\theta_0 <p>(t, \theta_0)
\end{aligned} \tag{43}$$

Similarly,

$$<p^2>(t) = \frac{1}{2\pi}\int_{-\pi}^{\pi} d\theta_0 <p^2>(t, \theta_0) \tag{44}$$

## Mean (<p>)

With the above definitions, we first obtain <p>. Using **Equation 39** to evaluate **Equation 37**:

$$\begin{aligned}
<p>(t, \theta_0) &= \left\langle\left(\frac{H}{t}\int_0^t dt' \cos[2\theta(t', \theta_0)]\right)\right\rangle \\
&= \frac{H}{t}\int_0^t dt' \int_{-\pi}^{\pi} d\theta \langle\cos[2\theta(t', \theta_0)]\rangle \\
&= \frac{H}{t}\int_0^t dt' \int_{-\pi}^{\pi} d\theta \cos[2\theta(t', \theta_0)] \cdot q(\theta, t', \theta_0) \\
&= \frac{H}{t}\int_0^t dt' \int_{-\pi}^{\pi} d\theta \frac{\cos(2\theta)}{2\pi I_0\left(\frac{1}{2D_z t'}\right)} \exp\left[\frac{\cos(\theta - \theta_0)}{2D_z t'}\right]
\end{aligned}$$

Making the substitution $u = \theta - \theta_0$:

$$<p>(t,\theta_0) = \frac{H}{t}\int_0^t dt'\int_{-\pi}^{\pi} du \frac{\cos(2(u+\theta_0))}{2\pi I_0\left(\frac{1}{2D_z t'}\right)}\exp\left[\frac{\cos(u)}{2D_z t'}\right]$$

$$= \frac{H}{t}\int_0^t dt'\int_{-\pi}^{\pi} du \frac{(\cos 2u \cos 2\theta_0 - \sin 2u \sin 2\theta_0)}{2\pi I_0\left(\frac{1}{2D_z t'}\right)}\exp\left[\frac{\cos(u)}{2D_z t'}\right]$$

$$= \frac{H\cos(2\theta_0)}{t}\int_0^t dt'\int_{-\pi}^{\pi} du \frac{\cos 2u}{2\pi I_0\left(\frac{1}{2D_z t'}\right)}\exp\left[\frac{\cos(u)}{2D_z t'}\right]$$

or

$$<p>(t,\theta_0) = \frac{H\cos(2\theta_0)}{t}\int_0^t dt' \frac{I_2\left(\frac{1}{2D_z t'}\right)}{I_0\left(\frac{1}{2D_z t'}\right)} \tag{45}$$

where $I_2(1/(2D_z t))$ is the 'modified Bessel function of order 2'. Using this to evaluate **Equation 43**, we find that:

$$<p>(t) = \frac{H}{2\pi t}\left[\int_{-\pi}^{\pi} d\theta_0 \cos(2\theta_0)\right]\int_0^t dt' \frac{I_2\left(\frac{1}{2D_z t'}\right)}{I_0\left(\frac{1}{2D_z t'}\right)} = 0 \tag{46}$$

for all $t$. This simply states that the average polarization for circularly polarized excitation is always 0 (due to the first integral over $\theta_0$), which is clearly indicated by the histograms in **Figure 2D**.

As the integral in **Equation 45** will become necessary for calculating $<p^2>(t)$, we describe some of its properties. The integral cannot be calculated analytically. However, it can be evaluated numerically. Making the substitution $x = D_z t'$, **Equation 45** can be rewritten as:

$$<p>(t,\theta_0) = \frac{H\cos(2\theta_0)}{D_z t}\int_0^{D_z t} dx \frac{I_2\left(\frac{1}{2x}\right)}{I_0\left(\frac{1}{2x}\right)} = H\cos(2\theta_0)\cdot M_1(D_z t) \tag{47}$$

defining the new function:

$$M_1(D_z t) = \frac{1}{D_z t}\int_0^{D_z t} dx \frac{I_2\left(\frac{1}{2x}\right)}{I_0\left(\frac{1}{2x}\right)} \tag{48}$$

which has the following properties:

For small $t$ (and thus, small $x$), $\frac{I_2\left(\frac{1}{2x}\right)}{I_0\left(\frac{1}{2x}\right)} \to 1$ so that $M_1(D_z t) \to \frac{1}{D_z t}(D_z t) = 1$

For large $t$, $\frac{I_2\left(\frac{1}{2x}\right)}{I_0\left(\frac{1}{2x}\right)} \to 0$, the integral converges to 0.2541, and $M_1(D_z t) \to 0$

Overall, $M_1$ is well approximated by:

$$M_1(D_z t) = \frac{0.2541}{0.2541 + D_z t} = \frac{1}{1 + 3.935 D_z t} \tag{49}$$

and more precisely with the higher order Pade-like approximation (**Appendix 2—figure 1**):

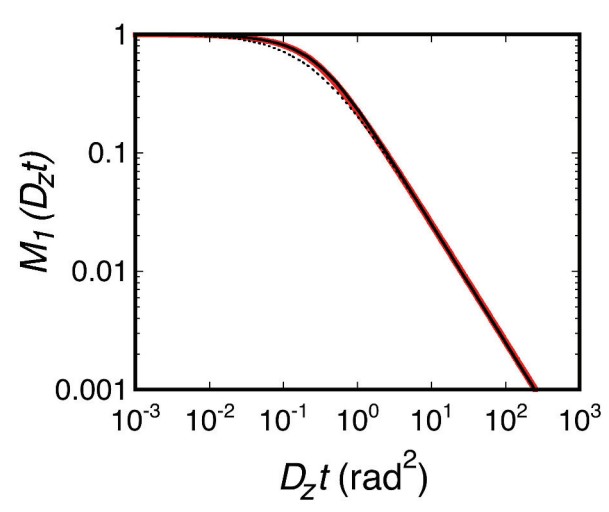

**Appendix 2—figure 1.** $M_1(D_z t)$. **Equation 48** was evaluated by numerical integration (red). The dashed black curve is **Equation 49**, and the solid black curve is **Equation 50**.
DOI: https://doi.org/10.7554/eLife.28716.043

$$M_1(D_z t) = \frac{0.2541 + 0.5 D_z t}{0.2541 + D_z t + 2(D_z t)^2}$$
$$= \frac{1 + 1.967 D_z t}{1 + 3.934 D_z t + 7.868(D_z t)^2} \tag{50}$$

## Second moment ($<p^2>$)

A similar approach was used to obtain $<p^2>$. From **Equation 37**:

$$
\begin{aligned}
<p^2>(t,\theta_0) &= \left\langle \left( \frac{H}{t} \int_0^t dt' \cos[2\theta(t',\theta_0)] \right)^2 \right\rangle \\
&= \left\langle \left[ \frac{H}{t} \int_0^t dt' \cos[2\theta(t',\theta_0)] \right] \left[ \frac{H}{t} \int_0^t dt'' \cos[2\theta(t'',\theta_0)] \right] \right\rangle \\
&= \frac{H^2}{t^2} \int_0^t dt' \int_0^t dt'' <\cos[2\theta(t',\theta_0)] \cdot \cos[2\theta(t'',\theta_0)]>
\end{aligned}
\tag{51}
$$

which can be rewritten as:

$$
\begin{aligned}
<p^2>(t,\theta_0) = {}& \frac{H^2}{t^2} \int_0^t dt' \int_0^{t'} dt'' \int_{-\pi}^{\pi} d\theta' \int_{-\pi}^{\pi} d\theta'' \cos(2\theta') \cos(2\theta'') q(\theta'',t'',\theta_0) q(\theta',t'-t'',\theta'') \\
& + \frac{H^2}{t^2} \int_0^t dt' \int_{t'}^t dt'' \int_{-\pi}^{\pi} d\theta' \int_{-\pi}^{\pi} d\theta'' \cos(2\theta') \cos(2\theta'') q(\theta',t',\theta_0) q(\theta'',t''-t',\theta')
\end{aligned}
\tag{52}
$$

The second integral in **Equation 52** can be rewritten as:

$$
\begin{aligned}
& \frac{H^2}{t^2} \int_0^t dt'' \int_0^{t''} dt' \int_{-\pi}^{\pi} d\theta' \int_{-\pi}^{\pi} d\theta'' \cos(2\theta') \cos(2\theta'') q(\theta',t',\theta_0) q(\theta'',t''-t',\theta') \\
& = \frac{H^2}{t^2} \int_0^t dt' \int_0^{t'} dt'' \int_{-\pi}^{\pi} d\theta' \int_{-\pi}^{\pi} d\theta'' \cos(2\theta') \cos(2\theta'') q(\theta',t'',\theta_0) q(\theta'',t'-t'',\theta') \\
& = \frac{H^2}{t^2} \int_0^t dt' \int_0^{t'} dt'' \int_{-\pi}^{\pi} d\theta' \int_{-\pi}^{\pi} d\theta'' \cos(2\theta') \cos(2\theta'') q(\theta'',t'',\theta_0) q(\theta',t'-t'',\theta'')
\end{aligned}
$$

Which is the same as the first integral in **Equation 52**. Thus,

$$<p^2>(t,\theta_0) = \frac{2H^2}{t^2} \int_0^t dt' \int_0^{t'} dt'' \int_{-\pi}^{\pi} d\theta' \int_{-\pi}^{\pi} d\theta'' \cos(2\theta')\cos(2\theta'')q(\theta'',t'',\theta_0)q(\theta',t'-t'',\theta'') \quad (53)$$

and

$$
\begin{aligned}
<p^2>(t) &= \frac{H^2}{\pi t^2} \int_{-\pi}^{\pi} d\theta_0 \int_0^t dt' \int_0^{t'} dt'' \int_{-\pi}^{\pi} d\theta' \int_{-\pi}^{\pi} d\theta'' \cos(2\theta')\cos(2\theta'')q(\theta'',t'',\theta_0)q(\theta',t'-t'',\theta'') \\
&= \frac{H^2}{\pi t^2} \int_0^t dt' \int_0^{t'} dt'' \int_{-\pi}^{\pi} d\theta' \int_{-\pi}^{\pi} d\theta'' \cos(2\theta')\cos(2\theta'')q(\theta',t'-t'',\theta'') \int_{-\pi}^{\pi} \frac{d\theta_0}{2\pi I_0\left(\frac{1}{2D_z t''}\right)} \exp\left[\frac{\cos(\theta''-\theta_0)}{2D_z t''}\right] \\
&= \frac{H^2}{\pi t^2} \int_0^t dt' \int_0^{t'} dt'' \int_{-\pi}^{\pi} d\theta' \int_{-\pi}^{\pi} d\theta'' \cos(2\theta')\cos(2\theta'')q(\theta',t'-t'',\theta'') \frac{2\pi I_0\left(\frac{1}{2D_z t''}\right)}{2\pi I_0\left(\frac{1}{2D_z t''}\right)} \\
&= \frac{H^2}{\pi t^2} \int_0^t dt' \int_0^{t'} dt'' \int_{-\pi}^{\pi} d\theta' \cos(2\theta') \int_{-\pi}^{\pi} d\theta'' \cos(2\theta'') \frac{\exp\left[\frac{\cos(\theta'-\theta'')}{2D_z(t'-t'')}\right]}{2\pi I_0\left(\frac{1}{2D_z(t'-t'')}\right)}
\end{aligned}
$$

Making the substitution $u = \theta' - \theta''$:

$$
\begin{aligned}
&= \frac{H^2}{\pi t^2} \int_0^t dt' \int_0^{t'} dt'' \int_{-\pi}^{\pi} d\theta' \cos(2\theta') \int_{-\pi}^{\pi} du \cos(2\theta' - 2u) \frac{\exp\left[\frac{\cos(u)}{2D_z(t'-t'')}\right]}{2\pi I_0\left(\frac{1}{2D_z(t'-t'')}\right)} \\
&= \frac{H^2}{\pi t^2} \int_0^t dt' \int_0^{t'} dt'' \frac{I_2\left(\frac{1}{2D_z(t'-t'')}\right)}{I_0\left(\frac{1}{2D_z(t'-t'')}\right)} \int_{-\pi}^{\pi} d\theta' \cos^2(2\theta') \\
&= \frac{H^2}{t^2} \int_0^t dt' \int_0^{t'} dt'' \frac{I_2\left(\frac{1}{2D_z(t'-t'')}\right)}{I_0\left(\frac{1}{2D_z(t'-t'')}\right)}
\end{aligned}
$$

or

$$<p^2>(t) = \frac{H^2}{(D_z t)^2} \int_0^{D_z t} y\,dy \cdot M_1(y) = H^2 \cdot M_2(D_z t) \quad (54)$$

where:

$$M_2(D_z t) = \frac{1}{(D_z t)^2} \int_0^{D_z t} y\,dy \cdot M_1(y) \quad (55)$$

Approximating $M_1(y)$ as in **Equation 50** and numerically integrating to obtain $M_2(D_z t)$, it was determined that $<p^2>$ is well-approximated by (**Appendix 2—figure 2**):

$$\mathrm{Var}(p)\,(t) = <p^2>(t) = \frac{[H(A,B,\varphi)]^2}{1.95 + 3.68 D_z t} \quad (56)$$

where Var(p) = $<p^2>$ since $<p>$ = 0 for all $D_r$ values (**Equation 46**). Though constant under the constraint of Case 2, $H^2$ depends on $\varphi$, and ranges from 0 to ~0.94, assuming that $\theta_{obj}$ = 74.1°.

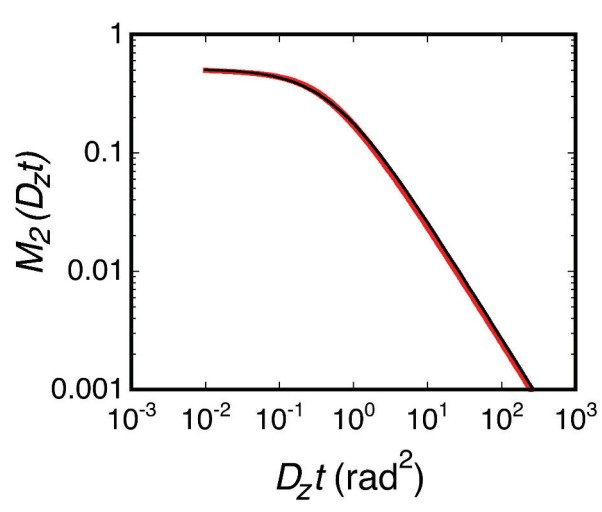

**Appendix 2—figure 2.** $M_2(D_z t)$. *Equation 55* was evaluated by numerical integration (red). The solid black curve is the best fit, given in *Equation 56* as $M_2(D_z t) = <p^2>(t)/H^2$ (see *Equation 54*).

DOI: https://doi.org/10.7554/eLife.28716.044

## Case 3: General 3D case for a perfect objective (*B* = 1)

### Definitions

The general 3D case is not treatable analytically in an exact fashion since the two summations in *Equation 31* are coupled to the path of the same particle and therefore are statistically correlated. However, assuming isotropic excitation (which eliminates the $\sin^2\varphi$ terms in *Equations 29a and 30a*) and the specific case where $B = 1$ (which corresponds to $\theta_{obj} = 180°$, i.e., all photons collected), $p$ (defined by *Equation 31*) reduces to:

$$p = \frac{A}{m}\sum_{k=1}^{m}\sin^2\varphi_k \cos 2\theta_k \tag{57}$$

We derive here the dependence of $<p^2>$ on $D_r t$ for the three-dimensional (3D) case where $D_r = D_x = D_y = D_z$ under the constraints of *Equation 57*. A 3D rotational random walk can be approximated by the Fisher-von Mises distribution (*Watson, 1982*), in which the probability density for a particle starting at angles $(\theta_0, \varphi_0)$ to be at $(\theta, \varphi)$ at time $t$ is given by:

$$q(\theta,\varphi,t,\theta_0,\varphi_0) = \frac{1}{2\pi F_0\left(\frac{1}{2D_r t},\theta_0,\varphi_0\right)}\exp\left[\frac{\cos[\omega(\theta,\varphi,\theta_0,\varphi_0)]}{2D_r t}\right] \tag{58}$$

where $\omega(\theta,\varphi,\theta_0,\varphi_0)$ is the angle between the initial and current position vectors on the unit sphere, and

$$F_0\left(\frac{1}{2D_r t},\theta_0,\varphi_0\right) = \frac{1}{2\pi}\int_0^\pi d\varphi \sin\varphi \int_{-\pi}^\pi d\theta \exp\left[\frac{\cos[\omega(\theta,\varphi,\theta_0,\varphi_0)]}{2D_r t}\right] \tag{59}$$

is the normalization constant. From stereometry,

$$\omega(\theta,\varphi,\theta_0,\varphi_0) = \cos\varphi\cos\varphi_0 + \sin\varphi\sin\varphi_0\cos(\theta - \theta_0) \tag{60}$$

and thus:

$$2\pi F_0\left(\frac{1}{2D_r t}, \theta_0, \varphi_0\right) = \int_0^\pi d\varphi \sin\varphi \int_{-\pi}^\pi d\theta \exp\left[\frac{\cos\varphi \cos\varphi_0 + \sin\varphi \sin\varphi_0 \cos(\theta-\theta_0)}{2D_r t}\right]$$

$$F_0\left(\frac{1}{2D_r t}, \varphi_0\right) = \int_0^\pi d\varphi \sin\varphi \exp\left[\frac{\cos\varphi \cos\varphi_0}{2D_r t}\right] I_0\left[\frac{\sin\varphi \sin\varphi_0}{2D_r t}\right]$$

(61)

indicating that there is no dependence of $F_0$ on $\theta_0$.

## Mean ($<p>$)

With the above definitions, $<p>$ is calculated similarly to the 2D case (Case 2), but now including φ rotations. The measured polarization of a single molecule after integrating for time $t$ is given by:

$$p(t, \theta_0, \varphi_0) = \frac{A}{t}\int_0^t dt' \sin^2\varphi(t', \theta_0, \varphi_0) \cos[2\theta(t', \theta_0, \varphi_0)]$$

(62)

Therefore, analogous to the derivation of **Equation 45**:

$$<p>(t, \theta_0, \varphi_0) = \frac{A}{t}\int_0^t dt' <\sin^2\varphi(t', \theta_0, \varphi_0)\cos[2\theta(t', \theta_0, \varphi_0)]>$$

$$= \frac{A}{t}\int_0^t dt' \int_0^\pi \sin^2\varphi \sin\varphi d\varphi \int_{-\pi}^\pi d\theta \cos 2\theta \cdot q(\theta, \varphi, t', \theta_0, \varphi_0)$$

$$= \frac{A}{t}\int_0^t dt' \int_0^\pi \sin^3\varphi d\varphi \int_{-\pi}^\pi \frac{d\theta \cos(2\theta)}{2\pi F_0\left(\frac{1}{2D_r t'}, \varphi_0\right)} \exp\left[\frac{\cos\varphi \cos\varphi_0 + \sin\varphi \sin\varphi_0 \cos(\theta-\theta_0)}{2D_r t'}\right]$$

(63)

Making the substitution $u = \theta - \theta_0$:

$$<p>(t, \theta_0, \varphi_0) = \frac{A}{t}\int_0^t dt' \int_0^\pi \sin^3\varphi d\varphi \int_{-\pi}^\pi \frac{du \cos(2u+2\theta_0)}{2\pi F_0\left(\frac{1}{2D_r t'}, \varphi_0\right)} \exp\left[\frac{\cos\varphi \cos\varphi_0 + \sin\varphi \sin\varphi_0 \cos(u)}{2D_r t'}\right]$$

$$= \frac{A\cos 2\theta_0}{t}\int_0^t dt' \int_0^\pi \sin^3\varphi d\varphi \int_{-\pi}^\pi \frac{du \cos 2u}{2\pi F_0\left(\frac{1}{2D_r t'}, \varphi_0\right)} \exp\left[\frac{\cos\varphi \cos\varphi_0 + \sin\varphi \sin\varphi_0 \cos(u)}{2D_r t'}\right]$$

$$= \frac{A\cos 2\theta_0}{t}\int_0^t dt' \int_0^\pi \sin^3\varphi d\varphi \frac{\exp\left[\frac{\cos\varphi \cos\varphi_0}{2D_r t'}\right] 2\pi I_2\left(\frac{\sin\varphi \sin\varphi_0}{2D_r t'}\right)}{2\pi F_0\left(\frac{1}{2D_r t'}, \varphi_0\right)}$$

$$= \frac{A\cos 2\theta_0}{t}\int_0^t dt' \frac{F_2\left(\frac{1}{2D_r t'}, \varphi_0\right)}{F_0\left(\frac{1}{2D_r t'}, \varphi_0\right)}$$

Where we have defined

$$F_2\left(\frac{1}{2D_r t}, \varphi_0\right) = \int_0^\pi \sin^3\varphi d\varphi \exp\left[\frac{\cos\varphi \cos\varphi_0}{2D_r t}\right] I_2\left(\frac{\sin\varphi \sin\varphi_0}{2D_r t}\right)$$

(64)

Thus,

$$<p>(t, \theta_0, \varphi_0) = \frac{A\cos 2\theta_0}{D_r t}\int_0^{D_r t} dx \frac{F_2\left(\frac{1}{2x}, \varphi_0\right)}{F_0\left(\frac{1}{2x}, \varphi_0\right)}$$

(65)

As for **Equation 45**, when the expression in **Equation 65** is integrated over $\theta_0$, we obtain $<p>(t) = 0$, for all $t$, as expected.

## Second moment ($<p^2>$)

An approach similar to that used to evaluate **Equation 51** was used to obtain $<p^2>$:

$$
\begin{aligned}
<p^2>(t,\theta_0,\varphi_0) &= \left\langle \left(\frac{A}{t}\int_0^t dt' \sin^2\varphi(t',\theta_0,\varphi_0)\cos[2\theta(t',\theta_0,\varphi_0)]\right)^2 \right\rangle \\
&= \left\langle \left[\frac{A}{t}\int_0^t dt' \sin^2\varphi(t',\theta_0,\varphi_0)\cos[2\theta(t',\theta_0,\varphi_0)]\right] \right. \\
&\quad \left. \left[\frac{A}{t}\int_0^t dt'' \sin^2\varphi(t'',\theta_0,\varphi_0)\cos[2\theta(t'',\theta_0,\varphi_0)]\right]\right\rangle \\
&= \frac{A^2}{t^2}\int_0^t dt' \int_0^{t'} dt'' \langle \sin^2\varphi'(t',\theta_0,\varphi_0)\cos[2\theta(t',\theta_0,\varphi_0)]\cdot \\
&\quad \sin^2\varphi''(t'',\theta_0,\varphi_0)\cos[2\theta(t'',\theta_0,\varphi_0)]\rangle \\
&= \frac{2A^2}{t^2}\int_0^t dt' \int_0^{t'} dt'' \int_0^\pi \sin^3\varphi' d\varphi' \int_0^\pi \sin^3\varphi'' d\varphi'' \\
&\quad \int_{-\pi}^\pi d\theta'' \cos(2\theta'') q(\theta'',\varphi'',t'',\theta_0,\varphi_0) \\
&\quad \cdot \int_{-\pi}^\pi d\theta' \cos(2\theta') q(\theta',\varphi',t'-t'',\theta'',\varphi'')
\end{aligned}
\tag{66}
$$

and

$$
\begin{aligned}
<p^2>(t) &= \frac{2A^2}{4\pi t^2}\int_{-\pi}^\pi d\theta_0 \int_0^\pi \sin\varphi_0 d\varphi_0 \int_0^t dt' \int_0^{t'} dt'' \int_0^\pi \sin^3\varphi' d\varphi' \int_0^\pi \sin^3\varphi'' d\varphi'' \\
&\quad \cdot \int_{-\pi}^\pi d\theta'' \cos(2\theta'') q(\theta'',\varphi'',t'',\theta_0,\varphi_0) \int_{-\pi}^\pi d\theta' \cos(2\theta') q(\theta',\varphi',t'-t'',\theta'',\varphi'')
\end{aligned}
\tag{67}
$$

where $1/4\pi$ normalizes the integrations over $\theta_0$ and $\varphi_0$. Simplifying:

$$
\begin{aligned}
<p^2>(t) &= \frac{2A^2}{4\pi t^2}\int_0^t dt' \int_0^{t'} dt'' \int_0^\pi \sin^3\varphi' d\varphi' \int_0^\pi \sin^3\varphi'' d\varphi'' \int_{-\pi}^\pi d\theta'' \cos(2\theta'') \\
&\quad \cdot \int_{-\pi}^\pi d\theta' \cos(2\theta') q(\theta',\varphi',t'-t'',\theta'',\varphi'') \int_{-\pi}^\pi d\theta_0 \int_0^\pi \sin\varphi_0 d\varphi_0 q(\theta'',\varphi'',t'',\theta_0,\varphi_0)
\end{aligned}
\tag{68}
$$

The last two integrals in **Equation 68** equate to 1 because $(\theta'', \varphi'', t'', \theta_0, \varphi_0)$ is symmetric in $(\theta'', \varphi'')$ and $(\theta_0, \varphi_0)$:

$$
\begin{aligned}
&\int_{-\pi}^\pi d\theta_0 \int_0^\pi \sin\varphi_0 d\varphi_0 q(\theta'',\varphi'',t'',\theta_0,\varphi_0) \\
&= \int_{-\pi}^\pi d\theta'' \int_0^\pi \sin\varphi'' d\varphi'' q(\theta'',\varphi'',t'',\theta_0,\varphi_0) = 1
\end{aligned}
$$

Thus, changing the order of integration in **Equation 68**, we get:

$$
\begin{aligned}
<p^2>(t) &= \frac{A^2}{2\pi t^2}\int_0^t dt'' \int_{t''}^t dt' \int_0^\pi \sin^3\varphi' d\varphi' \int_0^\pi \sin^3\varphi'' d\varphi'' \int_{-\pi}^\pi d\theta'' \cos(2\theta'') \\
&\quad \cdot \int_{-\pi}^\pi d\theta' \cos(2\theta') q(\theta',\varphi',t'-t'',\theta'',\varphi'')
\end{aligned}
$$

Making the substitution $t' - t'' = z$,

$$
\begin{aligned}
<p^2>(t) &= \frac{A^2}{2\pi t^2}\int_0^t dt'' \int_0^{t-t''} dz \int_0^\pi \sin^3\varphi' d\varphi' \int_0^\pi \sin^3\varphi'' d\varphi'' \int_{-\pi}^\pi d\theta'' \cos(2\theta'') \\
&\quad \cdot \int_{-\pi}^\pi d\theta' \cos(2\theta') q(\theta',\varphi',z,\theta'',\varphi'')
\end{aligned}
$$

Recognizing from **Equations 63–65** that

$$
\int_0^{t-t''} dz \int_0^\pi \sin^3\varphi' d\varphi' \int_{-\pi}^\pi d\theta' \cos(2\theta') q(\theta',\varphi',z,\theta'',\varphi'') = \cos 2\theta'' \int_0^{t-t''} dz \frac{F_2\left(\frac{1}{2D_r z},\varphi''\right)}{F_0\left(\frac{1}{2D_r z},\varphi''\right)}
$$

we obtain

$$
<p^2>(t) = \frac{A^2}{2\pi t^2}\int_0^t dt'' \int_0^\pi \sin^3\varphi'' d\varphi'' \int_{-\pi}^\pi d\theta'' \cos^2(2\theta'') \int_0^{t-t''} dz \frac{F_2\left(\frac{1}{2D_r z},\varphi''\right)}{F_0\left(\frac{1}{2D_r z},\varphi''\right)}
$$

Making the variable substitutions $x = D_r(t - t'')$ and $y = D_r z$, we obtain

$$<p^2>(t) = \frac{A^2}{2(D_r t)^2} \int_0^\pi \sin^3 \varphi'' d\varphi'' \int_0^{D_r t} dx \int_0^x dy \frac{F_2\left(\frac{1}{2y}, \varphi''\right)}{F_0\left(\frac{1}{2y}, \varphi''\right)}$$

and finally,

$$<p^2>(t) = \frac{A^2}{2(D_r t)^2} \int_0^\pi \sin^3 \varphi'' d\varphi'' \int_0^{D_r t} x G(x, \varphi'') dx \tag{69}$$

where

$$G(x, \varphi'') = \frac{1}{x} \int_0^x dy \frac{F_2\left(\frac{1}{2y}, \varphi''\right)}{F_0\left(\frac{1}{2y}, \varphi''\right)} \tag{70}$$

To numerically evaluate **Equation 69**, we first examine two limits.

## $D_r t \to 0$

For small values of $D_r t$, $G$ can be approximated by:

$$\lim_{x \to 0} G(x, \varphi'') \approx \sin^2 \varphi'' \tag{71}$$

and therefore,

$$<p^2>(t)_{D_r t \to 0} \approx \frac{A^2}{2(D_r t)^2} \int_0^\pi \sin^3 \varphi'' \sin^2 \varphi'' d\varphi'' \int_0^{D_r t} x dx$$

$$<p^2>(t)_{D_r t \to 0} = \frac{A^2}{4} \int_0^\pi \sin^5 \varphi'' d\varphi'' \tag{72}$$

Integration yields:

$$<p^2>(t)_{D_r t \to 0} = \frac{A^2}{4}\left(\frac{16}{15}\right) = \frac{4A^2}{15} = \frac{1}{15} \tag{73}$$

since $A = 0.5$ under the constraint that $\theta_{obj} = 180°$.

## $D_r t \to \infty$

For large values of $D_r t$, $G$ can be approximated by:

$$G(x, \varphi'') \approx \frac{M(\varphi'')}{x} \quad \text{where} \quad M(\varphi'') = \int_0^\infty dy \frac{F_2\left(\frac{1}{2y}, \varphi''\right)}{F_0\left(\frac{1}{2y}, \varphi''\right)} \tag{74}$$

Numerical evaluation reveals that $M$ is well approximated by:

$$M(\varphi'') \approx \frac{\sin^2 \varphi''}{5.2} \tag{75}$$

Using **Equations 71, 74 and 75**, the function $G(x, \varphi'')$ can be well approximated with a Pade-like expression over the entire range of possible $D_r t$ values as:

$$G(x, \varphi'') \approx \frac{M(\varphi'')}{(1/5.2) + x} = \frac{M(\varphi'')}{0.192 + x} \tag{76}$$

Therefore,

$$<p^2>(t)_{D_r t \to \infty} \approx \frac{A^2}{2(D_r t)^2} \int_0^\pi \sin^3 \varphi'' d\varphi'' \int_0^{D_r t} \frac{M(\varphi'')x dx}{0.192 + x}$$

$$= \frac{A^2}{2(D_r t)^2} \int_0^\pi \sin^3 \varphi'' d\varphi'' M(\varphi'') \int_0^{D_r t} \left(1 - \frac{0.192}{0.192 + x}\right) dx \qquad (77)$$

$$= \frac{A^2}{2(D_r t)^2} \int_0^\pi \sin^3 \varphi'' d\varphi'' M(\varphi'')(D_r t - 0.192 \ln(0.192 + D_r t))$$

Since $D_r t >> 0.192 \ln(0.192 + D_r t)$ for large values of $D_r t$, using **Equation 75**,

$$<p^2>(t)_{D_r t \to \infty} \approx \frac{A^2}{5.2(2D_r t)} \int_0^\pi \sin^5 \varphi'' d\varphi'' \qquad (78)$$

which, for $A = 0.5$ ($\theta_{obj} = 180°$), reduces to

$$<p^2>(t)_{D_r t \to \infty} = \frac{A^2}{5.2(2D_r t)} \left(\frac{16}{15}\right) = \frac{1}{39 D_r t}$$

## General expression

Combining the results from the two limits, and assuming a Pade-like expression for $<p^2>$, we obtain:

$$\mathrm{Var}(p)\,(t) = <p^2>(t) \approx \frac{1}{15 + 39 D_r t} \qquad (79)$$

**Appendix 2—figure 3** demonstrates good agreement between **Equation 79** and the simulation algorithm.

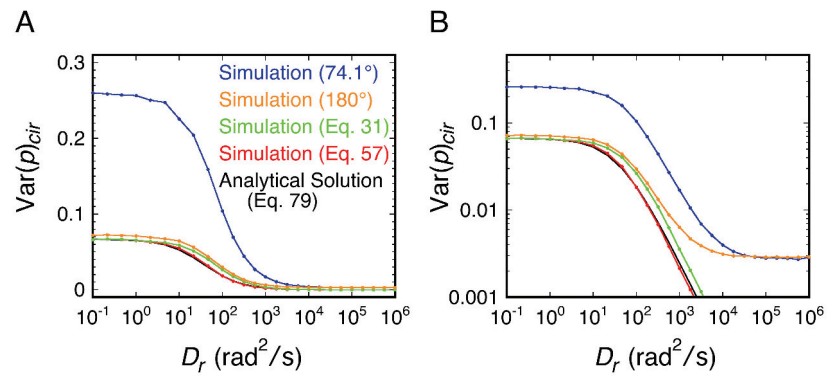

**Appendix 2—figure 3.** Comparison of the relationship between $<p^2>_{cir}$ (=Var$(p)_{cir}$) and $D_r$ as determined analytically and from rotational random walk simulations for an isotropically rotating spherical particle. Results are plotted on both linear (**A**) and log (**B**) ordinate scales. The *red* curve reproduces the *black* curve, providing confirmation of the simulation algorithm. As noted earlier, **Equation 57** requires isotropic excitation and $B = 1$. The relatively small difference between the *red* and *green* curves indicates that the error introduced by reducing **Equation 31** to **Equation 57** is not very high. The *blue* and *orange* curves reach a minimum of ~0.003 at high $D_r t$ values due to the limited photons recovered from the random walk simulations (shot-noise limited). (*black*) **Equation 79**; (*red*) simulation algorithm using **Equation 57** to calculate $p$; (*green*) simulation algorithm using **Equation 31** to calculate $p$; (*orange*) rotational random walk simulation (no noise, no threshold) for a perfect objective ($\theta_{obj}$ = 180°); (*blue*) simulation data from **Figure 2—figure supplement 1C** ($\theta_{obj}$ = 74.1°). $t$ = 10 ms; $N_s$ = 1400; $N$ = 10,000.
DOI: https://doi.org/10.7554/eLife.28716.045

