## [Decision Letter]

Thank you for submitting your article "Investigating molecular crowding within nuclear pores using polarization-PALM" for consideration by *eLife*. Your article has been favorably evaluated by Anna Akhmanova (Senior Editor) and three reviewers, one of whom is a member of our Board of Reviewing Editors. The following individual involved in review of your submission has agreed to reveal his identity: Edward A Lemke (Reviewer #2).

The reviewers have discussed the reviews with one another and the Reviewing Editor has drafted this decision to help you prepare a revised submission.

Summary:

The manuscript by Fu and colleagues describes a polarizing-PALM (p-PALM) approach to analyze the organization of the FG-repeat network within the nuclear pore complex. In the last few years, excellent progress has been made in our understanding of the structural arrangement of the nuclear pore core structure. However, the organization of the FG repeats has been difficult to elucidate as the natively unfolded FG-repeats cannot be studied with traditional structural approaches due to their inherent flexibility. Hence, innovative methods such as the p-Palm approach developed here are needed.

While there were some concerns about the impact of the biological findings within the nuclear pore field all three reviewers agreed that the work is of very high quality, technically profound and felt that the method was extensively validated. Furthermore, the experimental and theoretical pipeline described in this paper could be of broader use to examine also other disordered/dynamic structures in cells. Based on this, the consensus opinion was that the results are appropriate for *eLife* and the reviewers in principle supported publication. However, the manuscript in its present form is very difficult to read and hence it was unclear whether the current paper can be digested by a broader readership, and extensive edits will be needed prior to publication.

Essential revisions:

1) The writing has to be substantially revised in order to make the manuscript accessible to a broader readership. Furthermore, the style of the paper is very heterogeneous and some parts of the paper are almost disconnected from the rest. This is especially obvious for the results presented in the last three subsections of the Results. The conclusions of these considerations should be stated in the paper, but the detailed arguments could be transferred to the supplemental material. The same occurs again in the Discussion, when the pros and cons of the new method are evaluated. The following reorganization and consolidation of topics is suggested:

a) Concept of method;b) Key Readout (note, the result from rotational analysis are mainly negative, still the reader has to go through lots of stuff in the main text). What about:

Key observable: center of distribution from pPalm; width of distribution form pPALM and actual position from PALM;c) A separate pitfall analysis (supplement)?

2) The method that is developed here could be of broader use to examine also other disordered/dynamic structures and, for example, could be applied to analyze the poorly understood organization of membrane-less compartments such as nucleoli, processing bodies, stress granules, splicing bodies etc. This is an obvious strength of the paper yet the authors fail to discuss this and neglect to point this out. They should comment on the broader applicability of their approach in the "Digest" and the "Discussion".

3) The data shown in Figure 4 for the N- and C-terminally tagged version of POM121 is used to argue in favor of POM121 having 2 transmembrane helices placing the N-terminus into the FG-network. The difference in the effect of WGA on both mutants, however, is quite substantial and is not really commented on by the authors. The data in Figure 6 suggest a localization of the N-terminus in the very center of the transport channel (thus supporting the above result). A distinct and significant impact on the mobility by WGA should therefore be expected on the N-terminally tagged POM121, which was not seen. These data appear to be quite inconsistent and the authors need to address this or at least comment on this.

4) Subsection “Spatial Distribution of mEos3 Probes within the FG-Network”, last paragraph: the result on the spatial distribution of RanGAP is very important, but at the same time quite surprising. Can the authors validate these results by an alternative approach?

---

## [Author Response]

Essential revisions:1) The writing has to be substantially revised in order to make the manuscript accessible to a broader readership. Furthermore, the style of the paper is very heterogeneous and some parts of the paper are almost disconnected from the rest. This is especially obvious for the results presented in the last three subsections of the Results. The conclusions of these considerations should be stated in the paper, but the detailed arguments could be transferred to the supplemental material. The same occurs again in the Discussion, when the pros and cons of the new method are evaluated. The following reorganization and consolidation of topics is suggested:a) Concept of method;b) Key Readout (note, the result from rotational analysis are mainly negative, still the reader has to go through lots of stuff in the main text). What about:Key observable: center of distribution from pPalm; width of distribution form pPALM and actual position from PALM;c) A separate pitfall analysis (supplement)?

We have substantially revised the manuscript to improve readability. As suggested, most of the material from the Results was moved to a new supplement (Appendix 1), and briefly discussed in the manuscript (subsections “Effect of Anisotropic Rotation of the Probe on <*p*> and Var(*p*)” and “Effect of Numerical Aperture (NA) on <*p*> and Var(*p*)”). To present a more logical flow, we have also reordered the Results and subdivided it into four sub-sections: 1) the p-PALM method; 2) the application of p-PALM to NPCs; 3) PALM on NPCs; and 4) combined PALM and p-PALM measurements. Some of the figures and figure supplements were reordered to accommodate these changes and they have been reduced in number by transferring some figures to Appendix 1.

The original Appendix containing mathematical derivations has been renamed Appendix 2, and edited for clarity. The Discussion was shortened by deleting less important material, and clarified in numerous locations. The primary readouts were explicitly identified as the average polarization, <*p*>, and the variance of the polarization distribution, Var(*p*). In addition, we note that the overall shape of polarization histograms and photon scatterplots can provide additional clues as to the underlying physical constraints on the probe’s rotational mobility (subsection “Outline of the p-PALM Method”).

2) The method that is developed here could be of broader use to examine also other disordered/dynamic structures and, for example, could be applied to analyze the poorly understood organization of membrane-less compartments such as nucleoli, processing bodies, stress granules, splicing bodies etc. This is an obvious strength of the paper yet the authors fail to discuss this and neglect to point this out. They should comment on the broader applicability of their approach in the "Digest" and the "Discussion".

We are grateful that the reviewers encouraged us to promote the applicability of our work. We now briefly mention in the Abstract the suitability of the p-PALM method for probing other crowded environments, and specifically identify its application to membrane-less compartments at the end of the Discussion.

3) The data shown in Figure 4 for the N- and C-terminally tagged version of POM121 is used to argue in favor of POM121 having 2 transmembrane helices placing the N-terminus into the FG-network. The difference in the effect of WGA on both mutants, however, is quite substantial and is not really commented on by the authors. The data in Figure 6 suggest a localization of the N-terminus in the very center of the transport channel (thus supporting the above result). A distinct and significant impact on the mobility by WGA should therefore be expected on the N-terminally tagged POM121, which was not seen. These data appear to be quite inconsistent and the authors need to address this or at least comment on this.

The PALM density maps were generated by determining the position of the nuclear envelope, and by aligning clusters of spots based on their centroids. This procedure positioned all cluster centroids on the central axis of the NPC. This approach was necessary since there were at most a few tens of spots per NPC, and we had no independent marker for the NPC scaffold. Thus, while the axial dimension was calibrated based on the nuclear envelope position, the lateral dimension was artificially squeezed. In short, tags on the periphery of the pore can appear to be centrally distributed in the PALM density maps. Thus, the Pom121 maps are not inconsistent with a model wherein the N-terminus of Pom121 is near the scaffold and the C-terminus is in the center of the pore (Figure 4). This issue has now been clarified in the text (subsection “Spatial Distribution of mEos3 Probes within the FG-Network”, second paragraph).

In the presence of WGA, the difference in Var(*p*)_cir_ values for the probe on the N-and C-termini of Pom121 predicts a difference in *D_r_* values of about 2-fold (Figure 2), which is actually not that substantial, especially considering that this difference can be readily explained by a differential distribution of the probe molecules between two sub-populations (see Figure 6 and supplements). The Nup98 data show less effect of WGA near the scaffold (Figure 5), suggesting that the same would be true for Pom121. The mixed population hypothesis has now been evaluated further for the probes on Pom121 and Nup98 under wildtype conditions, and both proteins yield results consistent with higher crowding near the NPC scaffold (Discussion, eighth paragraph), assuming the 2 TMD model in Figure 4 for Pom121. We therefore disagree that the results on Pom121 are inconsistent, as suggested.

4) Subsection “Spatial Distribution of mEos3 Probes within the FG-Network”, last paragraph: the result on the spatial distribution of RanGAP is very important, but at the same time quite surprising. Can the authors validate these results by an alternative approach?

We agree that the RanGAP distribution is surprising, and we have expanded our discussion of these data (Discussion, third paragraph). We do not feel that we can further appropriately address this issue in the short period allotted to develop a revised version of this manuscript. Nonetheless, this will be an important issue for us moving forward as we develop a super-resolution 3D method for examining particle distributions in NPCs.